# Flood risks to the financial stability of residential mortgage borrowers: An integrated modeling approach

Kieran P. Fitzmaurice[1,2], Helena M. Garcia[3], Antonia Sebastian[3,4], Hope Thomson[5], Harrison B. Zeff[1,2], Gregory W. Characklis[1,2]

[1]Institute for Risk Management and Insurance Innovation, University of North Carolina at Chapel Hill, 27516, USA
[2]Department of Environmental Sciences and Engineering, University of North Carolina at Chapel Hill, 27516, USA
[3]Environment, Ecology, and Energy Program, University of North Carolina at Chapel Hill, 27514, USA
[4]Department of Earth, Marine, and Environmental Sciences, University of North Carolina at Chapel Hill, 27514, USA
[5]School of Government Environmental Finance Center, University of North Carolina at Chapel Hill, 27514, USA

*Correspondence to*: Kieran P. Fitzmaurice (kieranf@ad.unc.edu)

**Abstract.** Property damage from flooding creates urgent funding needs that uninsured households often struggle to meet, particularly when access to affordable credit is limited. While prior research links floods to higher rates of financial distress, little is known about the prevalence and drivers of credit constraints among flood-exposed property owners. In this study, we use a simulation-based approach to estimate the impact of uninsured damage on residential mortgage borrowers' financial conditions over a series of floods in North Carolina from 1996-2019. Our framework estimates key variables (e.g., damage cost, property value, mortgage balance) to project the number of flood-exposed borrowers experiencing credit constraints due to negative equity, liquidity issues, or both in combination. Conservative projections suggest that the seven floods evaluated generated $4.0 billion in property damage across the study area, of which 66% was uninsured. Among flood-affected mortgage borrowers, only 48% had insurance, and 32% lacked sufficient income or collateral to finance repairs through home equity-based borrowing, making their recovery uncertain. By identifying which borrowers are likely to have unmet funding needs due to flood-related credit constraints, these results can inform the design of interventions to improve the financial resilience of flood-prone households.

## 1 Introduction

Flooding is one of the most frequent and costly natural hazards in the United States, causing billions of dollars in property damage each year (Smith, 2020). Unlike other perils such as wind and fire, flood damage is generally not covered by standard homeowners' insurance and must be purchased as a separate policy (Marcoux and Wagner, 2025). This has contributed to a substantial protection gap, with a recent study estimating that 70% of annual U.S. flood losses are likely to be uninsured (Amornsiripanitch et al., 2025). Uninsured losses can magnify the financial impact of flood events by exposing property owners to substantial out-of-pocket costs. Households that lack insurance are typically forced to exhaust savings, take on debt, apply for government aid, or rely on friends and family in order to fund repair and recovery efforts, which increases

the potential for unaddressed funding needs (Collier and Kousky, 2025; Kousky et al., 2021; You and Kousky, 2024). For those without sufficient funding for repairs, a flood can act as a triggering event that pushes households into a state of financial distress characterized by adverse credit outcomes and housing insecurity (Ratcliffe et al., 2020).

Low-interest forms of debt financing such as federal disaster loans and home equity loans may provide uninsured property owners with the means to cope with flood-related financial shocks by providing an immediate infusion of funds for repairs that can be repaid gradually over many years. However, many property owners face credit constraints that limit their ability to access loans and take on additional debt. Approval rates for federal disaster loans vary substantially based on an applicant's income and credit score, with overall denial rates often exceeding 40% during major disasters (Begley et al., 2023;

Ellis and Collier, 2019). Home equity loans are the next-cheapest form of credit available to property owners but have strict income and collateral requirements. Lenders are typically unwilling to approve a loan that would cause the total amount of debt secured by a property to exceed its market value, or which would cause a borrower's total monthly debt obligations to exceed a set percentage of their income. For uninsured property owners, these collateral and income constraints can severely limit the amount they are able to borrow for repairs, particularly when they have high levels of preexisting mortgage debt or

severe damage to their property.

Understanding how income and collateral constraints affect the ability of uninsured property owners to cope with flood-related financial shocks is important for designing effective policy responses. While several empirical studies offer evidence that less insured and less creditworthy households exhibit higher levels of financial distress following disasters (Billings et al., 2022; Collier et al., 2024; You and Kousky, 2024), few studies have attempted to quantify the prevalence and

underlying drivers of credit constraints among flood-exposed property owners. Data limitations may be a contributing factor to these knowledge gaps: understanding whether a property owner has sufficient borrowing capacity to fund their recovery requires granular data on their income, debt obligations, property value, and level of uninsured damage exposure—information which is rarely captured by a single comprehensive dataset. In this context, simulation-based modeling approaches can help to address data scarcity issues by integrating data from multiple heterogeneous sources and explicitly representing the processes

that give rise to post-flood credit constraints, allowing researchers to estimate the financial impacts of flood events in settings where direct observations are unavailable.

In this study, we use an integrated modeling approach to simulate the impact of flood-related property damage on residential mortgage borrowers' financial conditions over a series of floods in North Carolina from 1996-2019 while examining the following research questions: (1) How much of the damage from these events was uninsured? (2) What share of flood-

exposed borrowers faced credit constraints that were likely to impair their ability to access home repair loans? (3) Were these credit constraints driven by insufficient income, insufficient collateral, or both in combination? To answer these questions, our modeling framework combines property-level estimates of flood damage, insurance coverage, and time-varying property value with neighborhood-level data on borrower characteristics and pre-flood financial conditions. We anticipate that our proposed approach and subsequent findings can be used to help inform the design of policies to improve the resilience of U.S. households

to floods and other natural disasters.

## 2 Background: Financing recovery

Property damage from flooding represents an unexpected shock to household finances that requires affected property owners to access large amounts of liquid funds to cover costs associated with repairs, which can easily run into the tens of thousands of dollars. To address this financing challenge, property owners can draw from a variety of funding sources, including insurance, government aid, secured debt, unsecured debt, personal savings, and informal borrowing from family and friends. Many property owners will only have access to a subset of these funding sources, which differ in their eligibility criteria, amount of funding provided, and costs associated with utilization. In this section, we provide a review of the sources of financing used by property owners to fund flood-related repairs and their implications for household financial stability.

For property owners with flood insurance, funding for repairs typically comes in the form of an insurance payout. The primary source of flood insurance in the U.S. is the federally-operated National Flood Insurance Program (NFIP), which in 2024 provided coverage to 4.8 million policyholders nationwide while collecting $4.0 billion in written premiums (FEMA, 2025c). During the 2000-2024 period, the median (IQR) claim payout provided to NFIP policyholders was $15,600 ($1,500-$70,000) in 2020 dollars (FEMA, 2025b). This infusion of funds helps to minimize the amount of disruption to household finances and typically allows property owners to pay for repairs without having to draw from savings or take on debt. In a survey of hurricane-exposed households, You and Kousky (2024) observed that those with insurance had better self-reported financial health in the year following the disaster and were much less likely to have unmet funding needs.

During major flood events, the majority of losses are typically uninsured: catastrophe modeling firms estimate that less than a third of all damage to residential properties from flooding during Hurricanes Helene, Florence, Irma, and Harvey was covered by insurance (CoreLogic, 2024; Reuters, 2017a, b; RMS, 2018). The reasons for this protection gap are multifaceted, but can be broadly attributed to the following factors: (1) purchase of flood insurance is voluntary for properties located outside of zones designated as Special Flood Hazard Areas (SFHAs) by the Federal Emergency Management Agency (FEMA) that have an estimated annual chance of flooding of 1% or greater; (2) the use of SFHA status as an indicator of flood risk can lead property owners outside the SFHA to believe insurance is unnecessary (Horn, 2022a); and (3) many households have limited ability or willingness to pay for flood insurance (Atreya et al., 2015; Kousky, 2011; Netusil et al., 2021), which poses a major barrier to increasing uptake. Since the implementation of a new rate-setting methodology known as Risk Rating 2.0 in April 2022, approximately 77% of NFIP policyholders have seen their premiums rise, sparking concerns that higher costs may prompt even more property owners to forgo flood coverage (Frank, 2022; Horn, 2022b).

Even when purchase of flood insurance is mandatory, instances of noncompliance are common (GAO, 2021; HUD, 2020). Those with mortgages backed by the federal government are typically required to purchase and maintain flood insurance if their property is located in a SFHA (GAO, 2021). Nationwide, approximately 70% of single-family mortgages receive federal backing through agencies such as the Federal Housing Administration (FHA) and government-sponsored enterprises (GSEs) such as Fannie Mae and Freddie Mac (GAO, 2019). Yet, challenges in enforcing mandatory purchase requirements have contributed to noncompliance among mortgage borrowers. A recent study by the U.S. Department of Housing and Urban

Development (HUD) found that in a sample of FHA-insured mortgages in North Carolina, less than half of those required to carry flood insurance actually had it (HUD, 2020). Prior studies suggest that borrowers often fail to renew their insurance in the years following mortgage origination: an analysis by Michel-Kerjan et al. (2012) observed that the median tenure of an NFIP policy (2-4 years) was far shorter than the median housing tenure (5-6 years) over the 2001-2009 period, implying that many policyholders allow their policy to lapse while remaining in their residence; similarly, a study by the Government Accountability Office found that only 72% of newly purchased properties located in SFHAs that had NFIP policies originated in 2014 were still covered by an NFIP policy in 2019 (GAO, 2021). In addition, the binary nature of SFHA boundaries means that there are many properties located just outside the SFHA that are not required to purchase flood insurance despite facing substantial risk from both larger return period events (e.g., a 1-in-200-year flood) and pluvial flood hazards that are not represented on existing maps (Brody et al., 2018; Pricope et al., 2022; Sebastian et al., 2021). As a result, uninsured damage from flooding remains a major financial threat to mortgage borrowers and (by extension) their lenders (CBO, 2023; Thomson et al., 2023).

For uninsured property owners, the primary sources of federal post-disaster aid are disaster loans from the U.S. Small Business Administration (SBA) and, to a lesser extent, individual assistance grants from FEMA (Horn, 2018). Grants provided by FEMA's Individuals and Households Program (IHP) are intended to meet basic needs and typically cover only a small fraction of the total cost of flood-related damages (Lindsay and Webster, 2022). Between 2002 and 2024, the median (IQR) IHP grant for property owners reporting flood damage to their primary residence was only $2,900 ($930-$7,510) in 2020 dollars (FEMA, 2025a). Disaster loans offered through the SBA provide a larger infusion of funds, have low interest rates (typically around 50% of the average 30-year mortgage rate), long repayment terms, flexible collateral requirements, and play an important role in recovery; however, many applicants are denied a loan due to unsatisfactory credit history or insufficient income (Ellis and Collier, 2019; Lindsay and Webster, 2022). A key metric used by the SBA to evaluate loan applicants is their debt-to-income (DTI) ratio, which describes the share of their monthly income that goes towards recurring debt obligations (e.g., payments on mortgage and auto loans). In an analysis of SBA loan applications from the 2005-2013 period, Collier et al. (2024) observed a sharp decrease in the probability of loan approval for applicants with a DTI ratio exceeding 40%: those with a DTI ratio just over this threshold were much less likely to receive a loan than those with a DTI ratio just under it (60% vs. 80% approved), with approval rates dropping below 50% for applicants with a DTI ratio of 45% or greater. Using a regression discontinuity design, the authors of this study were able to estimate the causal effect of access to low-interest forms of credit on recovery, finding that those who qualified for an SBA loan were far less likely to experience negative financial outcomes such as bankruptcy and mortgage delinquency in the years following a disaster. These findings underscore the contribution of income-related credit constraints to financial distress among property owners exposed to uninsured damage.

After SBA loans, the next-cheapest form of credit available to uninsured property owners are home equity loans. These loans are a form of secured debt that requires property owners to pledge their property as collateral, giving lenders the right to initiate foreclosure proceedings should they fail to repay. As such, the maximum amount that a property owner can borrow via this route is limited by both their income and the amount of equity they have in their home (i.e., the difference

between its market value and the amount of outstanding debt secured by the property). Applicants for home equity loans are evaluated based on their credit score, DTI ratio, and combined loan-to-value (CLTV) ratio, which is calculated by dividing the total balance of all loans secured by the property by its market value. The maximum allowable DTI and CLTV ratios for most conventional loans are 45% and 97% respectively, though certain government lending programs will allow disaster-affected property owners to borrow at CLTV ratios of up to 100% (Fannie Mae, 2024a; HUD, 2024). In a study of Houston-area residents during Hurricane Harvey, Billings et al. (2022) observed a significant increase in the use of home equity loans among property owners living in flooded areas outside the SFHA, where insurance penetration is low; this effect was strongest for those who were unlikely to qualify for an SBA loan. This suggests that property owners who are denied an SBA loan can sometimes use home equity loans as a substitute source of funding for recovery; however, it is important to note that there are disadvantages to this strategy. Home equity-based borrowing is a more expensive form of debt than SBA loans and is also constrained by the amount of equity a borrower has in their home. Prior studies suggest that flood events can depress property values in affected areas (Bin and Landry, 2013; Fang et al., 2023; Ortega and Taşpınar, 2018), though disentangling the direct effects of flood damage from changes in market perceptions of risk can often be a challenge (Atreya and Ferreira, 2015). Post-flood declines in property value can reduce a property owner's equity at the critical moment when it is needed as collateral for loans, potentially impeding their ability to finance repairs, especially when the amount of damage is severe. For property owners with low levels of pre-flood equity (e.g., those who recently purchased their home with a mortgage), the combination of uninsured damage and property value reductions can lead to a situation of "negative equity" in which the amount of debt secured by their property exceeds its market value, leading to a CLTV ratio greater than 100%. This prevents them from using their property as collateral to obtain additional loans and can also create a financial incentive to default on their existing debt obligations (Foote and Willen, 2018). These factors may help to explain the elevated rates of bankruptcy observed by Billings et al. (2022) among property owners with an outstanding mortgage and limited capacity to take on additional debt.

For property owners who are unable to access SBA loans or home equity loans, the available financing options narrow considerably, often leaving only higher-cost sources of funds that impose difficult tradeoffs between meeting urgent recovery needs and preserving long-term financial stability. Credit cards and other forms of unsecured debt can provide households with quick access to funds but have low borrowing limits and high interest rates. In a study of Houston residents following Hurricane Harvey, Del Valle et al. (2022) found that households in flooded areas tended to utilize credit cards strategically by taking advantage of promotional-rate offers and quickly repaying balances to avoid accruing interest. This study found that post-Harvey increases in credit card spending were largely offset by card payments, consistent with the earlier findings of Gallagher and Hartley (2017) who found that Hurricane Katrina had only a modest and short-lived impact on the credit card balances of New Orleans residents. Together, these two studies suggest that credit cards provide short-term liquidity to disaster-affected households but do not constitute a major source of funding for flood-related repairs.

In a survey of hurricane-exposed households, You and Kousky (2024) found that over half reported using their savings to fund repairs. While savings are a cost-effective way to manage small emergencies, many property owners are unlikely to have sufficient liquid savings to cover the full cost of flood-related repairs. In the Federal Reserve's 2024 Survey of Household

Economics and Decisionmaking, 43% of respondents with a mortgage reported that they would be unable to cover a $2,000 emergency expense using only their savings (Board of Governors of the Federal Reserve System, 2025). Households might also use retirement accounts to fund repairs when other savings prove insufficient: Deryugina et al. (2018) observed a large increase in withdrawals from retirement accounts among New Orleans residents following Hurricane Katrina. Among households included in the Federal Reserve's 2022 Survey of Consumer Finances, the median retirement account balance was $86,900 (Aladangady et al., 2023)—an amount that would be sufficient to cover the typical cost of most flood-related repairs. However, draining retirement accounts to fund repairs is likely to be a last resort for most property owners, as this strategy is likely to have negative consequences for their long-term financial health, particularly for those who are at or near retirement age. While property owners may be able to supplement their savings with financial support from family and friends (You and Kousky, 2024) and social safety net transfers (Deryugina, 2017), it is unclear whether these sources provide sufficient funds to meet the recovery needs of those with severe damage to their residence.

When these funding sources fall short, unmet repair needs can destabilize household finances. Numerous studies have linked floods and hurricanes to higher rates of mortgage delinquency (Calabrese et al., 2024; Du and Zhao, 2020; Kousky et al., 2020; Mota and Palim, 2024; Rossi, 2021) and personal bankruptcy (Billings et al., 2022; Collier et al., 2024), with effects varying based on households' access to insurance and affordable credit. After Hurricane Harvey, Kousky et al. (2020) found that mortgaged properties with moderate to severe flood damage had over double the odds of becoming 180 or more days delinquent than undamaged properties—a relationship significant only outside the SFHA, where insurance uptake is low. Similarly, Billings et al. (2022) documented higher rates of bankruptcy and credit delinquency among Harvey-affected households outside the SFHA. This study found that post-Harvey increases in bankruptcy were largely concentrated in a specific segment of the population: mortgage borrowers located outside the SFHA with below-median incomes and credit scores. Property owners in this group faced high levels of uninsured damage but had limited ability to finance repairs through additional borrowing. Collectively, these studies suggest that uninsured property owners experience lasting financial consequences from flooding, particularly when income or collateral constraints prevent them from accessing low-cost forms of debt financing. Despite growing awareness of these risks, few efforts have been made to systematically measure the prevalence, severity, and drivers of credit constraints among flood-exposed households at scale.

## 3 Methods

This analysis uses a data-driven modeling framework to estimate dynamic changes in the financial condition of residential mortgage borrowers in response to uninsured property damage incurred over a series of floods concentrated in the eastern part of the U.S. state of North Carolina (Fig. 1). The approach extends the framework used by Thomson et al. (2023) to estimate the prevalence of negative equity following Hurricane Florence in 2018 by including other drivers of credit constraints among flood-exposed mortgage borrowers (e.g., insufficient income) and by capturing the cumulative effects of multiple flood exposures occurring over a series of seven named tropical cyclones (including Florence) during the 1996-2019 period. The

initial financial conditions of mortgage borrowers are simulated using loan-level data on household income, monthly debt obligations, unpaid mortgage balance, and loan structure at the time of origination. Temporal changes in property values and home equity are simulated based on local trends in real estate prices and empirically observed repayment profiles in a sample of mortgages purchased by Fannie Mae and Freddie Mac. This information is combined with property-level NFIP policy enrollment data and flood damage estimates to assess the impact of financial shocks from uninsured flood damages on borrower equity and liquidity. Finally, we calculate post-damage adjusted combined loan-to-value (ACLTV) and adjusted debt-to-income (ADTI) ratios to estimate the number of flood-exposed borrowers facing credit constraints due to insufficient income, insufficient collateral, or both in combination.

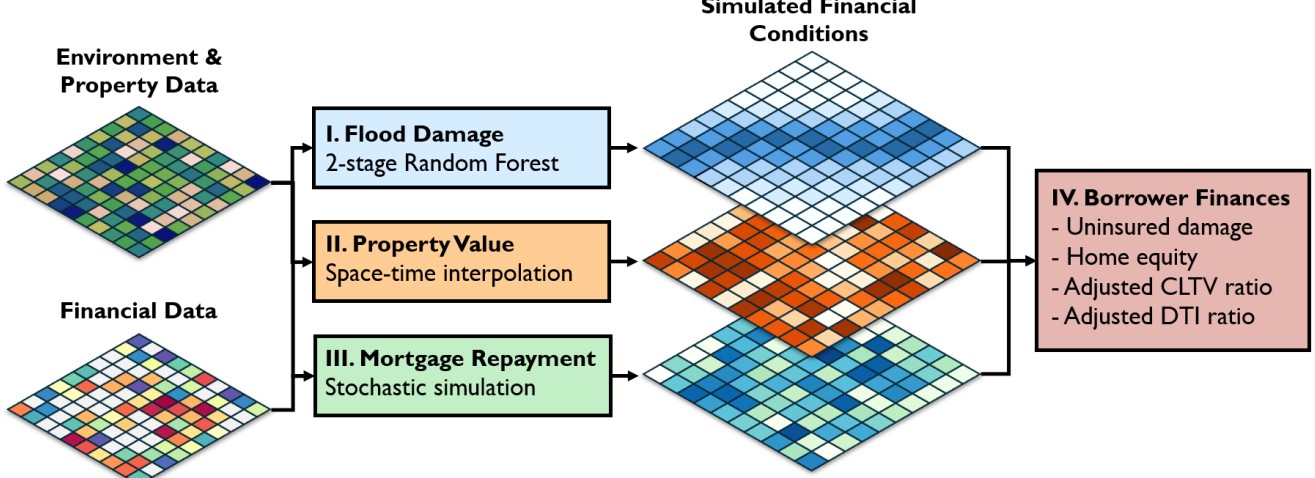

**Figure 1.** Overview of the integrated modeling framework. The top left grid represents environmental and building structural data (available at each property), while the bottom left grid represents financial data (available for a subset of properties). The various sub-models used to create spatially complete estimates of borrower financial conditions are denoted by colored boxes.

### 3.1 Study area and period

This study examines the historical impact of multiple flood events on the U.S. state of North Carolina during the 24-year period from 1996 through 2019. This region is home to over 10 million people, of whom approximately 4% live within the 100-year floodplain (NYU Furman Center, 2017; U.S. Census Bureau, 2023). Based on data from FEMA and the North Carolina Department of Emergency Management (NCEM), we estimate that only 47% of buildings inside the SFHA were covered by an NFIP policy in 2019, implying that many property owners lack financial protection despite facing substantial flood risk (FEMA, 2025c; NCEM, 2022). This is consistent with nationwide rates of flood insurance uptake inside the SFHA observed by Brandt et al. (2021). North Carolina frequently experiences flooding associated with tropical cyclones, which accounted for 14 major disaster declarations in the state between 1996 and 2019 (FEMA, 2024). Our analysis focuses on the seven largest named storms during this period as measured by the number of associated NFIP claims filed in North Carolina: Hurricanes Fran (September 1996), Bonnie (August 1998), Floyd (September 1999), Isabel (September 2003), Irene (August

2011), Matthew (October 2016), and Florence (September 2018). When evaluating the financial impact of these events, we restrict our focus to the 78 North Carolina counties for which we had access to address-level information on flood insurance coverage (Fig. 2); this region—hereafter referred to as the "study area"—encompasses 86% of the state's population and 82% of the land area (U.S. Census Bureau, 2023).

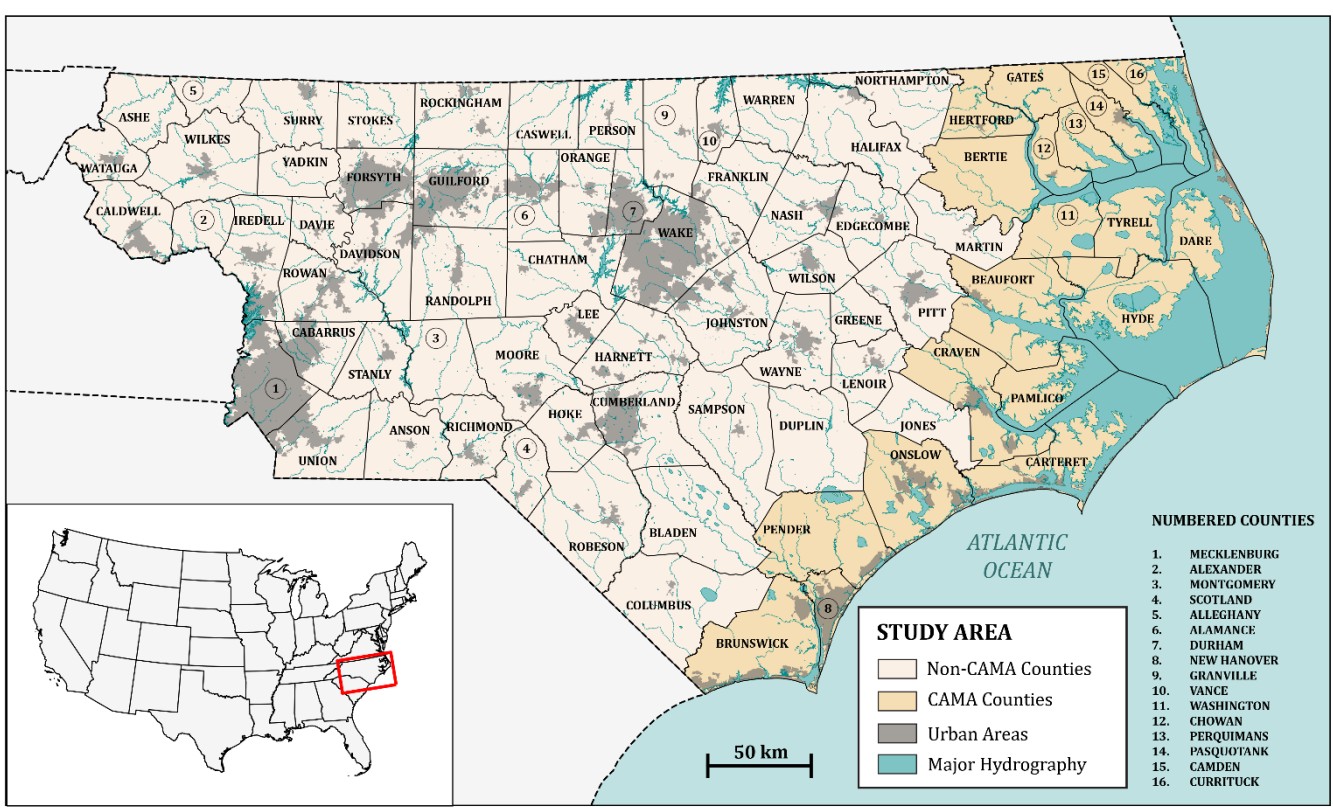

**Figure 2.** Overview map of the study area. County boundaries are shown in black, with counties under the jurisdiction of the Coastal Area Management Act (CAMA) colored in a darker shade of tan than non-CAMA counties. Major waterbodies are depicted in blue. Urban areas are shaded in dark grey.

## 3.2 Modeling framework overview

The focus of our analysis is on residential mortgage borrowers living in single-family detached homes. We selected this property type due to its ubiquity in North Carolina and because other property categories such as multifamily housing and mobile home parks exhibit complex ownership and disaster recovery patterns that require specialized modeling approaches (Mongold et al., 2024; Moradi and Nejat, 2020; Rumbach et al., 2020). In addition, we further restrict our focus to mortgage loans secured by a borrower's primary residence and thus exclude those associated with investment properties or secondary residences. Based on data from the 2021 American Housing Survey, we estimate that owner-occupied single-family detached homes account for 53% of all housing units and 84% of housing units with a mortgage in the South Atlantic Census Bureau Division (which includes North Carolina) (U.S. Census Bureau, 2021).

Our modeling framework combines property characteristics (e.g., square footage, foundation type, first floor elevation) and data on environmental variables affecting flood hazard (e.g., distance to rivers, height above nearest drainage) with financial observations (e.g., insurance claims, property sales, mortgage originations) to simulate the financial conditions of residential mortgage borrowers at a monthly timestep. This is accomplished through a series of sub-models describing: flood damage exposure (model I, Sect. 3.3); property value dynamics (model II, Sect. 3.4); simulated mortgage repayment profiles (model III, Sect. 3.5); and damage-adjusted measures of borrower equity and liquidity over the study period (model IV, Sect. 3.6) (Fig. 1). Where possible, our modeling framework incorporates property-specific data (e.g., structure characteristics, past sales); certain variables that are only available at the census tract level (e.g., mortgage loan characteristics) are stochastically sampled to create synthetic values for individual properties. As such, the estimates produced by our simulation model do not represent the exact conditions experienced by any specific borrower but are intended to reflect the distribution of key financial variables within a given census tract—a spatial resolution that is likely to be relevant for the targeting of policy interventions and post-disaster aid.

## 3.3 Model I: Flood damage

Unlike insured losses, uninsured damage from flooding is rarely tracked and often difficult to quantify. To overcome this data gap, we use a random forest machine learning model trained on NFIP policy and claim data to estimate damage to uninsured properties as a function of geospatial predictors. This method closely resembles the approach employed by Thomson et al. (2023) to estimate uninsured damage to properties during Hurricane Florence, and builds on a foundation of research that has utilized random forest models to construct spatially complete maps of historical flood exposure from sparse observations (Collins et al., 2022; Garcia et al., 2025; Mobley et al., 2021). For each flood event evaluated, we employ a two-stage approach to predict the presence of flooding (model I, stage I) and magnitude of damage (model I, stage II) at each property within the study area.

The location and structural characteristics (e.g., foundation type, first floor elevation) of each individual property are specified using a statewide building inventory complied by NCEM's GIS team (NCEM, 2022) that represents an approximate
snapshot of the building stock during the middle of the study period. This database includes information on occupancy classifications that allow for a distinction between various types of residential and commercial structures. It is spatially joined to a statewide parcels dataset that delineates the boundaries of individual properties (NC OneMap, 2022). For properties with multiple structures (e.g., a main building and an outbuilding), property characteristics are evaluated based on the structure with the largest aerial footprint. For properties missing data on key attributes (e.g., year built) missing values were spatially imputed
using nearest-neighbor interpolation. The values of environmental variables affecting flood hazard (e.g., rainfall, elevation, distance to nearest stream) are estimated at each property location from a variety of sources, including the Daymet V4 meteorological dataset (Thornton et al., 2022), North Carolina Flood Risk Information System (NC Floodplain Mapping Program, 2022), National Elevation Dataset (Gesch et al., 2018), National Hydrography Dataset (Moore et al., 2019), and National Land Cover Database (Homer et al., 2012). In total, a set of 17 variables describing the structural and environmental
characteristics of each property are included as predictors when estimating the presence and magnitude of flood damage. A complete list of these variables and their data sources is available in Table S1.

In the first stage of the damage estimation procedure (model I, stage I), a random forest machine learning model is trained with address-level NFIP policy and claims data to predict the presence or absence of flood damage as a function of environmental and property characteristics. The insurance status of each property at the time of an event is determined based
on records of NFIP policies and filed claims provided by FEMA Region IV (Text S1, Table S2). Observations from insured properties are used to train a random forest model predicting the probability of experiencing property damage during a specific flood event: properties filing a flood insurance claim during the event are labeled as "presence" points (i.e., locations with known flood damage exposure) while those with an active flood insurance policy, but no claims, are labeled as "absence" points (i.e., locations known to have not experienced flood damage). Due to NFIP record-keeping practices, a substantial
fraction of policy records (but not claims) from the pre-2009 period are missing from the address-level dataset provided by FEMA (NFIP, 2024); to address this imbalance in our dataset, we randomly select "pseudo-absence" points from the remaining properties in the study area using geographically stratified sampling to ensure that the number of presence and absence points within each geographic unit (e.g., the intersection of census tract and SFHA polygons) matches the totals implied by auxiliary sources of data—in this case, anonymized NFIP policy enrollment statistics obtained from FEMA (a detailed description of
the process is provided in the supplementary information Text S1 and Table S2). As such, the machine learning method for predicting the presence of flood damage is a hybrid approach that relies on a combination of actual presence-absence and stochastically generated pseudo-absence training data (Barbet-Massin et al., 2012). While the inclusion of pseudo-absences likely introduces some label noise into the training data, this step was necessary to correct for the bias inherent in the address-level insurance data, which disproportionately captured damaged (claim) locations. For each event, the trained random forest
model is used to classify uninsured properties as "damaged" or "undamaged." For further information on the implementation and performance of this method, we refer the reader to Garcia et al. (2025).

In the second stage of the damage estimation procedure (model I, stage II), a second random forest model is trained with NFIP claims data to predict the dollar amount of flood damage at properties classified as "damaged" in stage one. Unlike many physics-based inundation models, our data-driven approach does not explicitly model the depth of floodwaters at each property location; as such, the depth-damage relations commonly used to estimate damage to building structure and contents from flooding are not applicable here (Wing et al., 2020). Instead, observed claims at insured properties are used to train an event-specific random forest regression model predicting the dollar cost of flood damage among damaged properties as a function of structural and environmental characteristics (e.g., first floor elevation, proximity to sources of flooding). The trained model is subsequently used to estimate the cost of flood damage at each uninsured property classified as "damaged."

The performance of the two-stage damage estimation approach is evaluated using both random and spatial block cross-validation. When validation data are randomly selected for cross-validation from the entire spatial domain, training and validation data from nearby locations can exhibit dependence (i.e., spatial autocorrelation), leading to overly-optimistic estimates of model error when applied to more spatially distant locations (Roberts et al., 2017). To address this concern, random cross validation was used to assess the "interpolation" error of the model (i.e., how well it performs in areas with a high density of training examples, such as inside the SFHA) while spatial block cross validation was used to assess the "extrapolation" error of the model (i.e., how well it performs in areas with a lower density of training examples, such as outside the SFHA). Random cross validation was performed by splitting the presence-absence and pseudo-absence data into 10 equally sized random subsets (folds) and repeating model training 10 times, each time withholding one subset to validate the prediction results. Spatial block cross validation was performed by dividing the presence-absence and pseudo-absence data into $n$ spatial blocks defined by 5 km square grid cells, and repeating model training $n$ times, each time withholding one block for validation while also excluding adjacent blocks from the data used for training. This procedure was performed separately for each of the seven flood events included in this study to produce event-specific estimates of model performance (Figs. S2-S4). In addition, a subgroup analysis was performed to assess potential differences in model error inside and outside of the SFHA, and when pseudo-absences are excluded from the validation data.

In random cross-validation, the area under the receiver-operating characteristic curve (AUC) was between 0.86 and 0.95 for all included events, suggesting that the random forest model is able to clearly distinguish between damaged and undamaged properties across a variety of time periods and settings (Fig. S2, Table S3). This result was consistent inside and outside of the SFHA (AUC score of 0.85-0.94 inside vs. 0.83-0.97 outside across all storms) and when pseudo-absences were excluded from the validation data (AUC score of 0.75-0.95) (Figs. S2-S3). When identifying damaged properties, the model exhibited high accuracy (≥92%) and specificity (≥98%) but low sensitivity, with true positive rates of between 12% and 42% across events. This behavior is characteristic of machine learning classifiers trained on class imbalanced data where the positive class (e.g., presence of flood damage) is rare compared to the negative class (Haixiang et al., 2017; He and Cheng, 2021). Among properties that were misclassified by our model in cross-validation, false positive and false negative predictions respectively accounted for 12% and 88% of model errors across the seven evaluated events (Table S4). Collectively, these results suggest that our model often fails to detect properties that were damaged, which is likely to lead to a systematic

underestimation of the true level of flood exposure within the study area. As such, our projections of flood damage exposure (and, by extension, flood-related credit constraints) should be interpreted as a conservative bound as opposed to a central estimate.

In spatial block cross-validation, AUC scores were marginally lower for all events, ranging between 0.79 and 0.92 (Fig. S4, Table S3). This slight decrease in performance is expected and suggests that our random forest model can still distinguish between damaged and undamaged properties at spatially distant locations that are 5 km away from the nearest training datapoint. The accuracy and specificity of the model remained high across events ($\geq$90% and $\geq$98% respectively); however, there was a notable decrease in sensitivity associated with the transition from random to spatial block cross-validation. This effect was strongest for Hurricane Bonnie, the smallest event included in terms of NFIP claims, which had a sensitivity of <1% in spatial block cross-validation. Across the other 6 events, sensitivity ranged between 1% and 33% in spatial block cross-validation (Fig. S4).

The two-stage damage estimation procedure is assessed in terms of its ability to predict the dollar amount of flood damage to individual properties based on the out-of-sample coefficient of determination ($R_{os}^2$) calculated via cross-validation. At the individual property level, the model could only explain a small fraction of the total variance in damage costs ($R_{os}^2 \leq$ 0.39 across all events). The low observed $R_{os}^2$ scores likely arise at least in part due to the low sensitivity of the random forest model used to classify properties as "damaged" in the first stage of the damage estimation procedure: those classified as "undamaged" are assigned a damage cost value of zero in the second stage of the procedure regardless of their actual flood damage status. Among damaged properties that were correctly classified as such, damage cost $R_{os}^2$ scores were typically higher but still exhibited substantial variation, ranging between 0.03 and 0.45 across events (Table S3). One potential source of uncertainty in our damage cost estimates is that our data-driven approach does not explicitly account for the impact of key hydrodynamic variables (e.g., water depth, flow velocity, and duration) that have been shown to play an important role in determining the cost of flood-related damage (Amadio et al., 2019). The damage cost estimates produced by the model are more consistent with cross-validation targets when aggregated across spatial blocks defined by 5 km grid cells, with $R_{os}^2$ scores ranging between 0.52 and 0.93 (Fig. S5). These results suggest that while damage cost predictions at individual properties are highly uncertain, our method can produce reasonable estimates of neighborhood-level damage in an efficient manner.

### 3.4 Model II: Property value

The time-varying market value of each property included in the analysis is estimated across the study period on a quarterly basis using a dataset of residential real estate sales acquired from ATTOM Data Solutions (ATTOM, 2021). This dataset includes 2.3 million property transactions from North Carolina during the 1990-2019 period, and contains information on the property location, sale price, and date on which the transaction occurred. Property transactions were geolocated to building footprints via a two-step process: (1) transactions were first spatially joined to parcels based on the reported latitude

and longitude in the ATTOM dataset, and (2) each transaction's location was then refined to correspond to the largest building footprint within the associated parcel. The parcel and building datasets used in this process were the same as those described in Section 3.3. After discarding transactions that were not from single-family detached homes or which had missing data, the final dataset consisted of 1.8 million geolocated property sales.

Sale price data are only observed for a small fraction of properties within a given year but can be interpolated across space and time to estimate the value of properties with no recent sale transactions. To this end, a hedonic pricing model utilizing a random forest regression kriging (RFRK) method is used to predict home values as a function of property-specific characteristics (e.g., lot size, year built) while accounting for spatial and temporal autocorrelation in home prices. Hedonic models are an established property valuation technique that has previously been employed to examine property price trends following floods (Bin and Landry, 2013; de Koning et al., 2018). Kriging is a geostatistical technique that is commonly used to improve the accuracy of property value models by incorporating the effects of spatial autocorrelation on home prices, which can arise as a result of locational attributes (e.g., proximity to parks) that increase or decrease the price of a property relative to what would be expected given its basic characteristics (e.g., number of bedrooms) (Kuntz and Helbich, 2014).

For each fiscal quarter between 1990 and 2019, the market value ($P_{i,t}$) of each property within the study area is estimated via the following regression:

$$\log P_{i,t} = \log \hat{P}(x_{i,t}) + \hat{Z}(s_i, t) + \epsilon_{i,t} \tag{1}$$

where $x_{i,t}$ is a vector of available property-specific characteristics, $s_i$ is a vector of spatial coordinates describing the property location, and $t$ refers to the valuation date. Our hedonic pricing model assumes property values can be decomposed into three components: a deterministic component $\hat{P}(x_{i,t})$ reflecting basic property characteristics; a spatiotemporal component $\hat{Z}(s_i, t)$ reflecting location-specific amenities; and a zero-mean stochastic residual $\epsilon_{i,t}$.

The deterministic component $\hat{P}(x_{i,t})$ is estimated via random forest regression of observed sale prices on selected property characteristics available from the NCEM statewide building inventory and public sources of data such as the U.S. Census Bureau (Table S1). These predictors include property-specific attributes such as parcel size, heated square footage, and year built; census-tract level characteristics such as median income and mortgage loan amounts; and county-level housing market trends as measured by the FHA's annual home price index (Bogin et al., 2019). Using the trained random forest model and selected property characteristics, a hedonic property value ($\hat{P}(x_{i,t})$) is estimated for each property. Among properties with sale transactions, the difference between the estimated hedonic price and the observed market sale price ($P_{i,t}$) yields a "hedonic residual" ($Z_{i,t}$) such that:

$$Z_{i,t} = \log P_{i,t} - \log \hat{P}(x_{i,t}) \tag{2}$$

Because property sale prices reflect unobserved locational attributes and local market trends, hedonic residuals exhibit strong spatial and temporal autocorrelation. With this in mind, the spatiotemporal component of property value $\hat{Z}(s_i, t)$ is estimated via space-time interpolation of hedonic residuals using the simple lognormal kriging method (Chilès and Delfiner, 2012). Additional details regarding this procedure are available in Text S2 of the supplementary information.

To allow for regional variation in model parameters and to improve the computational efficiency of our method, property value estimation was carried out independently across 75 "kriging neighborhoods" created via k-means clustering of property sales, with each cluster containing an average of 30,000 sale transactions. To assess the performance of our property value estimation approach, predicted property values were compared against observed sale prices in 10-fold cross validation. In each fold, a separate model was fitted using 90% of the property sales data, with 10% withheld for validation.

The property values predicted by our model in cross-validation was within ±20% of the actual sale price for 54% of predictions, and within ±50% for 79% of predictions; when the 10% lowest-priced sale transactions in each year are excluded, these percentages increase to 59% and 86% respectively (Fig. S6). The scale of model errors varied over time as a result of property value appreciation and housing market trends; the distribution of absolute prediction error and median home prices by year are shown in Figure S7. Model performance varied somewhat across the study area (Fig. S8), with the lowest errors observed in urbanized counties having a high density of sale transactions. The substantial uncertainty in our property value estimates likely arises from a combination of factors, including: (1) the limited number of property-specific details in NCEM's statewide building inventory, which describes basic structural attributes but lacks information on other price-relevant characteristics such as recent improvements or deferred maintenance; (2) the presence of sales that do not reflect fair market values (e.g., intrafamily transfers) in the training and validation data, which can bias model predictions; and (3) geolocation errors that may result in mismatches between recorded sales and parcel geometries. Future work could potentially enhance the performance of the property valuation model by introducing filters to identify arms-length sales and by adding predictors that capture property-specific attributes related to structural defense and prior flood exposure (Nolte et al., 2024; Pollack and Kaufmann, 2022).

## 3.5 Model III: Mortgage repayment

The unpaid balance on mortgages within the study area was simulated on a monthly basis starting from the time of loan origination. Mortgage origination activity in each year was characterized using Home Mortgage Disclosure Act (HMDA) Loan Application Register data (CFPB, 2017; FFIEC, 2023; Forrester, 2021). This loan-level dataset contains 7.2 million mortgages originated in North Carolina from 1992 to 2019, and includes information on the loan amount, loan purpose, property type, census tract, and borrower income at the time of origination. After restricting our sample to loans from single-family, primary-residence homes located within the study area, our final dataset consisted of 4.7 million mortgage loans.

The HMDA data does not contain information on the interest rate, original loan-to-value (LTV) ratio, and original debt-to-income (DTI) ratio of each mortgage loan; thus, these variables were stochastically generated based on their observed distributions in the Fannie Mae and Freddie Mac (hereafter referred to as the GSEs) single-family loan datasets (Fannie Mae, 2023; Freddie Mac, 2023). These datasets include detailed loan-level origination and monthly performance data for all single-family mortgages in North Carolina that were acquired by the GSEs between 1999 and 2021. We restricted our sample to mortgages with 30-year and 15-year terms, which accounted for 98% of home purchase and 86% of refinance loans in the GSE

dataset. To adjust for temporal trends in mortgage rates, we converted the interest rate of each loan included in the GSE dataset into an interest rate spread by subtracting the average 30- or 15-year fixed mortgage rate at the time of origination (hereafter referred to as the "benchmark rate") from the loan-specific rate (Freddie Mac, 2016b, a).

We used the copula method to separately model the correlation structure and marginal distributions of the following five mortgage origination variables: borrower income, loan amount, LTV ratio, DTI ratio, and spread over the benchmark rate. The marginal distribution of each variable was nonparametrically modeled using empirical distribution functions estimated from GSE origination data; the correlation between variables was modeled using a Gaussian copula fit to GSE data using the maximum pseudo-likelihood (MPL) method (Genest et al., 1995). The resulting multivariate distributions were stratified by the year of origination, loan purpose (home purchase or refinance), and loan term (30 or 15 years). These distributions were then used to simulate the values of mortgage origination variables not included in the HMDA dataset (i.e., LTV ratio, DTI ratio, spread over the benchmark rate) conditional on borrower income and loan amount. Because the GSE dataset does not include loans originated prior to 1999, the joint distribution of origination variables for pre-1999 mortgages was modeled by combining the year-specific marginal distributions of borrower income and loan amount observed in the HMDA dataset; the marginal distributions of LTV, DTI, and rate spread among GSE mortgages originated in 1999; and the fitted Gaussian copula corresponding to the 1999 period.

Because HMDA mortgage origination data is anonymized to the census tract level, each mortgage loan is randomly assigned to a specific property within the listed census tract at origination. The likelihood of a given property being matched to a loan is determined based on its estimated value at the time of origination (model II, Sect. 3.4) and the probability density function (PDF) of potential property values implied by the mortgage loan amount and LTV ratio distribution. Once a mortgage loan is assigned to a specific property, no new mortgages can be assigned to that same property until the previous mortgage has been terminated.

Mortgage repayment is simulated on a monthly basis until the loan is either paid off in full or the end of the simulation time horizon (December 2019) is reached. The borrower's monthly mortgage payment ($c$) is calculated as a function of the original loan balance ($B_{t_0}$), monthly interest rate ($r$), and loan term in months ($N$) assuming a constant repayment schedule:

$$c = \frac{r}{1 - (1 + r)^{-N}} B_{t_0} \tag{3}$$

The unpaid balance is updated at the end of each month to reflect interest and payments:

$$B_{t+1} = B_t(1 + r) - c \tag{4}$$

For simplicity, our model assumes that all home purchase loans have a 30-year term; among single-family home purchase loans acquired by the GSEs in North Carolina, those with repayment periods of less than 30 years accounted for only 11% of the total (Fannie Mae, 2023; Freddie Mac, 2023). For refinance loans, two-thirds are randomly assigned a 30-year term while the remainder are assigned a 15-year term, which reflects the approximate ratio of 30-year to 15-year terms among refinance loans acquired by the GSEs.

Most mortgage loans in the U.S. are repaid well before the maturity date due to borrowers refinancing or selling their property. In an environment of falling interest rates, borrowers have a strong incentive to refinance their mortgage to obtain a lower rate; at a given point in time, this incentive is captured by the spread between their loan's interest rate and the prevailing "market" rate (i.e., the average 30- or 15-year fixed rate on new mortgages). With this in mind, we model the time-dependent prepayment rate as a function of both the loan age and interest rate spread using a Cox proportional hazards model (Cox, 1972):

$$\lambda(t) = \lambda_0(t) \exp\left(\beta(r - r_{m,t})\right) \tag{5}$$

where $\lambda(t)$ is the hazard (prepayment) rate $t$ months after origination for a loan with interest rate $r$, $\lambda_0(t)$ is the "baseline" hazard function, $r_{m,t}$ is the prevailing market rate, and $\beta$ is a coefficient controlling the degree to which a positive rate spread increases prepayment rates. Cox model coefficients and baseline hazard functions were estimated using 115 million loan-month observations from North Carolina mortgages included the GSE single-family loan performance datasets (Table S5).

These estimates were stratified by the loan purpose (home purchase or refinance) and, in the case of refinanced mortgages, the loan term (30 or 15 years). The fitted Cox models are used within our simulation to calculate the monthly probability of a borrower repaying their mortgage early; if this occurs, the balance on their loan is set to zero. Our model only simulates loan terminations resulting from voluntary payoffs (i.e., prepayments and maturity payments) and does not track terminations from defaults or foreclosures. The omission of default-related terminations is unlikely to materially affect the loan age distribution,

as the "background" rate of default was low relative to the rate of voluntary payoffs. Among GSE-backed single-family mortgages in North Carolina that were active at any point from 2000 to 2019, only 3.3% of loans were ever more than 120 days delinquent (a prerequisite for initiating foreclosure proceedings) and over 97% of loan terminations during this period resulted from voluntary payoffs (Fannie Mae, 2023; Freddie Mac, 2023). The simulated repayment profiles produced by our model closely align with those empirically observed in the GSE data (Fig. S9).

It is important to note that mortgages acquired by the GSEs—which account for approximately half of all U.S. mortgage originations (GAO, 2019)—consist of "conforming" loans that meet standardized requirements related to loan size, borrower credit quality, and documentation. Mortgages that are not represented in the GSE data include "jumbo" loans whose amounts exceed the conforming loan limit, which are typically associated with very expensive properties; "subprime" loans made to borrowers with questionable credit history or unverifiable income, which peaked at 15% of the U.S. mortgage market

in the years leading up to the 2007 subprime mortgage crisis (Agarwal and Ho, 2007); and loans insured by government programs targeting specific groups such as first-time homebuyers, veterans, and active-duty military personnel (Jones, 2022; Perl, 2018). As such, borrower attributes that were simulated based on GSE data primarily reflect the characteristics of middle-income, creditworthy borrowers, and may underrepresent the characteristics of households at both the upper and lower ends of the wealth distribution and of communities in North Carolina with a large military presence such as Cumberland, Onslow,

and Craven counties (N.C. Department of Military and Veterans Affairs, 2025).

## 3.6 Model IV: Borrower financial conditions

The financial conditions of mortgage borrowers are simulated on a monthly basis while accounting for the effects of flood damage exposure, insurance status, income growth, and property value dynamics on borrower equity and liquidity. Our approach integrates the outputs of the three sub-models (Fig. 1) — flood-related damages (model I, Sect. 3.3), property values (model II, Sect. 3.4), and unpaid mortgage balances (model III, Sect. 3.5) — to provide a comprehensive picture of a borrower's capacity to finance home repairs in the aftermath of a flood while continuing to meet their existing debt obligations.

At simulation onset, the monthly debt obligations of each borrower ($c_D$) are determined based on their debt-to-income ratio (DTI) and monthly income ($I$) at the time of origination ($t_0$):

$$c_{D,t_0} = DTI_{t_0} \cdot I_{t_0} \tag{6}$$

The monthly non-mortgage debt obligations of each borrower ($c_{NM}$) are calculated by subtracting their mortgage payment ($c_M$) from the total monthly liability implied by Eq. (6):

$$c_{NM} = c_{D,t_0} - c_M \tag{7}$$

This value represents the sum of recurring monthly obligations from sources of debt that are not explicitly modeled, but nevertheless affect a borrower's DTI ratio (e.g., student loans, revolving credit) (Fannie Mae, 2024b). These non-mortgage obligations are assumed to remain constant throughout time. If a borrower obtains a loan to fund flood-related repairs, their total monthly debt obligation is updated to reflect this additional liability:

$$c_{D,t} = c_{NM} + c_M + \sum_{i=1}^{N_t} c_{R,i} \tag{8}$$

where $N_t$ represents the number of separate home repair loans that are being repaid at a given point in time, and $c_{R,i}$ represents the monthly payment associated with each loan. The third term in Eq. (8) only applies to those who are still paying off home repair loans obtained following exposure to uninsured flood damage in an earlier simulation timestep. Borrower income is updated on an annual basis to reflect county-level trends in personal income growth:

$$I_{t+1} = I_t(1 + g_t) \tag{9}$$

where $g_t$ represents the average rate of growth in per-capita income for a specific county and period (BEA, 2023). At each timestep, DTI ratios are updated to reflect income growth and changes in total monthly debt obligations:

$$DTI_t = \frac{c_{D,t}}{I_t} \tag{10}$$

The time-varying DTI ratio from Eq. (10) is an important measure of a borrower's monthly cashflow that reflects their capacity to support additional debt payments. Lenders typically impose limits on DTI that can prevent those with a high ratio from obtaining a loan, with most conventional mortgages requiring a DTI ratio of 45% or lower (Fannie Mae, 2024b).

The ability of property owners to finance home repairs through debt is also affected by their loan-to-value (LTV) and combined loan-to-value (CLTV) ratio. At each simulation timestep, a borrower's LTV ratio is calculated based on the outstanding balance on their mortgage and the current value of their property:

$$LTV_t = \frac{B_{M,t}}{P_t} \tag{11}$$

where $P_t$ is the property value estimated by the hedonic home price model (model II, Sect. 3.4), and $B_{M,t}$ is the current unpaid mortgage balance (model III, Sect. 3.5). The LTV ratio in Eq. (11) only includes the primary mortgage and does not consider other debts secured by the property. In contrast, a borrower's CLTV ratio includes the outstanding balance on home repair loans obtained over the simulation time horizon:

$$CLTV_t = \frac{B_{M,t} + \sum_{i=1}^{N_t} B_{R,i,t}}{P_t} \tag{12}$$

where $N_t$ represents the number of separate home repair loans that are being repaid at a given point in time, and $B_{R,i,t}$ represents the current balance of each loan. The number of home repair loans associated with each borrower is updated over time as the loans are paid off or as new ones are acquired following successive exposures to uninsured property damage. The CLTV ratio is a dynamic measure of mortgage borrower's equity that reflects their capacity to borrow against the value of their property. In most cases, mortgage lenders are unwilling to approve a loan that would increase a borrower's CLTV ratio beyond 97% (Fannie Mae, 2024a).

If a mortgage borrower experiences flooding, adjusted debt-to-income (ADTI) and combined loan-to-value (ACLTV) ratios are calculated by assuming the borrower will attempt to pay for uninsured damage by applying for a home repair loan using their property as collateral:

$$ADTI_t = DTI_t + \frac{c_F}{I_t} \tag{13}$$

$$ACLTV_t = CLTV_t + \frac{B_F}{P_t} \tag{14}$$

where $B_F$ is the loan amount required to fully pay for uninsured damages, and $c_F$ is the monthly payment associated with a loan of this size having a 30-year term and interest rate equal to the prevailing average 30-year mortgage rate (Freddie Mac, 2016b). Borrowers are evaluated for loan approval or denial based on their damage-adjusted debt-to-income and adjusted loan-to-value ratios: those with an ADTI ratio of ≤45% and an ACLTV ratio of ≤100% are assumed to receive the loan, while those who fail to meet these criteria are assumed to be ineligible for a private loan. These thresholds reflect the underwriting criteria employed by the FHA's Section 203(h) program, which insures mortgages made by lenders to disaster-affected property owners (HUD, 2024). Unlike most other sources of home equity loans, which typically impose stricter CLTV limits, the 203(h) program permits property owners to borrow up to 100% of their equity with no down payment so long as their total monthly debt obligation does not exceed 45% of their gross monthly income (McCarty et al., 2006). It should be noted that borrowers meeting these ratio-based criteria can still be denied a loan due to unsatisfactory credit history—a process that is not represented in our modeling framework. While existing mortgage borrowers have (by definition) previously met lending standards and likely possess higher credit scores than the general population, the omission of factors related to credit history may cause us to underestimate the share of flood-exposed borrowers who would be denied a loan.

Borrowers with uninsured flood damage who are approved for a loan are assumed to fully repair the damage to their home while continuing to meet their existing debt obligations, while those who are prevented from obtaining a loan due to their ADTI or ACLTV ratio are removed from subsequent simulation timesteps. The recovery outcomes of those who are ineligible for private home repair loans are uncertain and highly dependent on the availability of alternative funding sources, including: personal savings, home disaster loans provided by the SBA, and housing assistance grants provided by FEMA's Individuals and Households Program (IHP). SBA loans have maturities of up to 30 years and offer below-market interest rates to borrowers meeting program credit score and debt-to-income ratio requirements (Ellis and Collier, 2019; Lindsay and Getter, 2023; Lindsay and Webster, 2022). IHP housing assistance grants can provide property owners with funding for repairs to their primary residence up to a fixed amount ($42,500 as of 2024) updated annually for inflation (U.S. GPO, 2023; Webster, 2024). Although we do not explicitly model these sources of federal disaster relief, in sensitivity analysis, we vary home repair interest rates and loan amounts to assess the potential impact of SBA loans and IHP grants on borrower ADTI and ACLTV ratios (Sect. 4.2).

Borrowers who are unable to obtain a home repair loan are considered to be "credit constrained" and are further categorized based on whether these constraints are driven by insufficient collateral, insufficient income, or both in combination. An ACLTV ratio of >100% denotes the presence of negative equity and indicates that a borrower cannot use their residence as collateral to obtain additional loans; these individuals are thus considered to be "collateral constrained." An ADTI ratio of >45% implies liquidity problems and indicates that a borrower does lacks the available income necessary to take on an additional monthly loan payment; these individuals are thus considered to be "income constrained." If both criteria are met, this indicates that a borrower is prevented from obtaining a home repair loan by both income and collateral constraints. It is important to note the ACLTV and ADTI thresholds employed in this framework are assumed to be necessary (but not sufficient) conditions for financial distress; as such, the credit constraint estimates generated by our procedure reflect the share of flood-exposed borrowers who may be forced to rely on other (less reliable) sources of funding for recovery such as savings, post-disaster aid, and support from family and friends. Additional information linking the post-flood financial conditions of mortgage borrowers to the probability of bankruptcy and default could be used to translate the estimates generated by our approach into projections of lender credit losses (Bellini, 2019).

Because we lack data on mortgages originated prior to 1992, our method is likely to underestimate the number of mortgages that were active during the earliest two flood events that occurred during the study period. For this reason, the years 1992-1998 are treated as a "warm-up" period for the simulation and Hurricanes Fran (1996) and Bonnie (1998) are excluded from estimates of flood-related credit constraints. For each simulation run, we simulate the financial conditions of 4.7 million borrowers with single-family mortgages originated during the 1992-2019 period at a monthly timestep over the life of their loan. Because certain variables describing the initial financial conditions and repayment profiles (model III) of mortgage borrowers are stochastically generated, model projections of flood-related credit constraints were averaged over ten simulation runs conducted with different random seeds. This number of replicates was found to be sufficient for achieving stable estimates

585    of the number of borrowers facing flood-related credit constraints across the study area; however, generating stable estimates for smaller geographic units (e.g., specific census tracts) would likely require additional simulation runs.

        Our approach to modeling household financial conditions focuses on how uninsured property damage affects the borrowing capacity of flood-exposed property owners through its influence on CLTV and DTI ratios. It does not, however, capture the full range of factors and processes that may play a role in shaping household financial outcomes following flood

590    events. These include household saving behaviors, which may be heterogenous by wealth and insurance status; the timing of insurance claim payouts (which are assumed to immediately offset the cost of flood damage for insured borrowers); exogenous shocks to income arising from changes in employment status and negative life events; and the ability of households to supplement or replace home equity-based borrowing with other sources of funding for recovery, as described in Section 2. A conceptual overview of common household budget components that were included and excluded from our model is provided

595    in Table S6.

## 4 Results

Results include analyses of seven flood events across the study period, with the financial impacts (e.g., loan repayment) from one event sometimes extending through the occurrence of the next. Model projections of flood damage exposure (Sect. 4.1) and financial risks to mortgage borrowers stemming from flood-related credit constraints (Sect. 4.2) are aggregated across a number of groups defined based on geographic and economic factors that may be relevant to flood resilience policy. Unless otherwise stated, monetary amounts are adjusted for inflation based on the U.S. Consumer Price Index and displayed in 2020 United States dollars (OECD, 2023).

### 4.1 Estimates of flood damage exposure

A total of 67,200 properties were projected to have flooded at least once over the study period, resulting in $4.0 billion in aggregate damage (Fig. 3, Table S7). Properties flooded two or more times accounted for 19% of all inundated structures and generated $694 million in repetitive damages, which we define as any damage to a property occurring after its first exposure to flooding during the study period. Only 34% ($1.4 billion) of all projected damages were covered by flood insurance, with a total of 43,300 properties exposed to $2.6 billion in uninsured flood damage over the study period. Among those exposed to uninsured damage, the median (IQR) cost of property damage was $45,100 ($38,200-$58,200)—an amount equal to over 70% of the 2020 median household income in North Carolina (U.S. Census Bureau, 2020).

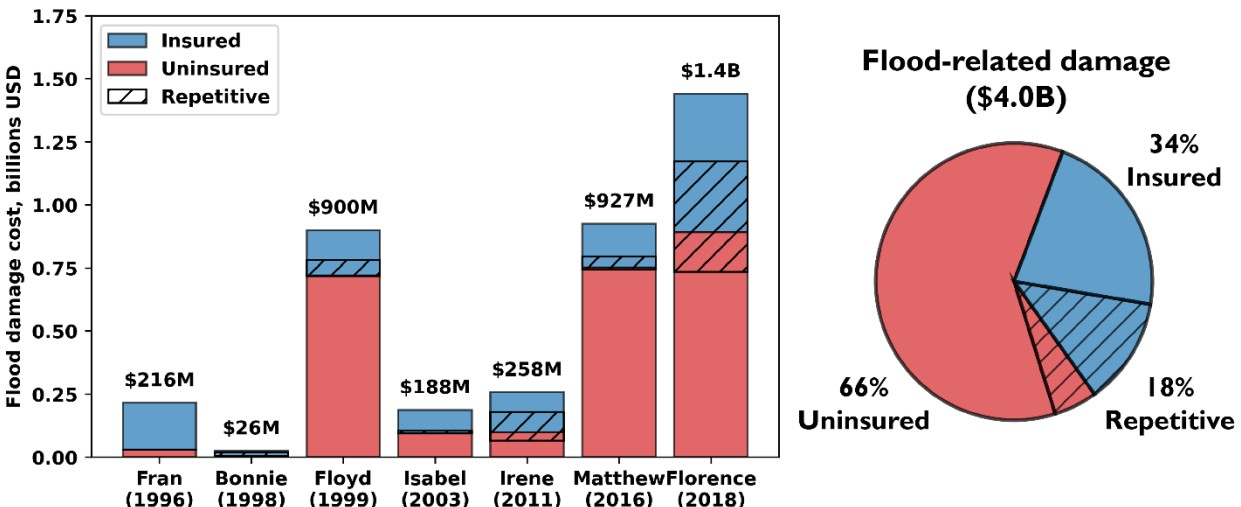

**Figure 3.** Flood damage to structures within the study area by event. Dollar amounts are adjusted for inflation and expressed in 2020 USD.

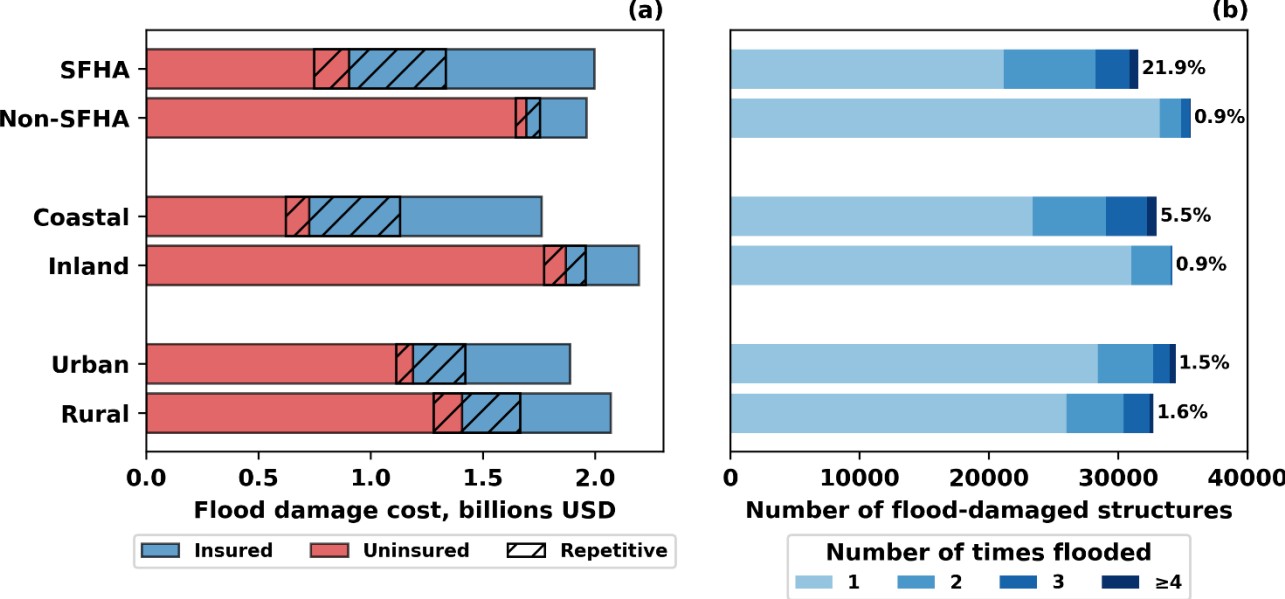

**Figure 4.** Flood damage to structures within the study area by comparative groups. Bars should only be compared within appropriate pairs (e.g., SFHA vs. non-SFHA) but not across pairs (e.g., SFHA vs. Coastal) as groups across pairs are not mutually exclusive. In panel (b), percentages denote the proportion of structures within the study area that flooded at least once during the seven evaluated events stratified by group.

The most severe events in terms of property damage were Hurricanes Florence ($1.4 billion), Matthew ($927 million), and Floyd ($900 million). Approximately 36% of properties damaged by flooding during Hurricane Florence also experienced damage during one of the other evaluated events, with 12% flooded two years prior during Hurricane Matthew. A high degree of overlap was observed between properties damaged by Hurricanes Isabel and those damaged by Hurricane Irene (20% overlap), as well as Hurricanes Fran and Bonnie (17% overlap). In general, events that mainly impacted coastal areas (which we define as counties under the jurisdiction of the Coastal Area Management Act) and SFHAs exhibited a higher degree of repetitive damage than events whose impacts extended to inland regions and areas outside of the SFHA (Figs. S10-S16). Across the study period, the share of damages attributable to repetitive flooding was 29% in coastal areas versus 8% in inland areas and 29% inside the SFHA versus 5% outside the SFHA (Fig. 4a). Among repeatedly flooded properties, the average number of times inundated was 2.4 over the 24-year study period. Because our analysis only considers the seven largest flood events that occurred during the study period, these estimates of repetitive damage are likely to be conservative.

The share of uninsured damages varied substantially by event, from a low of 14% during Hurricane Fran (whose flood impacts were mainly limited to areas near the coast and inside SFHAs) to a high of 81% during Hurricane Matthew (which caused widespread flooding in areas further inland and outside SFHAs) (Fig. 3, Table S7). In general, the uninsured fraction of damage increased with event size and accounted for over 70% of all flood-related damages during the three costliest

events (Hurricanes Florence, Matthew, and Floyd). The share of uninsured damages was highest for properties located outside of the SFHA (86%) and in inland areas (85%). Even inside the SFHA, where uptake of flood insurance is relatively higher, nearly half of all property damage was uninsured (Fig. 4, Table S7). Rural areas were exposed to over $1.4 billion in uninsured flood damage over the study period, compared to $1.2 billion for areas classified by the Census Bureau as urban (which account for a higher share of the state's population and housing units) (Fig. 4, Table S7) (U.S. Census Bureau, 2024). This was likely driven by the large amount of damage concentrated in North Carolina's Coastal Plain region, which accounts for approximately 45% of the state's land area but contains relatively few large metropolitan areas (NC Parks, 2024).

Properties at the extreme ends of the property value distribution exhibited the highest levels of flood damage exposure over the study period. During Hurricanes Floyd and Isabel, a plurality (31%) of flood-damaged homes were in the bottom 20% of the statewide property value distribution; during Hurricanes Florence and Matthew, these homes accounted for over half of all properties damaged by flooding (Fig. S17). In contrast, homes in the top 20% made up the greatest share of flood-damaged homes during Hurricanes Fran, Bonnie, and Irene (Fig. S17). This is largely due to the concentration of high-valued real estate along the North Carolina coastline and Outer Banks, which accounted for the bulk of inundated properties during events driven primarily by coastal flooding (Hurricanes Fran, Bonnie, Irene, and Isabel). In contrast, events producing large amounts of pluvial and fluvial flooding in inland areas such as Hurricanes Matthew and Florence damaged many homes in rural areas of the Coastal Plain, where property values tend to be lower (Anton and Cusick, 2018). Lower-valued homes exposed to flooding experienced much higher levels of relative damage (damage cost per dollar of property value) than their more expensive counterparts. For example, the median relative damage to properties flooded during Hurricane Florence was 70% for those in the bottom two property value quintiles versus just 5% for those in the top two property value quintiles (Fig. S18). During Hurricanes Floyd, Matthew, and Florence, over 37% of flooded homes in the bottom property value quintile experienced damage exceeding 90% of their pre-flood property value; in contrast, less than 1% of flooded homes in the top property value quintile experienced this outcome. This has important implications for neighborhood-level recovery outcomes: areas with a large number of "total loss" properties are likely to see elevated rates of property vacancy and abandonment since both property owners and lenders have less to gain financially by quickly repairing these homes (Zhang, 2012). High rates of property vacancy can lead to negative spillover effects that reduce the value of nearby homes and also impose costs on local governments in the form of lost tax revenue and expenses related to maintenance, demolition, and crime prevention (GAO, 2011; Gerardi et al., 2015; Lin et al., 2009).

Spatial differences in the intensity and type of flood damage exposure are illustrated by aggregating estimates of insured, uninsured, and repetitive damages on a uniform 15 km hexagonal grid (Fig. 5). For display purposes, plots of damage intensity only include counties that are members of the nine easternmost regional councils of North Carolina (NCARCOG, 2024), which collectively accounted for over 99% of estimated damages during the seven evaluated flood events. Damages are normalized by the number of properties within each grid cell and averaged over the 24-year study period to produce estimates of average annual damage (AAD). The highest concentrations of uninsured damage were observed in Pamlico, Edgecombe, Nash, and Pender counties, which all experienced uninsured AADs exceeding $200 per property. Two spatial

clusters with high levels of uninsured and repetitive damages were identified: the first encompasses an area spanning Pamlico, Craven and Carteret counties, which were collectively exposed to $744 million in flood damage over the study period, of which 45% was uninsured; and the second cluster spans Robeson, Bladen and Columbus counties, which were exposed to $521 million in total damage with an uninsured fraction of 86%.

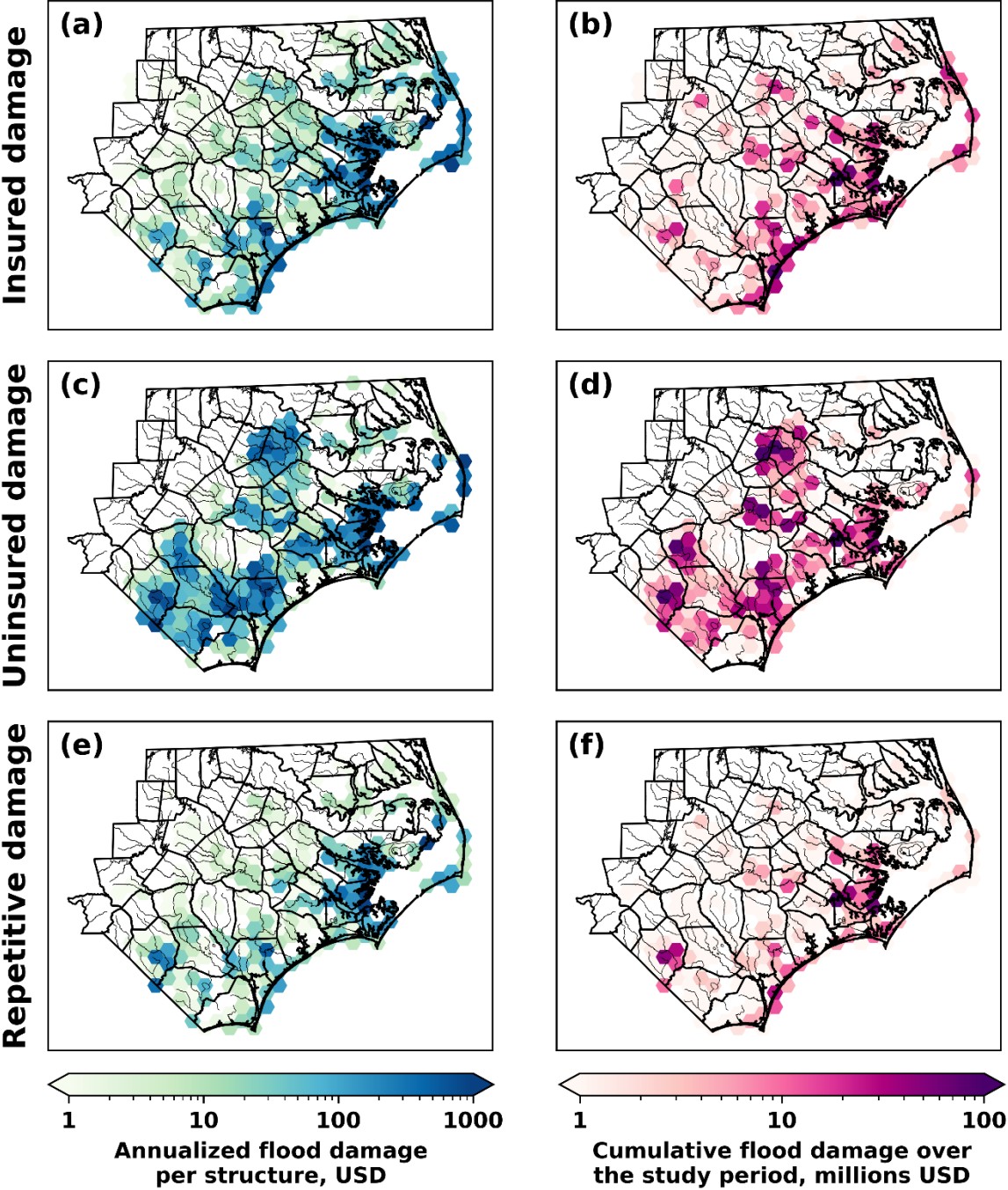

**Figure 5.** Spatial distribution of insured, uninsured, and repetitive flood damage. Estimates of damages occurring over the 24-year study period are aggregated on a uniform 15 km hexagonal grid. For display purposes, only counties that are members of the nine easternmost regional councils in North Carolina are shown.

## 4.2 Financial risks to mortgage borrowers

Among 4.7 million single-family mortgages originated in the study area from 1992 to 2019, approximately 22,100 (0.47%) are estimated to have experienced flood damage at least once over the life of the loan from one or more of the seven evaluated events. Among borrowers exposed to flood damage, 11,100 (50%) were located outside of the SFHA and 11,500 (52%) lacked flood insurance at the time of their exposure, with non-SFHA borrowers accounting for 73% of those exposed to uninsured damage. The median (IQR) loan amount required to fully cover the uninsured cost of flood damage repairs was $46,000 ($39,200-$57,200); in relative terms, the cost of these repairs represented a median (IQR) of 32% (20%-47%) of a borrower's pre-flood property value. Borrowers with uninsured flood damage had ACLTV and ADTI ratios that were a median of 32 and 4 percentage points higher (respectively) than their pre-flood CLTV and DTI ratios; these increases were most pronounced for lower-income borrowers and those with lower-valued properties (Fig. 6).

Over the study period, 7,180 mortgage borrowers were projected to face flood-related credit constraints as indicated by ACLTV > 100% or ADTI > 45%. This number represents a small share (0.15%) of all mortgages originated during the study period but a substantial fraction (32%) of those exposed to flooding. Given the relatively low sensitivity of our flood damage model observed in cross-validation (Section 3.3), our projections may underestimate the true number of borrowers exposed to flooding over the study period and prevalence of flood-related credit constraints. Among borrowers exposed to flooding, 28% were projected to experience negative equity (ACLTV > 100%) after accounting for flood-related property damage; for comparison, 23% of U.S. mortgage borrowers had negative equity during the peak of the global financial crisis (James, 2009). Among credit constrained borrowers, the median (IQR) shortfall in funding for home repairs was $29,600 ($16,700-$44,600) (Fig. 7). This quantity represents the difference between a property owner's borrowing capacity (i.e., the maximum amount of additional debt they can take on without exceeding CLTV and DTI limits) and the total cost of uninsured property damage. Of those facing flood-related credit constraints, 89% were collateral constrained (ACLTV > 100%), 35% were income constrained (ADTI > 45%), and 24% were constrained by both (ACLTV > 100% and ADTI > 45%). Because these categories are not mutually exclusive, we hereafter use the terms "collateral constrained only" and "income constrained only" to distinguish those who faced collateral or income constraints but not both simultaneously. Those facing simultaneous collateral and income constraints exhibited high levels of financial stress (average shortfall of $50,200) compared to those who were constrained by collateral only or income only (average shortfall of $31,600 and $32,600 respectively).

The role of liquidity as a driver of credit constraints varied substantially across the income distribution. Among those experiencing uninsured damage whose income put them in the bottom 20% of mortgage borrowers, over half faced income constraints (ADTI > 45%) that would impair their access to home repair loans (Fig. 6c). These borrowers also exhibited high rates of negative equity (ACLTV > 100%), with those facing simultaneous income and collateral constraints accounting for a large share (47%) of all credit constrained borrowers in this income group (Fig. 8a). These findings indicate that the monthly cashflows of many lower-income borrowers are already stretched to the limit, and that these households would likely require

modification of their existing mortgage loan (e.g., reduced interest rate, extended repayment term) to support additional debt payments associated with home repairs. The importance of liquidity as a driver of credit constraints was diminished for high-income borrowers whose monthly cashflows had more capacity to absorb additional debt payments: among borrowers in the top income quintile, 87% of credit constrained borrowers had sufficient income to take on a home repair loan but were prevented from doing so by negative equity (i.e., insufficient collateral).

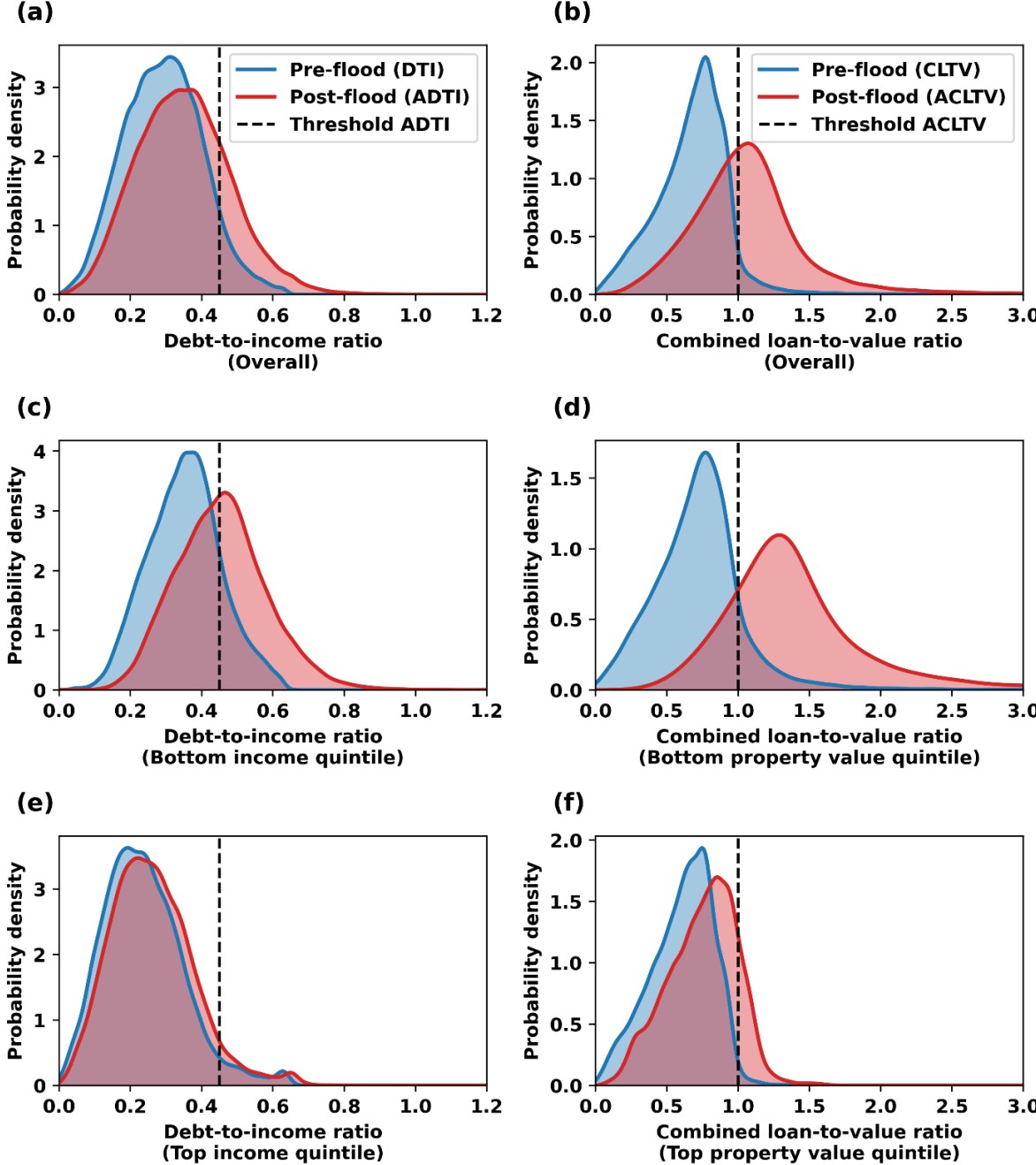

**Figure 6.** Damage-adjusted debt-to-income (DTI) and combined loan-to-value (CLTV) ratios among mortgage borrowers exposed to uninsured flood damage. The DTI ratio measures the share of a borrower's monthly income consumed by recurring debt obligations, while the CLTV ratio measures home equity as the total balance of all loans secured by a property divided by its market value. The post-flood adjusted DTI (ADTI) and adjusted CLTV (ACLTV) ratios capture the projected effects of financing flood-related repairs through home equity-based borrowing on borrowers' cashflow and equity positions. Dashed lines indicate the ADTI and ACLTV thresholds used to classify borrowers as credit constrained following exposure to uninsured damage.

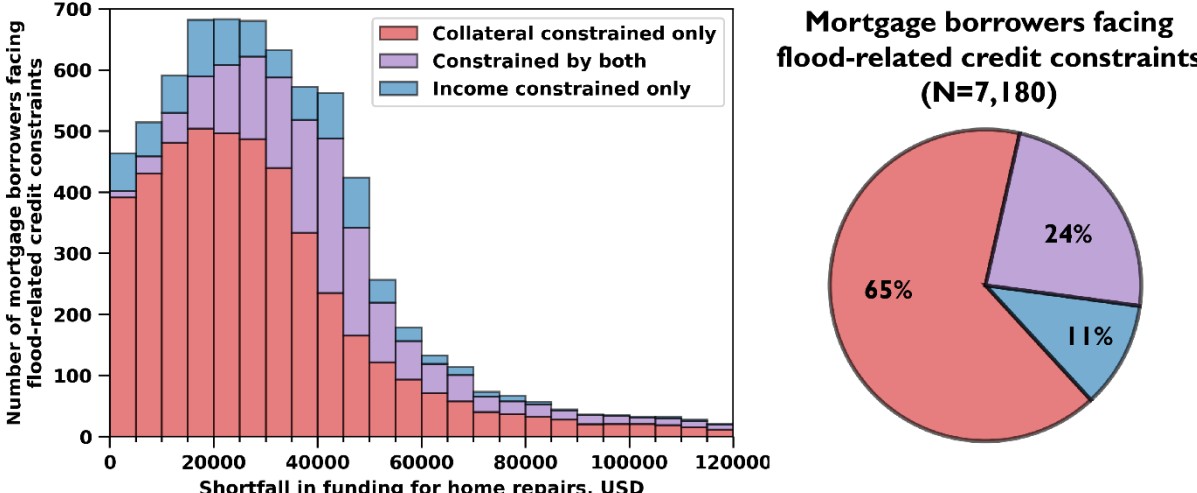

**Figure 7.** Distribution of shortfall in funding for home repairs among mortgage borrowers facing flood-related credit constraints. Shortfall is calculated by subtracting a household's borrowing capacity (i.e., the maximum amount of additional debt they can take on without exceeding a CLTV limit of 100% and DTI limit of 45%) from the total cost of uninsured property damage. Borrowers constrained by collateral only (ACLTV > 100%, ADTI ≤ 45%) are shown in red; those constrained by income only (ACLTV ≤ 100%, ADTI > 45%) in blue; and those constrained by both (ACLTV > 100%, ADTI > 45%) in purple.

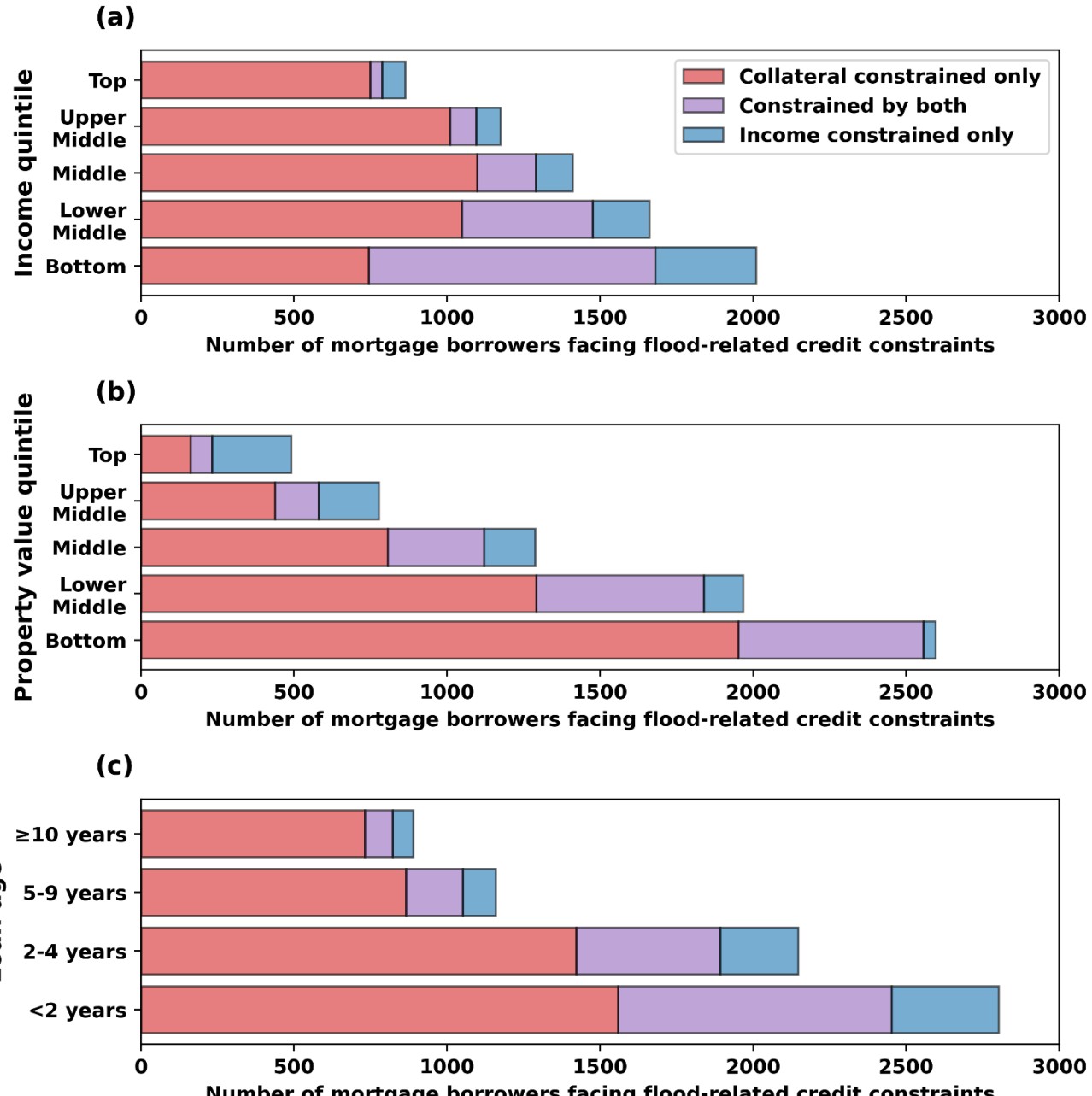

**Figure 8.** Characteristics of mortgage borrowers facing flood-related credit constraints. Horizontal bars represent the cumulative number of mortgage borrowers facing flood-related credit constraints over the study period stratified by (a) income quintile, (b) property value quintile, and (c) loan age. Borrowers constrained by collateral only (ACLTV > 100%, ADTI ≤ 45%) are shown in red; those constrained by income only (ACLTV ≤ 100%, ADTI > 45%) in blue; and those constrained by both (ACLTV > 100%, ADTI > 45%) in purple.

Borrowers in the lowest property value quintile disproportionately faced flood-related credit constraints, primarily due to negative equity. Because a given dollar amount of flood damage produces more relative damage at a lower-valued property, borrowers in the bottom property value quintile were more likely to have ACLTV exceeding 100% following a flood. The probability of uninsured damage exceeding the value of a borrower's pre-flood home equity was 82% for those in the bottom property value quintile (Fig. 6d) compared with 14% for those in the top property value quintile (Fig. 6f). Higher pre-flood property values reduced the potential for negative equity: those in the top 40% of the property value distribution accounted for only 17% of all collateral constrained borrowers (Fig. 8b). However, high value properties were still susceptible to income constraints, which occurred for 20% of borrowers in the top property value quintile that experienced uninsured flood damage. Given that most households derive the bulk of their net worth from the value of their primary residence (Jones and Neelakantan, 2023), these findings indicate that households with lower initial wealth face substantial barriers to affordable credit in the aftermath of a flood, which may exacerbate wealth inequality in affected areas (Howell and Elliott, 2019).

Those with recently originated loans under two years old accounted for over a third of all credit constrained mortgage borrowers (Fig. 8c). Compared to those with loans aged ≥10 years, borrowers with loans aged <2 years were almost twice as likely to experience either collateral or income constraints following an exposure to uninsured flood damage. The protective effect of loan age likely arises from the interaction of three dynamic processes: (1) reductions in the unpaid mortgage balance over time as the loan is repaid, (2) property value appreciation, and (3) income growth over time. Loan repayment and property value appreciation act in combination to increase a borrower's pre-flood equity (thus lowering their CLTV ratio), while income growth causes their existing mortgage payment to represent a smaller share of their total monthly cashflow (thus lowering their DTI ratio) which increases their capacity to support additional debt payments associated with home repairs.

Our projections of flood-related credit constraints by income, property value, and loan age (Fig. 8) should be interpreted in light of several modeling assumptions that may influence comparisons across groups. First, we did not account for the positive correlation between income and flood insurance uptake when assigning loans to specific properties within a census tract (Section 3.5). Incorporating this source of heterogeneity in flood insurance adoption would likely increase projected credit constraints for low-income borrowers and reduce them for high-income borrowers, particularly in areas outside the SFHA where insurance purchase is voluntary. Second, we assumed that borrower income evolves according to county-level trends in per-capita income growth (Section 3.6) and did not model changes in employment status. This assumption may overstate the protective effect of loan age, particularly for income-related credit constraints. Finally, our framework focuses on the ability of uninsured borrowers to finance repairs through home equity-based borrowing and does not capture how the ability to draw upon other sources of funding for recovery (such as savings and investments) may differ by income and property value. Because wealthier households tend to hold a greater share of their net worth in non-physical assets such as stocks (Jones and Neelakantan, 2023), the reliance on home equity-based borrowing for recovery is likely less pronounced among higher-income and higher-property-value households.

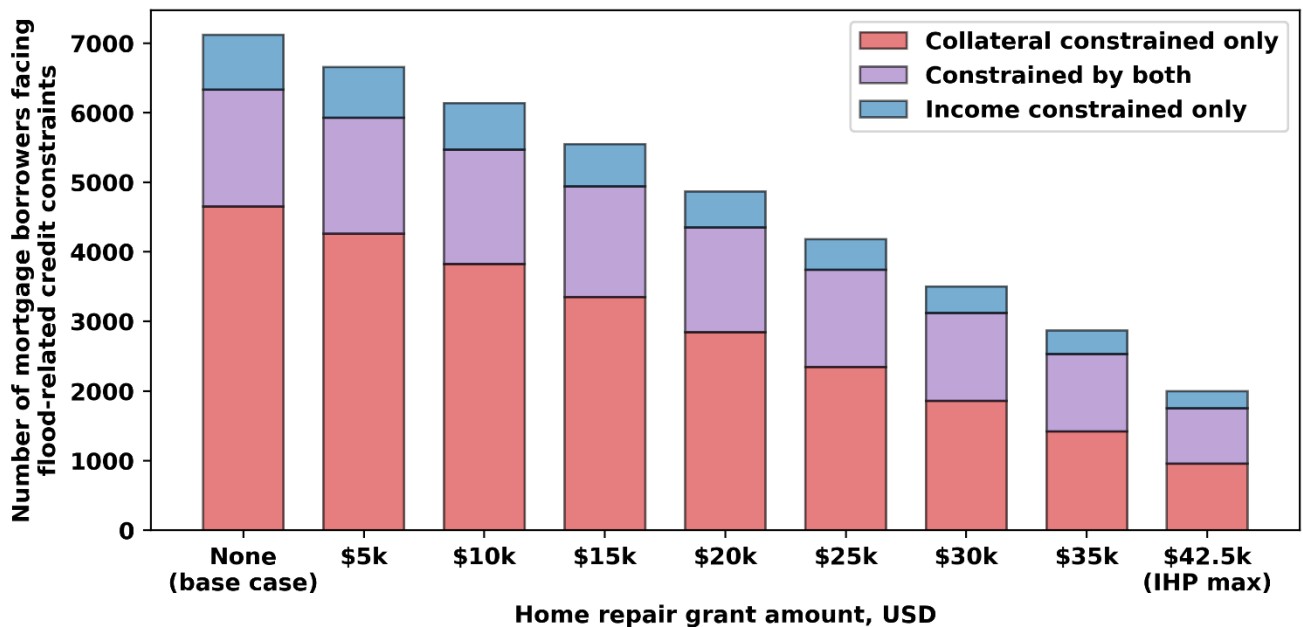

**Figure 9.** Scenario analysis on the amount of home repair grant assistance available to mortgage borrowers should they experience uninsured flood damage. The amount of available assistance is varied between zero and $42,500—the maximum award that households can receive from FEMA's Individuals and Households Program (IHP) as of 2024 (U.S. GPO, 2023).

In a scenario analysis examining how alternative assumptions regarding the cost of debt would impact these results, the interest rate at which uninsured borrowers can finance home repairs had only a modest effect on the total number of mortgage borrowers facing flood-related credit constraints (Fig. S19). This likely occurs because the vast majority of credit constraints arise as a result of negative equity (ACLTV > 100%), which depends on the loan amount required to finance repairs but not on the interest rate. For this reason, interest rate subsidies only affected the number of borrowers who were constrained by income alone—a relatively small percentage of the total. An interest rate subsidy equal to 50% of the average 30-year mortgage rate (which approximates the below-market rate on SBA disaster loans) reduced the number of borrowers with ADTI > 45% by 22%; however, this only translated into a 3% overall reduction in the number of borrowers facing flood-related credit constraints due to the frequent co-occurrence of income and collateral constraints.

In a scenario analysis examining the potential impact of a generic home repair grant program, the number of mortgage borrowers facing flood-related credit constraints was highly sensitive to the maximum amount of assistance available to those without flood insurance. Providing property owners with up to $15,000 in funding for repairs reduced the number of credit constrained borrowers by 22%, while further increasing this limit to $30,000 reduced the number of credit constrained borrowers by over 50%. An infusion of up to $42,500 in funding for repairs—equivalent to the maximum award that households can receive through FEMA's IHP program (U.S. GPO, 2023)—led to a 72% reduction in the number of credit

constrained borrowers. These findings suggest that home repair grants can improve the financial stability of flood-affected mortgage borrowers in a much more dramatic fashion than other forms of post-disaster aid such as low-interest loans. However, given that less than half of applicants to FEMA's IHP program are approved, with only a tiny fraction receiving the maximum award (GAO, 2020a), it appears likely that a substantial number of mortgage borrowers will remain credit constrained even after accounting for the allocation of post-disaster aid under current policies. In addition, the slow pace of the award determination process (which takes an average of 48 days from start to finish) can create short-term financial disruptions even for those who are ultimately approved for IHP aid (GAO, 2020a).

In a variance-based sensitivity analysis, we evaluated how uncertainty in estimates of flood damage (model I), property value (model II), and income (model IV) contributed to variation in borrower-level financial outcomes. Ranking uncertain parameters by their Sobol' total effect index (Saltelli et al., 2010; Sobol', 1993, 2001) for the outcome of a borrower being credit constrained due to either insufficient collateral or income (ACLTV > 100% or ADTI > 45%) revealed property value as the most influential parameter for 79% of flood-exposed borrowers (Table S8). Averaged over the simulated borrower population, the Sobol' total effect index of property value was approximately twice that of damage costs, the second-most influential parameter for this outcome (Fig. S21d). Uncertainty in estimates of damage costs had the greatest influence on the outcome of a borrower being simultaneously constrained by collateral and income (ACLTV > 100% and ADTI > 45%) (Table S8, Fig. S21c), likely because of the higher average level of flood damage required to trigger both constraints together than either constraint individually. Overall, these findings suggest that error associated with the estimation of property values and damage costs represent the largest sources of uncertainty in our projections of flood-related credit constraints. Additional details regarding our analysis of parameter uncertainty are available in Text S3 of the supplementary information.

## 5 Discussion

This analysis quantified the magnitude of uninsured property damage from seven flood events in North Carolina over the 24-year period from 1996-2019 at highly resolved spatial scales and evaluated the impact of these losses on the financial stability of residential mortgage borrowers. Our approach utilized a novel, data-driven framework combining property-level damage predictions with simulations of household income, debt, and property value dynamics to calculate the monthly value of financial metrics relevant to the post-flood credit constraints; these included damage-adjusted debt-to-income (ADTI) and combined loan-to-value ratios (ACLTV) for individual mortgages. This bottom-up approach provides insight into how the underlying drivers of credit constraints vary spatially and by borrower characteristics. In general, borrowers with lower amounts of home equity and higher debt-to-income ratios had diminished capacity to fund property repairs through low-cost sources of debt financing and were more likely to be credit constrained in the aftermath of a flood. Approximately one third of mortgages exposed to flooding were found to be credit constrained due to uninsured damage, with a large proportion of these risks stemming from recently originated loans and lower-valued properties.

Our results suggest that the number of North Carolina properties with past flood exposure is much larger than implied by NFIP records. During the seven flood events evaluated in this study, there were over 26,100 properties in the study area with at least one recorded NFIP claim and over 9,200 with two or more; however, this number does not reflect damage occurring to properties without insurance, which accounted for 66% of all estimated losses (Fig. 3). When uninsured damage is considered, the number of properties projected to have flooded at least once increases by a factor of 2.6 to 67,200, while the number flooded multiple times increases by a factor of 1.4 to 12,800. Among newly identified properties with past flood exposure, 12,000 (29%) were located inside the SFHA, 13,800 (34%) were located within a 250-meter distance of the SFHA, and 15,300 (37%) were located more than 250 meters from the SFHA boundary.

These findings, coupled with the pace of new construction in areas immediately adjacent to the SFHA, suggest that the number of uninsured properties exposed to flooding is likely to grow substantially over the coming decades. A recent study by Sanchez et al. (2024) estimates that 21% of new development in North Carolina between 2020 and 2060 is likely to be concentrated within 250 meters of current SFHA boundaries—an area that is typically exempt from flood-related building codes and insurance purchase requirements. As such, many of the flood events evaluated in this analysis would likely produce even greater amounts of uninsured damage were they to occur in the future simply due to the increased density of asset value in harm's way. Whether the bulk of future losses from flooding are internalized by affected households and communities within the state or transferred to the NFIP and private insurers depends strongly on future levels of flood insurance uptake. Given the high concentration of uninsured damage observed in inland areas of the Coastal Plain (Fig. 5), increasing the adoption of flood insurance in this region should be a priority for North Carolina policymakers, and likely those in other states as well. Many counties in the inner Coastal Plain exhibit high levels of economic distress relative to the rest of the state, and the cost of NFIP premiums is likely to be burdensome for many lower-income households in the region (County Distress Rankings, 2024). Thus, there is an urgent need for further research examining the cost-effectiveness of interventions to promote flood insurance uptake while simultaneously addressing affordability concerns, especially in light of recent premium increases under Risk Rating 2.0 (GAO, 2023).

This is particularly true as lower-valued properties were found to experience higher levels of flood damage relative to their market value and accounted for a disproportionate share of credit constrained mortgage borrowers. Consistent with the findings of Wing et al. (2020), we observed that the dollar amount of flood damage experienced by a property was not directly proportional to its market value; thus, the structural damage sustained by lower-valued properties represented, on average, a much greater share of their pre-flood property value than the damages experienced by higher-valued properties. As a result, mortgage borrowers in the bottom half of the property value distribution—those that had lower absolute amounts of home equity to begin with—lost a much larger share of their equity to uninsured damage and were more likely to face collateral constraints than those in the top half of the distribution. It is also worth noting that less wealthy households often derive a greater share of their net worth from the value of their primary residence than wealthier households that tend to have more diversified holdings, which may intensify the distributional impacts of equity losses due to uninsured damage (Jones and

Neelakantan, 2023). These findings highlight the mechanisms by which natural disasters such as floods reinforce and compound existing wealth gaps in the United States (Howell and Elliott, 2019).

860    Mortgage borrowers with minimal pre-flood home equity (e.g., those that have recently purchased their first home) were more likely to experience challenges in financing home repairs when confronted with uninsured damage given the reduced ability to use equity as a form of collateral. Traditional lenders such as banks and credit unions typically require home equity as collateral for loans, and a lack of equity can leave those with uninsured damage with few options for obtaining funds for repairs. Those with lower-valued properties or recently originated mortgages often experienced damage exceeding the value of their home equity, which severely constrains their borrowing capacity in the aftermath of a flood. The recovery

865    outcomes of those with negative equity are uncertain and depend strongly on the availability of federal sources of aid such as low-interest SBA disaster loans and FEMA IHP grants. The SBA's disaster lending program has flexible collateral requirements that in theory should not preclude those with negative home equity from obtaining a loan (Lindsay and Getter, 2023); however, in practice, many applicants are denied a loan on the basis of their credit history or debt-to-income ratio (Ellis and Collier, 2019; Lindsay and Webster, 2022). From 2016-2022, the SBA approved and denied a roughly equal number of

870    home disaster loan applications meeting minimum qualifying requirements, with the top reasons for denial being unsatisfactory credit and lack of repayment ability (GAO, 2024). This suggests that the importance of negative equity as a driver of post-flood financial distress is diminished for higher-income mortgage borrowers, who are likely to qualify for SBA loans due their higher average credit scores and lower post-flood debt-to-income ratios (Fig. 6e). However, lower-income borrowers, who often experience both negative equity and cashflow problems in the aftermath of a flood, may face difficulty in accessing SBA

875    loans due to their high post-flood DTI ratios (Fig. 6c, Fig. 8a). Additional data on how SBA loan approval rates vary by credit score, LTV, and DTI would allow for this source of post-disaster aid to be explicitly incorporated into the modeling framework.

    The results of this analysis should be interpreted in the context of several limitations. First, we used a machine learning model trained on insurance policies and claims data to estimate flood damage exposure within the study area, which creates the potential for selection bias due to differences between insured and uninsured households. For example, properties in high-

880    risk flood zones are overrepresented in our training data due to regulations requiring property owners with federally-backed mortgages to purchase flood insurance if their property is located inside the SFHA (GAO, 2021). In addition, higher-income households may also be overrepresented in our training data due to the positive association between wealth and flood insurance uptake (Atreya et al., 2015; Kousky, 2011). Although it is difficult to predict how these biases may influence our projections of flood damage, cross-validation results suggest that model performance was similar for insured properties inside and outside

885    the SFHA (Fig. S2). While insured properties located outside the SFHA are an imperfect proxy for uninsured households (for whom we lack data), this group provides insight into how our model is likely to perform in areas that were underrepresented in the training data.

    Second, we evaluated only the seven largest flood events (in terms of associated NFIP claims) between 1996 and 2019, and did not include the larger number of smaller, more localized events that occurred during the study period. As such,

890    our approach may underestimate past exposure to flood damage; an analysis by Garcia et al. (2025) examining a larger set of

78 events suggests that the number of buildings in the study area with past flood exposure could be as high as 90,000—a number 34% higher than our estimate of 67,000. Including these unmodeled events would likely increase the amount home repair debt carried by borrowers within the study area, leading to higher projections of flood-related credit constraints among those that also flooded during one of the seven evaluated events. In addition, cross-validation results suggest that our machine learning-based approach often failed to detect properties that were damaged, which is likely to contribute to a systematic underestimation of the true level of flood exposure within the study area. For these reasons, our projections of flood damage exposure and flood-related credit constraints should be interpreted as conservative bounds as opposed to central estimates.

Third, when modeling the financial conditions of residential mortgage borrowers, household income was assumed to grow over time at a rate equal to the change in average personal income for a given county and year. Data from longitudinal studies of income dynamics suggest that in reality, the rate of income growth varies depending on a household's initial wealth, and that year-to-year changes in income can be highly volatile even within a given income stratum (Fisher et al., 2016). In addition, our modeling approach does not consider exogenous income shocks arising from events such as job loss, illness, or divorce. Including these sources of variability in household income would likely increase the number of mortgage borrowers projected to experience income-related credit constraints following exposure to flooding.

Fourth, our model framework does not account for how factors related to the 2008 global financial crisis (GFC) may have impacted the financial health and credit access of mortgage borrowers during the study period. These include elevated rates of unemployment that persisted for several years following the GFC and a tightening of mortgage lending standards that reduced the availability of credit to property owners. Mortgage lending standards in the U.S. underwent a gradual loosening during the early 2000s leading up to the crisis, followed by a sharp tightening during the 2007-2009 period that led to increases in loan denial rates and more stringent LTV and DTI requirements (Vojtech et al., 2020). Of the seven flood events evaluated in this study, the effects of the GFC would be most relevant for Hurricane Irene, which occurred in 2011 when the economy was still recovering from the crisis. If the elevated rate of unemployment and reduced credit supply during this period were incorporated into our model, projections of credit constraints among borrowers exposed to flooding from Hurricane Irene would likely be higher.

Finally, we did not explicitly model the various sources of funding for post-disaster recovery that might be available to uninsured mortgage borrowers who lack sufficient equity or liquidity to obtain private home repair loans. These include: federal sources of post-disaster aid such as SBA loans and FEMA IHP grants; alternative finance sources such as payday lenders, auto title loans, and pawnbrokers; and liquid assets such as personal savings and retirement accounts. To examine how the availability of low-interest SBA loans and FEMA IHP grants may impact our findings, we conducted scenario analyses on the interest rate at which borrowers can finance home repairs as well as the amount of home repair assistance available to those without insurance. The number of credit constrained mortgage borrowers was sensitive to the amount of grant aid available but relatively insensitive to the interest rate on home repair loans. During Hurricanes Matthew and Florence, less than a third of property owners who applied for IHP aid were approved, and the average grant awarded was under $5,000 (GAO, 2020b); thus, it appears unlikely that the inclusion of IHP aid would substantially alter estimates of the number of mortgage borrowers

facing flood-related credit constraints. Future research could examine how these programs are likely to shape the long-term recovery outcomes of credit constrained mortgage borrowers by explicitly incorporating the timing and distribution of post-disaster aid into the integrated modeling framework.

Future research could also build upon the integrated modeling framework developed in this study to analyze the cost-effectiveness of policy interventions to improve the post-flood financial resilience of U.S. households. By coupling the
financial components of our framework (models II-IV) with a probabilistic flood hazard event set, future studies could evaluate borrower outcomes over a wider range of plausible flood scenarios than the seven historical events examined in this study. While generating synthetic inundation footprints for probabilistic flood risk assessment is non-trivial, recent research has developed a suite of methods and datasets to support this task, particularly for tropical cyclone-induced flooding (Grimley et al., 2025; Nederhoff et al., 2024; Sarhadi et al., 2025). Pairing these approaches with simulations of household financial
conditions would allow for the expected costs and benefits of various policy interventions to be comprehensively assessed, including their impact on the share of mortgage borrowers projected to face flood-related credit constraints. Based on the findings of our study, insurance policies whose deductibles are tailored, or "right sized," to a household's borrowing capacity are an intervention worthy of examination in future policy analyses. For example, a mortgage borrower whose CLTV and DTI ratio allows them to take on $30,000 in additional debt could select a policy with a $15,000 deductible (equivalent to 50% of
their borrowing capacity). Although a high-deductible policy of this kind would not fully cover the cost of flood damage, it may help to reduce the probability of a mortgage defaulting by ensuring that the borrower can cover the remaining cost of repairs through home equity-based borrowing. Due to the higher deductible (which could be adjusted over time as the mortgage is repaid), such a policy could be offered at a lower premium than traditional flood insurance through the NFIP, which may make it an attractive option for areas outside the SFHA where uptake of flood insurance is quite low. Given that properties
located outside of the SFHA accounted for the majority of mortgages facing flood-related credit constraints, requiring such a policy on homes located in moderate-risk areas outside the SFHA (e.g., the FEMA 500-year floodplain) could potentially reduce the exposure of mortgage borrowers and their lenders to flood-related credit risk.

## 6 Conclusion

Over 40 million Americans live in flood-prone areas, many of whom are uninsured and just one storm away from
potentially losing their home (Wing et al., 2018). Although floods are a threat to rich and poor alike, the consequences of uninsured damage are much more severe for less wealthy and credit-insecure households who lack the borrowing capacity of those with substantial home equity and available income, reducing their ability to obtain funding for post-disaster recovery from traditional sources. This paper presents a novel, data-driven method for characterizing how the pre-flood financial conditions of residential mortgage borrowers (i.e., insurance status, equity, and liquidity) affect their ability to access low-cost
sources of financing for flood-related repairs. The findings of this analysis shed light on the relative contribution of negative equity and liquidity issues to credit constraints among flood-affected mortgages and provide information on the capacity of

property owners to fund repair and recovery efforts. These results and methodological approaches may inform the nature and targeting of interventions to improve the financial resilience of U.S. households by providing highly resolved information on which households and communities are likely to be credit constrained in the aftermath of a flood. While the focus of this work

is on flooding, the methods and modeling approach are generalizable to other natural hazards such as wind or wildfires.

## Code and data availability

This analysis was conducted using Python version 3.11 and R version 4.2.1. The code used in this analysis is available in a Zenodo repository (https://doi.org/10.5281/zenodo.15313723) and on GitHub (https://github.com/UNC-Cofires/flooding-financial-risk). Most data used in this analysis are publicly available; select data, including address-level NFIP policies and

claims, contain personally identifiable information and are not publicly available at the scale of individual properties. Address-level data on NFIP claims and policies were obtained through an Information Sharing Access Agreement (ISAA) between FEMA and the University of North Carolina at Chapel Hill. Data on residential real estate sales were purchased from ATTOM Data Solutions.

## Author contribution

GWC, AS, and HBZ conceived and designed the project. GWC and AS acquired funding and supervised the work. KPF, HG, and HT developed the model code and performed the simulations. KPF prepared the manuscript with contributions from all co-authors.

## Competing interests

The authors declare that they have no conflict of interest.

## Acknowledgements

The authors would like to thank the University of North Carolina at Chapel Hill and the Research Computing group for providing computational resources and support that have contributed to these research results. The authors are also grateful to the North Carolina Policy Collaboratory and FEMA for facilitating access to information on NFIP claims and policies.

**Financial support**

This research was supported by the North Carolina General Assembly and the North Carolina Policy Collaboratory through Session Law 2019-224. This work was also supported in part by the UNC Institute for the Environment and the National Oceanic and Atmospheric Administration (NOAA-OAR-CPO-2021-2006677).

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
