# Peer review of "Flood risks to the financial stability of residential mortgage borrowers: An integrated modeling approach"

_EGUsphere, 2025_

## Referee Comment (RC1)

**Review of "Flood risks to the financial stability of residential mortgage borrowers: An integrated modeling approach"**

**General comments**

The manuscript "Flood risks to the financial stability of residential mortgage borrowers: An integrated modeling approach" presents an impressive framework for evaluating how flooding impacts mortgage borrowers' financial stability. The authors develop a comprehensive modeling approach that integrates flood damage estimates, property values, mortgage balances, and insurance coverage to identify which borrowers face heightened exposure to mortgage default following flood events.

The study makes a valuable contribution by introducing a model framework that models the relationship between pre-flood financial conditions, flood impacts, and post-flood financial conditions. The integrated modeling approach allows for the assessment of different theorized causal pathways of mortgage default (strategic, cashflow, and double-trigger).

While I appreciate many aspects of the study, I believe it faces several limiting issues that I would like to see the authors address in a revision:

1. **Unclear framing**: The study lacks a clearly articulated research question that aligns with its methodological approach. While the introduction suggests the study addresses "how pre-flood financial conditions affect the relationship between uninsured damage exposure and post-flood risk of mortgage default," the methods and results don't directly answer this question. The study would benefit from explicitly stating what scientific questions it aims to answer and how its modeling approach addresses these questions.

2. **Absence of calibration/validation data for defaults**: Despite citing empirical research on drivers of default, the authors don't calibrate or validate their framework against observed mortgage outcomes. Without these crucial modeling steps, it's difficult to assess whether the modeled financial conditions offer predictive value. The utility of the integrated modeling framework is questionable without demonstrating its predictive accuracy for the outcome of interest, unless the authors pursue a more exploratory approach with more detailed uncertainty quantification and sensitivity analysis.

3. **Unconvincing causal mechanism of flood damage to default**: In light of the comment above, the findings from Kousky et al. (2020), a key reference for the authors, demonstrate significant concerns with using modeled damage estimates to predict mortgage outcomes. Their study shows that catastrophe model damage estimates—even those potentially more accurate than those in the current study because they come from a proprietary catastrophe model—found spurious relationships between predicted flood damage and default compared to results based on actual damage inspections. Specifically, they found that for rare events like default, "predicted damage needs to match better with actual damage at a property level in order to deliver a robust estimated impact." As another example, they found that when the catastrophe model predicted damage of less than 10%, the odds of deep delinquency or default increased, but not when the catastrophe model predicted greater than 10% damage. They wrote, "This counterintuitive risk ranking, which we have not seen in other loan performance outcomes, suggests that the inaccurate property-level damage prediction by the catastrophe model can be problematic for a rare outcome, such as deep delinquency or default." This raises fundamental questions about the reliability of the current study's approach to modeling default risk.

4. **Potential need for reframing study around sensitivity analysis**: Despite the complex integrated modeling approach, the paper doesn't sufficiently explore how uncertainties in model components propagate through to default projections. A more comprehensive uncertainty quantification and sensitivity analysis would strengthen the study by identifying which factors most influence projected outcomes and how robust the findings are to different assumptions. This approach would be particularly valuable given the lack of validation data and would better demonstrate the framework's utility for policy analysis.

5. **Missed opportunity for policy analysis**: The study introduces interesting policy analyses (such as the home repair grant program) but doesn't fully leverage its framework to explore how various policies could influence default rates under different scenarios and assumptions. A more thorough exploration of policy interventions, coupled with comprehensive sensitivity analysis, would significantly enhance the paper's contributions and better justify the development of the integrated modeling approach.

6. **Omission of important contextual factors**: The study takes a real-world framing, which is compelling and raises the stakes of the findings, but the model excludes real-world factors that would influence default outcomes, such as disaster aid programs, employment changes, and the effects of the 2008 financial crisis, without adequate justification for these simplifications.

**Specific comments**

*Abstract*

- The writing is generally clear, but the abstract should clarify the degree to which the study is model-based or observational (or both). For instance, is "Here, we evaluate the impact" in L14 an estimated impact or an observed impact (or maybe an estimated impact with methods calibrated to observed outcomes?) This is crucial to clarify because the motivation is about highly consequential (and overlooked in the literature) outcomes, such as mortgage delinquency, default, and foreclosure. The statement I quoted makes it seem like the study observes these outcomes. However, the next line talks about conditions "indicative of default" and then reports somewhat superficial statistics. Lacking insurance or income/collateral to finance repairs does not seem like a predictor of mortgage default in previous empirical studies. It is possible that I (and other readers) am unfamiliar with research showing this, so the authors should add context about the degree to which the conditions are strong indicators of default. Alternatively, if this is an observation-based study, do the authors find that is a different case in North Carolina? If so, this is a major result that the abstract should highlight more prominently.

- Some ambiguous grammar. For example, do the authors mean in L16-17 that they look at default *and* negative equity, or does negative equity refer to one of the financial conditions indicative of default? The ", including" grammar makes the remaining text unclear.

*Introduction*

- The details are strong and the authors are clearly knowledgeable about this topic, but long paragraphs and sentences make the narrative difficult to follow. Claims like "Given these gaps in existing knowledge" (L131) are ineffective because the authors do not clearly signal knowledge gaps in preceding paragraphs.
- Some of the introduction texts reads like a literature review, which disrupts the flow. It might be helpful to break out a succinct introduction that introduces the paper's focus and contributions, and a separate literature review section. Given that the abstract's focus on default, the paragraph starting on L81 is the most relevant, but we only hear about default after three long preceding paragraphs.
- The text on flood insurance and limited coverage are important, but I wonder if they would be more effective *after* going straight into the main text about financial instability. There is a lot of literature about natural disasters and financial instability – why don't the authors just start on their focus? The issues with the NFIP and flood-risk information are important contributors to these issues since a lack of insurance could be a major driver of bad financial outcomes for households, such as default.
- I don't know that the references in L77 are appropriate. It seems like the point here is that damaged properties are worth less on the market and home equity loan principal will be lower. I'm not sure that the cited studies about post-flood property prices are relevant here because these studies do not control for flood damages. Thus, it is unclear if the change in property prices after floods are due to a damage effect (or similar), and thus concentrated within segments of the housing market (which would be a form of double counting relative to the rest of this paragraph – if a property is structurally comprised, it is worth less and you can't take out a home improvement loan of the same value as pre-flood conditions). There are some studies that control for flood damage and find changes in post-flood prices unrelated to damage. The authors could cite Atreya and Ferreira (2015) - https://doi.org/10.1111/risa.12307 - but this is a small case study and there is conflicting evidence. See Davlasheridze and Fan (2019) - https://doi.org/10.1007/s41885-019-00045-z- or Pollack and Kaufmann (2022) - https://doi.org/10.1016/J.ECOLECON.2022.107350 for evidence on how and why property price changes may not be market-wide after a flood event and specifically indicate double counting in this context.
- There is something difficult about the logic of the key outcome of focus, default. The authors talk about how most of the damage from several of the largest storms in US

history was uninsured, and they also talk about a lack of insurance as a major indicator of mortgage default. So, where is the evidence from these storms that many uninsured households defaulted? The references to literature on this central framing point, starting on L81, would benefit substantially from reporting statistics from the papers to help the reader understand the connection between uninsured damage and default. The most specific statistic in this paragraph is the "50 times higher" one from Kousky, but this is on 90-day delinquency. While an important negative financial outcome, the current study primarily frames a focus on *default* (e.g., L16-17 of the abstract: "Our framework estimates key financial variables to identify borrowers exhibiting financial conditions of default"). The evidence on that from Calabrese et al., (2024) should state what the quantitative evidence is so readers understand how large the effect is.

- o I know the Kousky paper better than the Calabrese one, so I will focus on evidence in there that I believe the authors should pay more attention to in their framing, since they focus on mortgage default as the key outcome. Table 7 of that study seems to suggest that moderate to severe damage is the main predictor of 180 or more days delinquent or default. While the result for moderate to severe damage X in SFHA is not significant, that may be due to the very small sample size of treatment observations. The authors should highlight that damage amount is an important factor and can point out that the study doesn't control for whether a property is insured (which supports the authors' claims either way). However, the study combines 180 or more days delinquency and default into one outcome because the number of defaulted loans is not large enough for identifying a statistical effect. Out of 27,000 loans, there were only 24 defaults as of August 2019 (2 years after the storm). Overall, there are a rather small proportion of observations in the combined outcome. The authors should be transparent that there is not strong empirical evidence on the causal mechanisms their claims rely on.

- I like the Thomson paper a lot, but that study is fully model-based and assumes the causal mechanism of a house being underwater leading to default. As this is the main paragraph on empirical evidence relating uninsured damage and financial preconditions to the main outcome of interest, default, the authors should also review the following related empirical literature (mostly flood, but also related to wildfire risk). These studies generally support the work cited in this study, but some offer surprising insights into household financial resilience in the wake of large natural disasters that this study should reconcile in its framing:

  - o Biswas, S., Hossain, M., & Zink, D. (2023). California Wildfires, Property Damage, and Mortgage Repayment. Federal Reserve Board of Philadelphia Working Paper, 23-5.

- Mota, N., & Palim, M. (2024). Mortgage Performance and Home Sales for Damaged Homes Following Hurricane Harvey. Fannie Mae Working Paper Series.
- Del Valle, A., Scharlemann, T., & Shore, S. (2024). Household financial decision-making after natural disasters: Evidence from Hurricane Harvey. Journal of Financial and Quantitative Analysis, 1-27.
- Hopkins, C., Marr, A., & Wilson, N. (2024). How Does Mortgage Performance Vary Across Borrower Demographics Following a Hurricane? Federal Housing Finance Agency Working Paper Series.
- Issler, P., Stanton, R., Vergara-Alert, C., & Wallace, N. (2020). Mortgage markets with climate-change risk: Evidence from wildfires in california. Available at SSRN 3511843.
- Rossi, C. V. (2021). Assessing the impact of hurricane frequency and intensity on mortgage delinquency. Journal of Risk Management in Financial Institutions, 14(4), 426-442.
- Gallagher, Justin, and Daniel Hartley. 2017. "Household Finance after a Natural Disaster: The Case of Hurricane Katrina." American Economic Journal: Economic Policy 9 (3): 199–228.
- Deryugina, Tatyana, Laura Kawano, and Steven Levitt. 2018. "The Economic Impact of Hurricane Katrina on Its Victims: Evidence from Individual Tax Returns." American Economic Journal: Applied Economics 10 (2): 202–33.
- Deryugina, Tatyana. 2017. "The Fiscal Cost of Hurricanes: Disaster Aid versus Social Insurance." American Economic Journal: Economic Policy 9 (3): 168–98.

- In particular, Deryugina (2017) and Deryugina et al., (2018) discuss drivers of financial outcomes after natural disasters that are largely missing from the present study's framing. The first study, particularly important for the authors to engage with, investigates the role of non-disaster-based social insurance that can actually improve some households' well-being after disasters. The second examines the role of employment and income, highlighting the role of savings in supplementing households in the aftermath of Katrina (which seems relevant especially here because households outside the SFHA may have lower probability of flooding, so may have higher savings if they don't pay into the insurance program over time; this may not be the case with NFIP because of its risk-rating procedure before Risk Rating 2.0, but seems plausible and is worth mentioning).
- The paragraph on L119 seems too repetitive with previous paragraphs. Can the authors consolidate the presentation of each topic?
- The "gaps in existing knowledge" (L131) are unclear, and it's not clear the authors address them. It seems like the gap (as far as I can tell, the authors state only one knowledge gap) is "Although prior studies such as those by Kousky et al. (2020) and Calabrese et al. (2024) have examined the association between insurance uptake, flood exposure, and mortgage credit risk, there exists a need for additional research into how

the pre-flood financial conditions of a borrower (i.e., equity and liquidity) affect the relationship between uninsured damage exposure and the post-flood risk of mortgage default" (L127-130). However, the objectives of the study do not appear to reconcile the gap. While the authors describe a very impressive analytical workflow for linking flood damage to preconditions, the link between the preconditions and outcome of interest, default, is not addressed in this study. But the authors explicitly claim that there is additional research into how pre-flood financial conditions affect post-flood risk of mortgage default. The last paragraph reads as if the authors address something more like post-flood *exposure* to mortgage default – they link flood damage and pre-flood financial conditions to a post-flood financial state, but do not appear to build evidence on the link between those financial states and mortgage outcomes.

- The Introduction needs to streamline the narrative around the research gaps and what the study focuses on. The current structure does not flow well because it is unfocused. It reads as if the authors developed the analytical workflow and then backed out the research gap the workflow could address, but did not identify a clear science question.

- I think the Introduction needs to soften its claims about what the framework can achieve. The authors need to be more explicit and transparent about what the framework does (from what I have read so far, it is an advanced framework to estimate exposure to bad mortgage outcomes but does not estimate the risk of those outcomes).

- It is now clear that the study is fully model based, not even using data on financial outcomes to calibrate the projections of post-flood financial conditions or vet its performance. This is concerning given the current framing of the study. The main framing of the Intro is "there exists a need for additional research into how the pre-flood financial conditions of a borrower (i.e., equity and liquidity) affect the relationship between uninsured damage exposure and the post-flood risk of mortgage default" (L129-130)." How does this study address this need if there is no data on mortgage default outcomes? At this point, I see a few options to reconcile this. First, the authors might instead frame their study as estimating post-flood exposure to mortgage default. This seems defensible because in the typical risk framing, risk is the potential for adverse consequences, driven by interactions of hazards, exposure, and vulnerability. The workflow the authors describe does not appear to fit this definition. Second, the authors could use the three introduced theories on default causal pathways as their representations of vulnerability. This would enable them to take their exposure estimates and translate them into default outcomes, conditioned on the assumption that one of the theories represents a valid causal pathway. The authors raise nice evidence that these theories have weaknesses, so they would have to be careful in their framing if they take this approach. Third, and complementary to the second, it seems like the authors could take a more exploratory approach by embracing the uncertainty in the system and investigating how different assumptions (e.g., about causal pathways of default) and model uncertainties (e.g., both well-characterized uncertainty around damage projections and deeper structural

uncertainties about the integrated modeling chain) propagate into projections of default risk. I have a preference for the third option because it seems the most appropriate for the question stated on L129-130. Given the model-based approach of the study, with no observations on the outcome of interest, an exploratory modeling approach that identifies drivers of uncertainty in the outcome of interest appears the best way to investigate the relationship between uninsured damage exposure and the post-flood risk of mortgage default. The authors could also clarify their current framing in a revision and explain where I misunderstand a gap between their science question and methods.

*Methods*

- The first sentence of this section seems like a more modest and appropriate framing than what the Introduction suggests. It also seems to support the need for an exploratory modeling and sensitivity analysis approach. The authors should consider streamlining the Introduction around this.
- Why only use loan-level data for initial financial conditions? If there are new originated loans over the full time period, isn't it possible to identify changes in financial conditions as well? I'm not familiar with the HMDA data. Using only data for initialization leads to very strong assumptions about income and loans over a 23 year period that again seems to support a framing around exploratory modeling and sensitivity analysis.
- I'm confused by the claim that strategic, cashflow, and double-trigger mechanisms are types of defaults (L146-149). The Introduction text specifically frames these as theories of default and talks about limitations in the first two theories. Why are all three theories modeled then? I think it would be helpful to reframe the paper to accommodate the methods. It seems like a worthwhile exercise to map the exposure to three types of default mechanisms, as long as the authors provide more context in the Introduction for why. One reason could be that although there is very insightful empirical research on drivers of default in the context of natural disasters, we don't have a complete understanding of the causal mechanisms (data limitations, a limited number of events, etc.,). There are competing theories about these causal mechanisms, which we can represent with bottom-up models (if taking this approach, please provide evidence that these theories are prominent and inform decisions – which is necessary to support some claims from the Introduction). I think it would be easy to frame the contribution in this way, and helps the authors explain that as more research comes out on the causal mechanisms, their framework could adapt to those pathways and better model mortgage default risk.
- I don't understand why the authors simulate financial conditions at a monthly time step over the 1996-2019 period but only focus on the 7 largest (when they point out that the state faced 14 major disaster declarations over the period). I greatly appreciate the

transparency on this point. Can the authors please justify this modeling choice and explain its potential implications on their results? Why can't the authors simulate just at the storm time steps? What are the consequences of overlooking other major storms (in addition to other flooding events and possibly more important events that affect the outcome of interest such as the great financial recession of '08)?

- The stochastic sampling for certain variables (mortgage loan characteristics – what else?) again suggests the value of an exploratory modeling approach and sensitivity analysis. Is it just one draw from the tract distribution for each household? One draw could lead to spurious projections given the distribution of other model inputs that are correlated with the financial variables (but not sufficiently sampled with one draw).

- In general, the methods and text on cross validation are great and the authors are exceptional related to previous literature. Great job! However, since the point of this study appears to be about the integrated modeling approach to modeling financial stability, it's a first-order concern to investigate how sensitive the modeling framework is to uncertainties in the modeling steps and inputs. The validation does suggest sizable uncertainty in both interpolation & extrapolation, which begs the question: how much does this matter for projections of mortgage default? While I recognize this paper builds on methods under review elsewhere, it would be helpful to contextualize how the sensitivities in the underlying methods are particularly relevant with respect to this study's prediction goals. I think that given the interest in the connection between uninsured losses and mortgage defaults, the most relevant sensitivity is the degree to which the model may overestimate exposure and damage outside of the training data. The authors do a good job of talking about these uncertainties and how well their model does. But the authors probably recognize that there is something different about homes that don't have flood insurance than the homes that do (given the authors' framing around affordability and willingness to pay for insurance and their reference to Bradt et al., 2021, I think they recognize that there are different factors between these populations, including that houses facing higher hazard are more likely to purchase insurance even if they are outside the SFHA). There is also a selection of properties into the claims data, based on factors such as income, deductible, and loss size. I reviewed the Garcia et al., (2025) preprint and sections 4.2 and 4.3 in detail and it's hard to see how their sensitivity checks get at these concerns of selection. I think it's important for the authors, given their current framing and seeming data limitations on "validation" data for default, to incorporate uncertainty in the modeling steps and evaluate how those propagate into sensitivity in the mortgage default projections. I want to emphasize that the cross-validation approach is rigorous, and the authors did a great job writing about it, but in the context of this highly complex modeling workflow, it seems very important to evaluate how the uncertainties in each model component propagate. Again, this especially seems the case because the integrated modeling approach seems to be the study's main contribution (if this is not the case, the study needs to substantially reframe its title and introduction).

- I don't see how the neighborhood-level cross validation results for the damage model are relevant if the outcome of interest gets simulated at the property level. The relevant validation metrics are at the property level, where errors can interact with other uncertain inputs and could propagate into errors in the default projections. The low $R^2$ is not surprising given previous research but is concerning and it would be very insightful to see how uncertainty in damage at the property-level propagates (especially as it interacts with other uncertain inputs). The later sensitivity analysis that appears to uniformly lower and increase property-level damages by 20% does not justify if 20% is enough. The Wing et al., (2020) study that the authors cite suggest that for most inundation depth, damage at the property-level can vary from 0% to 100% of a structure's value, and that the damage is heteroskedastic with inundation. This suggests that a uniform 20% adjustment is not sufficient for sampling uncertainty and seeing how it propagates.
- I would like to know more about the hedonic model. In particular, can the authors provide more information about the ATTOM dataset? Are the coordinates tax parcels or building footprints? Are the records complete across the state in all observable characteristics? For more context on why it's important to provide more details about this dataset, please see Nolte, Christoph, et al. "Data practices for studying the impacts of environmental amenities and hazards with nationwide property data." Land Economics 100.1 (2024): 200-221.
- How does the property value model account for the existence of flood exposure, events, and damage over the study period? Is there any heterogeneity in property valuation based on exposure, structural defense, or damage? On that note, does the damage model have input features related to structural defense? In addition to the overall sensitivity of the model, it is important to consider sensitivity of the default projections based on heterogeneity in inputs. The results aggregate on geographic and economic factors that may be relevant to flood resilience policy – what about accounting for those factors in modeling the outcome?
- How many samples for each property from the HMDA data? How do the authors account for correlation in income and factors such as property value when sampling from the HMDA data? Accounting for the correlation structure in the HMDA data is great, but it's crucial how those draws are assigned to properties with different hazard, exposure, and vulnerability characteristics.
- Is there anything like Fig. S9 for defaults?
- I don't think Model IV has enough information. How does it account for things like heterogeneous savings for households that are insured or not? With the same income, a household without insurance would accrue more savings by not paying insurance costs. At the property level, there is also a difference in the degree to which a household saves money on purchasing their home based on salient price signals for risk (particularly, in North Carolina Pope (2008) showed how seller disclosures improved price signals to buyers looking to live in the SFHA: https://doi.org/10.3368/le.84.4.551). Seems like the

model assumes everyone's income goes up? There is no job loss? What do recovery trajectories look like after flood events? Is it only possible through loans for uninsured properties? Does income grow for all homeowner types based on county-level annual trends? Do damaged properties recover value over some time period? Does price appreciation/depreciation occur differently for insured and uninsured properties? The authors mentioned some factors like employment and income in some of their mechanisms for default, but I don't see how it's incorporated in the modeling framework in a realistic way. Can the authors talk more about what their assumptions are about exogenous factors over the sample? It would help to have a conceptual model of household finances and indicate what this study includes/excludes and why.

- I recognize that the methods state the study does not look at aid, but why not? How do the results of Deryugina (2017) on social insurance relate to recovery trajectories? In the US, IHP actually requires households to purchase insurance as a condition of the aid – why exclude this type of mechanism from the integrated workflow? It is crucial that the authors justify modeling choices based on their study framing, not based on simplicity. Modeling choices based on simplicity may not always be appropriate for satisfying a study's goals. I'm not sure that the current choices around simplicity are justifiable given the current framing. The study uses real historical flood events and projects defaults based on different theories of how financial conditions produce defaults, but ignores other real-world characteristics that would mediate these outcomes. The "real-world" framing requires very careful justification for departures from reality.

- How does this framework account for the '08 financial recession, which occurs right before Irene? Does the property model adequately project the drop in value during this period? Do the financial models adequately account for the rise in poor financial conditions (and I assume defaults) at the household level? It also begs the question of contextualizing projected defaults from flood stress relative to defaults from the '08 recession. It seems to me that neglecting important exogenous drivers of poor financial conditions could lead to both under estimation in this modeling workflow. The decrease in property values and employment from the great recession, plus Irene in 2011, might mean the model underestimates a large default event based on the double-trigger mechanism?

*Results*

- What is "flood damage exposure" in Fig 3 and 4a? Is it damage to structures?
- I'm surprised by the amount of attention in the results to describing the damage estimates both overall and stratified across a few groups. I think the authors should restructure their results to focus on their main goal, which is the degree to which default occurs under

different assumptions of causal pathways of damage, financial pre-conditions, insurance, and mortgage debt.

- L576-577, what is the proportion of both located outside SFHA and lacking flood exposure?

- The results on default would be more interpretable if the authors showed how many defaults there would be under the three different causal pathways of default. Most readers will not be able to interpret the 6 metrics of Figure 6 and what it means for default projections. If not this, it could really help to reintroduce the acronyms in the results section and to not use acronyms in the figure caption. But I highly encourage taking the conditions required for certain default mechanisms (e.g., ACLTV > 100% and ADTI > 45% for double-trigger) and showing the proportion of households pre & post flood that met those conditions. I encourage the authors to really focus on their key outcome of interest and focus on interpretability since their topic is extremely important and many readers of this journal come from different disciplines and will need help understanding new concepts.

- I think it could be helpful to compare the default results for both uninsured and insured properties, perhaps to emphasize the additional risk of being uninsured. However, to do this comparison, one would have to account for the additional savings uninsured households might accrue by not paying for flood insurance. While the sample period predates Risk Rating 2.0, that is an especially important consideration as the price of insurance goes up. Also important is that many households, even if insured, need to rely on savings for some time after a large event due to slow timing in damage assessments and payouts. This ties into my questioning about whether the "real-world" framing is appropriate.

- I also think it would be helpful to compare the default projections under the floods versus alternative "normal" rates of defaults (some households go under hard times and may have to default) and "financial crisis" rates of defaults (like '08) to help contextualize the projections.

- The modeling assumptions seem to force the modeling results, particularly that uninsured households have few ways to financially recover between events and have to wait for their income to bounce back. What happens if the main income bringer lose their jobs (insured or uninsured households)? What happens if uninsured households end up with more aid? I'm not suggesting the study has to address all of these questions in its model, but I do think it needs to acknowledge how certain modeling assumptions will lead to certain outcomes when making comparisons across groups. What is left out of the model that might overstate the differences in default outcomes across groups of comparison? This is one reason a conceptual model could help.

- I think comparing the number of at-risk default by causal pathways is great, which is what Figures 7 & 8 do, but I'm confused about the result. Because these are not mutually exclusively, these are likely not the correct visualization types for this result. Separately,

it would help to contextualize these results in terms of the overall residential stock to help readers understand how large this problem is. Further, I think the current framing of the study leads the reader to expect much more results focused on these causal pathways and the default outcome. As I mentioned in my comments on the methods, I think it would be very effective to focus on how uncertainty in inputs & models propagates through the integrated modeling workflow and leads to different projections of default outcomes, conditioned on the different causal pathways one believes.

- For Figure 8, proportions could be helpful to contextualize the results (like the % for the other bar charts). 22,100 damaged homes out of 4.7M is important context when thinking about multisector impacts. Are we talking about financial risks that affect homeowners exclusively, or are there cascading impacts across sectors? It would be helpful to see more about whether 11,000 is a large number or not in this setting.

- The text about liquidity (L594-L604) raises questions about the HMDA sampling. Is it possible that the sampling assigns lower incomes and higher loan amounts to certain high damage households? Looking at results in terms of the sensitivity analysis I suggested above could help readers understand what drives different default outcomes in terms of modeling assumptions. As it currently is, it seems like the authors may be overinterpreting results strongly reliant on plausible but highly uncertain and insufficiently sampled modeling implementation.

- I think it is important not to overinterpret results, which I am worried about for the results shown in Figures 7 and 8. The authors could draw stronger conclusions about liquidity, income, property values, and defaults if they took more of a sensitivity analysis approach and implemented a method like scenario discovery.

- It's great to see the sensitivity analysis about the home repair grant program. It reinforces for me that the strongest insights in this paper would come from a more comprehensive approach to characterizing the drivers of sensitivity in default projections. I think it is the best way for the authors to make an important and reusable contribution in this area. In particular, the current default model results occur under the assumption that there is no grant for home repair, but there are major monetary transfers after large disasters (specifically presidentially declared disasters such as the focus of this study) so it is invalid to ignore this in the "baseline" case. The IHP program specifically requires uninsured households to purchase NFIP insurance, so that requirement seems like "good" policy in terms of reducing default (compliance is a different story, not modeled here but part of the data generating process). This again reinforces why a sensitivity analysis approach would be constructive and insightful. It would be unfortunate to overestimate default risk relative to the current policy environment (though with recent changes to FEMA the authors could frame some of their study around how important these programs are to avoid default risk!).

- It would have been nice to signal earlier to readers that the authors do a sensitivity analysis on damage costs and property values. This sensitivity analysis approach needs a

better description of the methods and a justification for whether it's valid to do a uniform application of the uncertainty instead of randomly sampling from an uncertain distribution for each unit. I'm not sure that it is. For example, the methods show that for the property valuation model, only 54% of observations fall within +/-20%. Given the large uncertainty in the projected outcome based on this under-representation of uncertainty, it seems very important to better contextualize sensitivity of the projected outcome to the actual uncertainty in the property valuation model. In addition, as mentioned earlier, 20% seems like an inadequate representation of uncertainty for the damage model, especially given previous findings on heteroskedasticity with inundation depth.

- It would be better to sample from uncertain factors using an appropriate experimental design at the spatial resolution of the model inputs (i.e., property-level). It would be very insightful for the authors to evaluate at least first-order sensitivities of the default projections to uncertainty in inputs and the authors could interpret results using relatively fast methods such as Method of Morris and scenario discovery. Since there are only ~22k houses under study for the default analysis, it seems like this is feasible. Several of the co-authors are more expert in these methods than I am, so I am interested to hear more about why they did not approach this complex new modeling workflow in a sensitivity analysis approach (e.g., among many works by Saltelli, please see https://www.jstor.org/stable/2676831 given the predominant framing of the integrated modeling workflow).

*Discussion*

- Nice, tight framing in the first paragraph. I encourage the authors to reflect on the differences in this framing to that of the Introduction, and to better synchronize the two sections to have a consistent story throughout the paper. In addition, the Results should focus on the main story.
- One issue with the first paragraph is the claim on L683-684: "Our results underscore the status of pre-flood home equity and debt-to-income ratio as important determinants of post-flood financial resilience." Is that true, or is that an artifact of implementing theories on causal pathway of default that treat these as determinants of default?
- The discussion is well-written. It also introduces claims that point to some unfocused results on the distribution of damages, as opposed to the most interesting points about the default projections and the uncertain factors surrounding those. For instance, the paragraph on L741 is very interesting. Why didn't the authors model this insurance policy with a deductible equal to 50% of a borrower's equity? Seems like to really illustrate the value of this integrated modeling approach, showing that an end-user can use the approach to stress-test candidate policies against a range of uncertain factors

seems appropriate and very useful to the research and policy community. Just speculating about this when the authors have the tool at their disposal to test their hypotheses is underwhelming.

- I'm confused by the 67,000 exposure estimates on L758. Why make a comparison to this number if the main number of interest is the ~ 22,000 damaged properties with mortgages?

- The discussion does a nice job of mentioning limitations, but generally doesn't justify why the study did not account for some uncertain factors. The idea that the Matthew and Florence grants were small and wouldn't affect the number of mortgages at risk of default is something the authors can test with their framework. Why just speculate?

- What's missing from the discussion is more acknowledgement about the deep uncertainty about whether the outcomes the authors measure actually are strong predictors of defaults. As discussed in the comments on the Introduction, the authors do not present that evidence. I think mortgage default *exposure* framing might be more appropriate than mortgage default risk, unless the authors clarify that the default projections are conditioned on the belief one causal mechanism holds. This calls for more of a deep uncertainty framing to the study, which would work well with the sensitivity analysis approach that I think this complex, prediction-based modeling integration study calls for.

**Technical corrections**

- L26: "losses from flooding are expected to surpass…" -> not sure this reference helps or is accurate. For one, it doesn't seem like any of the framing points rely on claims about changes in flood risk over time. Second, this is just one study and the reference reads more into the study than the study's findings support. For example, using the passive voice here for "are expected" makes this seem like the estimate is a confident one. It would be more appropriate to say that one study estimates annual losses from flooding may surpass $40B… I also think the claim that losses will surpass $40B as a result of increases in extreme precipitation under climate change is not supported by this reference. The study does not highlight the role of changes in extreme precipitation in changing risk estimates.

- L38 – the text about household ability and willingness to pay and demand could benefit from more specificity. What do the authors mean by "further reduces demand" on L39? Depending on whether they mean the demand curve or quantity demanded, it may be more accurate to say "which reflect low demand."

- The Gourevitch et al., (2023) reference on L125 does not seem to apply here. Where is the event-based specification of Gourevitch? They estimate "overvaluation" based on the degree to which the market capitalized information about properties mapped into the SFHA between sales. The studies that the authors cited on L77 would be more appropriate here, but note the references I mentioned in my comment that suggest the evidence on post-flood price adjustments is mixed (with more detailed specifications on

drivers of risk and household characteristics showing a more heterogeneous market adjustment).

---

## Author Comment (AC1)

**Response to Reviewer #1**

**Author's remarks**

We thank the reviewer for their valuable comments and suggestions for improvement. We appreciate the time and effort they put into reviewing our manuscript and find the research and writing to be much improved after incorporating many of their suggestions.

In light of the reviewer's feedback, we have reframed the analysis around flood-related credit constraints and reduced the emphasis placed on mortgage default as the main outcome of interest. As the reviewer noted, our model framework estimates the degree to which uninsured flood damage increased the potential for financial states (e.g., negative equity and cashflow problems) that can impair a mortgage borrower's ability to fund repairs to their property through additional borrowing; however, given the lack of data on mortgage performance, the degree to which these states are predictive of default remains unclear. Whether negative equity or cashflow problems trigger a borrower to default is also likely to depend on the availability of alternative sources of funding for home repairs that were not explicitly modeled, including disaster assistance grants, personal savings, and informal transfers from family and friends. As such, we have moved away from describing these borrowers as "at risk of default" and instead describe them as "credit constrained," reflecting their diminished capacity to fund repairs by taking on debt while acknowledging that their long-term recovery outcomes are uncertain and depend on several factors not captured by our analysis. We believe that this more modest framing aligns our study objectives and research questions more closely with our methodological approach.

As part of this reframing, we have made substantial changes to the introduction and background sections of the manuscript, and have modified our terminology as follows:

| Old terminology                     | New terminology                             |
|-------------------------------------|---------------------------------------------|
| "at risk of default"                | "credit constrained"                        |
| "at risk of strategic default"      | "collateral constrained"                    |
| "at risk of cashflow default"       | "income constrained"                        |
| "at risk of double-trigger default" | "constrained by both income and collateral" |

We hope that these changes will address the reviewer's main concerns regarding the manuscript. A point-by-point response to review is included below, with reviewer comments shown in **black** and our replies in **blue**.

**General comments**

The manuscript "Flood risks to the financial stability of residential mortgage borrowers: An integrated modeling approach" presents an impressive framework for evaluating how flooding impacts mortgage borrowers' financial stability. The authors develop a comprehensive modeling approach that integrates flood damage estimates, property values, mortgage balances, and insurance coverage to identify which borrowers face heightened exposure to mortgage default following flood events.

The study makes a valuable contribution by introducing a model framework that models the relationship between pre-flood financial conditions, flood impacts, and post-flood financial conditions. The integrated modeling approach allows for the assessment of different theorized causal pathways of mortgage default (strategic, cashflow, and double-trigger).

**Our reply:** We thank the reviewer for their thoughtful evaluation of our manuscript.

While I appreciate many aspects of the study, I believe it faces several limiting issues that I would like to see the authors address in a revision:

1. Unclear framing: The study lacks a clearly articulated research question that aligns with its methodological approach. While the introduction suggests the study addresses "how pre-flood financial conditions affect the relationship between uninsured damage exposure and post-flood risk of mortgage default," the methods and results don't directly answer this question. The study would benefit from explicitly stating what scientific questions it aims to answer and how its modeling approach addresses these questions.

Our reply: Thank you for identifying this issue with our manuscript. As described above, we have reframed our analysis around post-flood credit constraints to ensure that our study objectives and research questions align with our methodological approach. We have added the following text to the introduction that explicitly states the scientific questions our analysis seeks to answer.

Revised text from lines 57-61: In this study, we use an integrated modeling approach to simulate the impact of flood-related property damage on residential mortgage borrowers' financial conditions over a series of floods in North Carolina from 1996-2019 while examining the following research questions: (1) How much of the damage from these events was uninsured? (2) What share of flood-exposed borrowers faced credit constraints that were likely to impair their ability to access home repair loans? (3) Were these credit constraints driven by insufficient income, insufficient collateral, or both in combination?

**2. Absence of calibration/validation data for defaults:** Despite citing empirical research on drivers of default, the authors don't calibrate or validate their framework against observed mortgage outcomes. Without these crucial modeling steps, it's difficult to assess whether the modeled financial conditions offer predictive value. The utility of the integrated modeling

framework is questionable without demonstrating its predictive accuracy for the outcome of interest, unless the authors pursue a more exploratory approach with more detailed uncertainty quantification and sensitivity analysis.

Our reply: We agree that the lack of data on observed mortgage defaults is a limitation. Unfortunately, we do not currently have access to loan performance data at the spatial resolution needed to compare the outputs of our model framework against historical rates of mortgage delinquency and default. While there is evidence linking credit constraints to elevated rates of personal bankruptcy and mortgage delinquency following disasters (Billings et al., 2022; Collier et al., 2024) this question is not directly tested by our analysis. With this in mind, we have highlighted how future research could use data on observed defaults to translate our projections of post-flood credit constraints into estimates of expected credit losses.

Revised text from lines 571-576: It is important to note the ACLTV and ADTI thresholds employed in this framework are assumed to be necessary (but not sufficient) conditions for financial distress; as such, the credit constraint estimates generated by our procedure reflect the share of flood-exposed borrowers who may be forced to rely on other (less reliable) sources of funding for recovery such as savings, post-disaster aid, and support from family and friends. Additional information linking the post-flood financial conditions of mortgage borrowers to the probability of bankruptcy and default could be used to translate the estimates generated by our approach into projections of lender credit losses (Bellini, 2019).

3. Unconvincing causal mechanism of flood damage to default: In light of the comment above, the findings from Kousky et al. (2020), a key reference for the authors, demonstrate significant concerns with using modeled damage estimates to predict mortgage outcomes. Their study shows that catastrophe model damage estimates—even those potentially more accurate than those in the current study because they come from a proprietary catastrophe model—found spurious relationships between predicted flood damage and default compared to results based on actual damage inspections. Specifically, they found that for rare events like default, "predicted damage needs to match better with actual damage at a property level in order to deliver a robust estimated impact." As another example, they found that when the catastrophe model predicted damage of less than 10%, the odds of deep delinquency or default increased, but not when the catastrophe model predicted greater than 10% damage. They wrote, "This counter- intuitive risk ranking, which we have not seen in other loan performance outcomes, suggests that the inaccurate property-level damage prediction by the catastrophe model can be problematic for a rare outcome, such as deep delinquency or default." This raises fundamental questions about the reliability of the current study's approach to modeling default risk.

Our reply: We recognize that substantial uncertainty in property-level loss estimates continues to be a major challenge in flood damage assessments, including those presented in our study. Prior studies of building inspection data have found that flood damage costs are highly variable

even when stratified by inundation depth and building type (Freni et al., 2010; Merz et al., 2004; Pollack et al., 2022; Wing et al., 2020). We observe similar patterns in our analysis, finding that our damage models explain only a small fraction of the variance in damage costs at the individual property level but produce reasonable estimates of neighborhood-level losses. While this might be problematic if our intention was to predict the post-flood outcomes of specific mortgage borrowers, we believe the damage estimates used in our study are sufficient for modeling the distribution of financial conditions within a given neighborhood or census tract, which is the primary goal of our simulation-based approach. We have included text in the manuscript to clarify that our simulation model does not represent the exact conditions experienced by any specific borrower and have added a sensitivity analysis that explores the influence of uncertainty in property-level estimates of flood damage on projections of post-flood credit constraints. Further details regarding this sensitivity analysis are provided in our response to the reviewer's next comment.

Revised text from lines 248-253: Where possible, our modeling framework incorporates property-specific data (e.g., structure characteristics, past sales); certain variables that are only available at the census tract level (e.g., mortgage loan characteristics) are stochastically sampled to create synthetic values for individual properties. As such, the estimates produced by our simulation model do not represent the exact conditions experienced by any specific borrower but are intended to reflect the distribution of key financial variables within a given census tract—a spatial resolution that is likely to be relevant for the targeting of policy interventions and post-disaster aid.

**4. Potential need for reframing study around sensitivity analysis:** Despite the complex integrated modeling approach, the paper doesn't sufficiently explore how uncertainties in model components propagate through to default projections. A more comprehensive uncertainty quantification and sensitivity analysis would strengthen the study by identifying which factors most influence projected outcomes and how robust the findings are to different assumptions. This approach would be particularly valuable given the lack of validation data and would better demonstrate the framework's utility for policy analysis.

Our reply: We have added a variance-based sensitivity analysis that utilizes the method of Sobol' to evaluate how uncertainty in estimates of flood damage (model I), property value (model II), and borrower income (model IV) contribute to variance in borrower-level financial outcomes. These parameters were selected because they represent the primary drivers of flood-related credit constraints and are used directly within the calculation of damage-adjusted combined loan-to-value (ACLTV) and debt-to-income (ADTI) ratios for flood-exposed borrowers. Sensitivity analysis results suggest that error in property value estimates represents the largest source of uncertainty in our projections of flood-related credit constraints. We thank the reviewer for prompting us to investigate this issue, as we believe these new analyses

substantially strengthen the manuscript.

**Revised text from lines 799-810:** In a variance-based sensitivity analysis, we evaluated how uncertainty in estimates of flood damage (model I), property value (model II), and income (model IV) contributed to variation in borrower-level financial outcomes. Ranking uncertain parameters by their Sobol' total effect index (Saltelli et al., 2010; Sobol', 1993, 2001) for the outcome of a borrower being credit constrained due to either insufficient collateral or income (ACLTV > 100% or ADTI > 45%) revealed property value as the most influential parameter for 79% of flood-exposed borrowers (Table S8). Averaged over the simulated borrower population, the Sobol' total effect index of property value was approximately twice that of damage costs, the second-most influential parameter for this outcome (Fig. S21d). Uncertainty in estimates of damage costs had the greatest influence on the outcome of a borrower being simultaneously constrained by collateral and income (ACLTV > 100% and ADTI > 45%) (Table S8, Fig. S21c), likely because of the higher average level of flood damage required to trigger both constraints together than either constraint individually. Overall, these findings suggest that error associated with the estimation of property values and damage costs represent the largest sources of uncertainty in our projections of flood-related credit constraints. Additional details regarding our analysis of parameter uncertainty are available in Text S3 of the supplementary information.

**Text S3 in the Supplementary Information:**

To better understand the contribution of key model parameters to uncertainty in the postflood financial conditions of mortgage borrowers, we conducted a variance-based sensitivity analysis using the method of Sobol' (Sobol', 1993, 2001). This approach decomposes the variance of model outputs into terms that can be attributed to uncertain input parameters and their interactions. In our analysis, we focused on uncertainty in the following components of our integrated modeling framework: damage costs (model I), property values (model II), and borrower incomes (model IV) at the time of their flood exposure. These parameters were selected because they represent the primary drivers of flood-related credit constraints and are used directly within the calculation of combined loan-to-value (ACLTV) and debt-to-income (ADTI) ratios for flood-exposed borrowers. When examining how uncertainty in these input parameters contributes to uncertainty in model outputs, we focused on the following outcomes of interest: (1) the outcome of a borrower being collateral constrained (ACLTV > 100%), (2) the outcome of a borrower being income constrained (ADTI > 45%), (3) the outcome of being constrained by both measures (ACLTV  $\geq$  100% and ADTI  $\geq$  45%), and (4) the outcome of being constrained by either measure (ACLTV > 100% or ADTI > 45%). Because these model inputs and outcomes of interest are defined at the level of individual borrowers, sensitivity indices were calculated separately for each borrower based on their simulated financial conditions at the time of flood exposure.

Damage costs were assumed to follow a lognormal distribution with a mean equal to the model-predicted cost at each property location and variance estimated from cross-validation residuals using the conditional variance estimator of Fan and Yao (1998). This approach allows the amount of variance in damage costs to vary as a smooth function of the mean estimate, reflecting the higher uncertainty in total costs for properties predicted to have severe damage (Fig. S20). These mean-variance relationships were fit separately for each of the seven evaluated flood events.

Property values were assumed to follow a lognormal distribution with a mean equal to the model-predicted property value at each location and variance estimated via space-time interpolation of hedonic residuals using the simple lognormal kriging method (Chilès and Delfiner, 2012, p.150, 193). Because the kriging method provides an estimate of the error variance at each prediction point, it is well-suited for characterizing the uncertainty in property value estimates.

Borrower income was assumed to evolve over time as a stochastic process following geometric Brownian motion (GBM). GBM is frequently used to model the evolution of asset prices and other financial quantities that are assumed to be lognormally distributed (Hull, 2018). For each borrower, the initial conditions of this process were specified based on their simulated income at the time of mortgage origination ( $I_{t_0}$ ). In timepoints following origination, their income is modeled according to GBM as a lognormal distribution with the following mean and variance:

$$E[I_t] = I_{t_0} e^{\mu(t - t_0)} \tag{S3}$$

$$V[I_t] = I_{t_0}^2 e^{2\mu(t-t_0)} \left( e^{\sigma^2(t-t_0)} - 1 \right)$$
 (S4)

where  $\mu$  and  $\sigma$  represent the expected annual growth and annualized volatility of borrower income respectively. For each borrower,  $\mu$  was calculated based on the average continuously-compounded growth in per-capita income in their county of residence since the time of origination (BEA, 2023). The value of  $\sigma$  was fixed at 7% per year; this assumption is loosely based on Figure 3 of Dynan et al. (2012), who observed that the standard deviation of two-year changes in income for households in the middle 50% of the income distribution was approximately 10% during the 1971-2008 period (10% /  $\sqrt{2} \approx 7\%$ ).

First order and total effect Sobol' indices were calculated using the estimator of Saltelli et al. (2010) implemented by the SciPy Python library (Virtanen et al., 2020). The first order index  $(S_i)$  reflects the share of output variability that can be directly explained by a given parameter in isolation while ignoring interaction effects with other inputs. The total effect index  $(S_{T_i})$  reflects the share of output variability that a given parameter contributes to either directly or through its interactions with other variables. For each borrower, our calculation procedure results in a total of 12 index pairs (3 input parameters × 4 outcomes of interest). To evaluate the relative contribution different parameters to uncertainty in model outputs, parameters were ranked

individually for each borrower based on the total effect index, and the frequency of different ranking orders summarized across the simulated population of mortgage borrowers (Table S8). Similarly, population-averaged index values were computed by weighing the Sobol' indices of individual borrowers by the variance in their outcomes of interest:

$$\overline{S}_{i} = \frac{\sum_{k=1}^{N} V_{k} S_{ik}}{\sum_{k=1}^{N} V_{k}}$$
 (S5)

$$\overline{S_{T_{l}}} = \frac{\sum_{k=1}^{N} V_{k} S_{T_{ik}}}{\sum_{k=1}^{N} V_{k}}$$
 (S6)

In Equations S5 and S6,  $V_k$  denotes the variance of the outcome of interest for borrower k, while  $\overline{S_l}$  and  $\overline{S_{T_l}}$  denote the weighted average first order and total effect indices (respectively) of parameter i across the population. Weighted average index values for different parameter-outcome combinations are displayed in Figure S21.

**Table S8 in the Supplementary Information:**

Table S8. Ranking uncertain parameters by their influence on borrower outcomes.

| Outcome of interest / Parameter ranking a | Parameter ranking frequency b (%) |             |        |
|------------------------------------------------------|----------------------------------------------|-------------|--------|
|                                                      | Property value                               | Damage cost | Income |
| Collateral constrained (ACLTV > 100%)                |                                              |             |        |
| 1 st most influential parameter           | 95.9                                         | 4.1         | 0.0    |
| 2 nd most influential parameter           | 4.1                                          | 95.8        | 0.0    |
| 3 rd most influential parameter           | 0.0                                          | 0.0         | 100.0  |
| Income constrained (ADTI > 45%)                      |                                              |             |        |
| 1 st most influential parameter           | 1.9°                                         | 58.3        | 39.9   |
| 2 nd most influential parameter           | $0.4^{\rm c}$                                | 39.8        | 59.7   |
| 3 rd most influential parameter           | 97.7                                         | 1.9         | 0.4    |
| Constrained by both (ACLTV > 100% and ADTI > 45%)    |                                              |             |        |
| 1 st most influential parameter           | 14.8                                         | 70.6        | 14.7   |
| 2 nd most influential parameter           | 12.6                                         | 22.8        | 64.5   |
| 3 rd most influential parameter           | 72.6                                         | 6.6         | 20.8   |
| Constrained by either (ACLTV > 100% or ADTI > 45%)   |                                              |             |        |
| 1 st most influential parameter           | 78.8                                         | 9.2         | 12.0   |
| 2 nd most influential parameter           | 11.1                                         | 79.7        | 9.2    |
| 3 rd most influential parameter           | 10.1                                         | 11.1        | 78.8   |

ACLTV: Adjusted combined loan-to-value ratio. ADTI: Adjusted debt-to-income ratio.

'In theory, property value should have no influence on the outcome of a borrower being income constrained. However, this parameter occasionally has a non-zero Sobol' total effect index due to numerical error in the calculation. This issue only occurs when the amount of variance in the outcome of interest is close to zero.

<sup>aFor each borrower, uncertain parameters are ranked from most to least influential based on their Sobol' total effect index for the outcome of interest.

<sup>bRanking frequencies reflect the share of flood-exposed borrowers for which a given parameter was found to be the *n*th most influential.

**Figure S20 in the Supplementary Information:**

Figure S20. Uncertainty in damage costs at flooded properties.

In sensitivity analysis, damage costs were assumed to follow a lognormal distribution with a mean equal to the model-predicted cost and variance estimated from cross-validation residuals using the conditional variance estimator of Fan and Yao (1998). In the above figure, the conditional means and 95% credible intervals of the fitted lognormal distributions for each event are denoted by black and red lines respectively.

**Figure S21 in the Supplementary Information:**

**Figure S21.** Weighted average Sobol' indices decomposing the relative importance of property value, damage cost, and income in determining the outcome of borrowers being credit constrained following flood exposure.

Population averages are calculated by weighing the Sobol' indices of individual borrowers by the variance in their credit constraint outcomes. Results are shown separately for (a) the outcome of being collateral constrained, (b) the outcome of being income constrained, (c) the outcome of being constrained by both measures, and (d) the outcome of being constrained by either measure. Darker bars indicate first-order effects, while lighter bars indicate interaction effects.

**5. Missed opportunity for policy analysis:** The study introduces interesting policy analyses (such as the home repair grant program) but doesn't fully leverage its framework to explore how various policies could influence default rates under different scenarios and assumptions. A more thorough exploration of policy interventions, coupled with comprehensive sensitivity analysis, would significantly enhance the paper's contributions and better justify the development of the integrated modeling approach.

Our reply: We agree that there is an urgent need to compare and evaluate the impact of different policy interventions on household financial conditions; however, this question is beyond the scope of our present study, which introduces the integrated modeling framework, describes its development, and demonstrates how it can be used to generate projections of flood-related credit constraints for historical flood events. In future work, we plan to use the model components developed in this study to perform a detailed policy analysis examining different interventions for improving household financial resilience to floods such as novel insurance products and expansions to post-disaster aid programs. Quantifying the cost-effectiveness of these interventions would likely require the use of a stochastic event catalog that includes a broader set of events than the seven historical floods examined in this study. We have added text to the discussion highlighting these important areas of future research.

Revised text from lines 928-936: Future research could also build upon the integrated modeling framework developed in this study to analyze the cost-effectiveness of policy interventions to improve the post-flood financial resilience of U.S. households. By coupling the financial components of our framework (models II-IV) with a probabilistic flood hazard event set, future studies could evaluate borrower outcomes over a wider range of plausible flood scenarios than the seven historical events examined in this study. While generating synthetic inundation footprints for probabilistic flood risk assessment is non-trivial, recent research has developed a suite of methods and datasets to support this task, particularly for tropical cyclone-induced flooding (Grimley et al., 2025; Nederhoff et al., 2024; Sarhadi et al., 2025). Pairing these approaches with simulations of household financial conditions would allow for the expected costs and benefits of various policy interventions to be comprehensively assessed, including their impact on the share of mortgage borrowers projected to face flood-related credit constraints.

**6. Omission of important contextual factors:** The study takes a real-world framing, which is compelling and raises the stakes of the findings, but the model excludes real-world factors that would influence default outcomes, such as disaster aid programs, employment changes, and the effects of the 2008 financial crisis, without adequate justification for these simplifications.

Our reply: Developing our model framework required us to make several simplifying assumptions; where possible, we chose those that would yield more conservative projections of post-flood credit constraints, and we used sensitivity analysis to assess the impact of alternative assumptions on our findings.

Although we did not explicitly model the allocation of post-disaster aid, the sensitivity analyses we conducted on the amount of grant assistance available for home repairs (Fig. 9) and on the interest rate at which borrowers can finance repairs (Fig. S19) should respectively capture the potential impacts of FEMA IHP grants and SBA loans on borrower financial conditions. It is worth noting that timing and amount of post-disaster aid received from these sources is highly uncertain: less than half of applicants to FEMA's IHP program are approved, with only a tiny fraction receiving the maximum award (GAO, 2020a). These awards are typically small: between 2002 and 2024, the median (IQR) IHP grant for property owners reporting flood damage to their primary residence was only \$2,900 (\$930-\$7,510) in 2020 dollars (FEMA, 2025). SBA loans provide a larger infusion of funds but have more stringent eligibility requirements: among applicants with a DTI ratio over 45%, less than half are approved for a loan (Collier et al., 2024).

As the reviewer noted, we did not model dynamic changes in the employment status of mortgage borrowers and assumed that their income grows over time according to the annual change in percapita income of their county of residence. This assumption is conservative: were we to include income shocks due to job loss, illness, or divorce in our model, this would likely increase the number of borrowers projected to face income-related credit constraints following flood exposure. Similarly, our model framework does not account for how factors related to the 2008 global financial crisis (GFC) may have impacted the financial health and credit access of mortgage borrowers during the study period. If the elevated rate of unemployment and reduced credit supply during this period were incorporated into our model, projections of credit constraints among borrowers exposed to flooding during Hurricane Irene (which occurred in 2011) would likely be higher. It is worth noting that our property value model, which uses observed property sales as an input, reflects the effects of the GFC on home prices and (by extension) borrower equity.

We have highlighted these simplifications as limitations of our model framework in the discussion section of the manuscript while describing how the assumptions we made may have impacted our findings.

Revised text from lines 915-927: Finally, we did not explicitly model the various sources of funding for post-disaster recovery that might be available to uninsured mortgage borrowers who lack sufficient equity or liquidity to obtain private home repair loans. These include: federal sources of post-disaster aid such as SBA loans and FEMA IHP grants; alternative finance sources such as payday lenders, auto title loans, and pawnbrokers; and liquid assets such as personal savings and retirement accounts. To examine how the availability of low-interest SBA loans and FEMA IHP grants may impact our findings, we conducted scenario analyses on the interest rate at which borrowers can finance home repairs as well as the amount of home repair assistance available to those without insurance. The number of credit constrained mortgage borrowers was sensitive to the amount of grant aid available but relatively insensitive to the interest rate on home repair loans. During Hurricanes Matthew and Florence, less than a third of property

owners who applied for IHP aid were approved, and the average grant awarded was under \$5,000 (GAO, 2020b); thus, it appears unlikely that the inclusion of IHP aid would substantially alter estimates of the number of mortgage borrowers facing flood-related credit constraints. Future research could examine how these programs are likely to shape the long-term recovery outcomes of credit constrained mortgage borrowers by explicitly incorporating the timing and distribution of post-disaster aid into the integrated modeling framework.

Revised text from lines 898-904: Third, when modeling the financial conditions of residential mortgage borrowers, household income was assumed to grow over time at a rate equal to the change in average personal income for a given county and year. Data from longitudinal studies of income dynamics suggest that in reality, the rate of income growth varies depending on a household's initial wealth, and that year-to-year changes in income can be highly volatile even within a given income stratum (Fisher et al., 2016). In addition, our modeling approach does not consider exogenous income shocks arising from events such as job loss, illness, or divorce. Including these sources of variability in household income would likely increase the number of mortgage borrowers projected to experience income-related credit constraints following exposure to flooding.

Revised text from lines 905-914: Fourth, our model framework does not account for how factors related to the 2008 global financial crisis (GFC) may have impacted the financial health and credit access of mortgage borrowers during the study period. These include elevated rates of unemployment that persisted for several years following the GFC and a tightening of mortgage lending standards that reduced the availability of credit to property owners. Mortgage lending standards in the U.S. underwent a gradual loosening during the early 2000s leading up to the crisis, followed by a sharp tightening during the 2007-2009 period that led to increases in loan denial rates and more stringent LTV and DTI requirements (Vojtech et al., 2020). Of the seven flood events evaluated in this study, the effects of the GFC would be most relevant for Hurricane Irene, which occurred in 2011 when the economy was still recovering from the crisis. If the elevated rate of unemployment and reduced credit supply during this period were incorporated into our model, projections of credit constraints among borrowers exposed to flooding from Hurricane Irene would likely be higher.

**Figure 9 in the Manuscript:**

**Figure 9.** Scenario analysis on the amount of home repair grant assistance available to mortgage borrowers should they experience uninsured flood damage. The amount of available assistance is varied between zero and \$42,500—the maximum award that households can receive from FEMA's Individuals and Households Program (IHP) as of 2024 (U.S. GPO, 2023).

**Figure S19 in the Supplementary Information:**

Figure S19. Scenario analysis examining alternative assumptions regarding home repair loan interest rates, property values, and flood damage costs.

Each panel corresponds to a different interest rate scenario: (a) one in which the interest rate on home repair loans is equivalent to the prevailing "market" rate (i.e., the average 30-year fixed rate on new mortgages); and (b) one in which the interest rate on home repair loans is equal to 50% of the prevailing market rate. Within each panel, property-level estimates of flood damage and property value are perturbed by  $\pm 20\%$  to create a range of scenarios. Each box in the  $3 \times 3$  plot depicts the number of borrowers projected to face flood-related credit constraints under a given scenario, as well as the share of credit constraints attributable to various drivers (e.g., insufficient collateral, insufficient income, or both in combination).

**Specific comments:**

**Abstract**

• The writing is generally clear, but the abstract should clarify the degree to which the study is model-based or observational (or both). For instance, is "Here, we evaluate the impact" in L14 an estimated impact or an observed impact (or maybe an estimated impact with methods calibrated to observed outcomes?) This is crucial to clarify because the motivation is about highly consequential (and overlooked in the literature) outcomes, such as mortgage delinquency, default, and foreclosure. The statement I quoted makes it seem like the study observes these outcomes. However, the next line talks about conditions "indicative of default" and then reports somewhat superficial statistics.

Our reply: We have updated the abstract to clarify that our analysis is model-based, and that we are using a model to estimate the number of mortgage borrowers who experienced credit constraints following flood exposure.

Revised text from lines 14-18: In this study, we use a simulation-based approach to estimate the impact of uninsured damage on residential mortgage borrowers' financial conditions over a series of floods in North Carolina from 1996-2019. Our framework estimates key variables (e.g., damage cost, property value, mortgage balance) to project the number of flood-exposed borrowers experiencing credit constraints due to negative equity, liquidity issues, or both in combination.

Lacking insurance or income/collateral to finance repairs does not seem like a predictor of mortgage default in previous empirical studies. It is possible that I (and other readers) am unfamiliar with research showing this, so the authors should add context about the degree to which the conditions are strong indicators of default. Alternatively, if this is an observation-based study, do the authors find that is a different case in North Carolina? If so, this is a major result that the abstract should highlight more prominently.

Our reply: As described in the author's remarks, we have reframed the analysis around flood-related credit constraints and reduced the emphasis placed on mortgage default as the main outcome of interest. That said, the following empirical studies provide evidence that income and collateral constraints can lead to elevated rates of mortgage delinquency and default following natural disasters:

Billings, S. B., Gallagher, E. A., and Ricketts, L.: Let the rich be flooded: The distribution of financial aid and distress after hurricane harvey, Journal of Financial Economics, 146, 797–819, https://doi.org/10.1016/j.jfineco.2021.11.006, 2022.

Collier, B., Hartley, D., Keys, B., and Ng, J. X.: Credit When You Need It, National Bureau of Economic Research, Cambridge, MA, <a href="https://doi.org/10.3386/w32845">https://doi.org/10.3386/w32845</a>, 2024.

Du, D. and Zhao, X.: Hurricanes and Residential Mortgage Loan Performance, Office of the Comptroller of the Currency Working Papers, <a href="https://www.occ.treas.gov/publications-and-resources/publications/economics/working-papers-banking-perf-reg/pub-econ-working-paper-hurricanes-residential-mort-loan-perf.pdf">https://www.occ.treas.gov/publications-and-resources/publications/economics/working-papers-banking-perf-reg/pub-econ-working-paper-hurricanes-residential-mort-loan-perf.pdf</a>, 2020.

Billings et al. (2022) observed elevated rates of personal bankruptcy and credit delinquency among "low ability-to-repay" mortgage borrowers living in areas outside the SFHA that were affected by flooding during Hurricane Harvey. These borrowers were unlikely to have flood insurance and were also likely to have trouble accessing SBA loans due to their income and credit score. This finding highlights how lacking sufficient income to access home repair loans can increase the risk of adverse credit outcomes among those who lack flood insurance.

Collier et al. (2024) were able to estimate the causal effect of access to low-interest forms of credit on recovery using a regression discontinuity design that compared SBA loan applicants with a DTI ratio just above and just below a 40% threshold. Their estimates indicate that those who qualified for an SBA loan due to the discontinuity were far less likely to experience negative financial outcomes such as bankruptcy and mortgage delinquency in the years following a disaster, further underscoring the importance of income-related credit constraints on recovery outcomes.

Du and Zhao (2020) found that increases in 180-day delinquency rates following Hurricanes Harvey and Maria could be partly explained by changes in current loan-to-value (LTV) ratio triggered by property damage. The authors in this study also observed that default rates increased monotonically and nonlinearly with LTV. Although the authors in this study do not distinguish between flood and wind damage, these findings highlight how low initial levels of home equity (which can be used as collateral for further borrowing) can increase the risk of default following natural disasters.

It is important to note that the recovery outcomes of credit constrained borrowers are uncertain and depend on access to sources of funding not included in our model framework such as personal savings, post-disaster aid, and support from family and friends. With this in mind, we've included the following text in the manuscript to describe the evidence linking post-flood credit constraints to negative financial outcomes while acknowledging these sources of uncertainty.

Revised text from lines 121-127: In an analysis of SBA loan applications from the 2005-2013 period, Collier et al. (2024) observed a sharp decrease in the probability of loan approval for applicants with a DTI ratio exceeding 40%: those with a DTI ratio just over this threshold were much less likely to receive a loan than those with a DTI ratio just

under it (60% vs. 80% approved), with approval rates dropping below 50% for applicants with a DTI ratio of 45% or greater. Using a regression discontinuity design, the authors of this study were able to estimate the causal effect of access to low-interest forms of credit on recovery, finding that those who qualified for an SBA loan were far less likely to experience negative financial outcomes such as bankruptcy and mortgage delinquency in the years following a disaster.

Revised text from lines 184-188: Similarly, Billings et al. (2022) documented higher rates of bankruptcy and credit delinquency among Harvey-affected households outside the SFHA. This study found that post-Harvey increases in bankruptcy were largely concentrated in a specific segment of the population: mortgage borrowers located outside the SFHA with below-median incomes and credit scores. Property owners in this group faced high levels of uninsured damage but had limited ability to finance repairs through additional borrowing.

Revised text from lines 571-576: It is important to note the ACLTV and ADTI thresholds employed in this framework are assumed to be necessary (but not sufficient) conditions for financial distress; as such, the credit constraint estimates generated by our procedure reflect the share of flood-exposed borrowers who may be forced to rely on other (less reliable) sources of funding for recovery such as savings, post-disaster aid, and support from family and friends. Additional information linking the post-flood financial conditions of mortgage borrowers to the probability of bankruptcy and default could be used to translate the estimates generated by our approach into projections of lender credit losses (Bellini, 2019).

• Some ambiguous grammar. For example, do the authors mean in L16-17 that they look at default and negative equity, or does negative equity refer to one of the financial conditions indicative of default? The ", including" grammar makes the remaining text unclear.

Our reply: To remove any ambiguity, we have revised the abstract to make it clear that negative equity is a financial condition that can cause a borrower to be credit constrained.

Revised text from lines 16-18: Our framework estimates key variables (e.g., damage cost, property value, mortgage balance) to project the number of flood-exposed borrowers experiencing credit constraints due to negative equity, liquidity issues, or both in combination.

**Introduction**

• The details are strong and the authors are clearly knowledgeable about this topic, but long paragraphs and sentences make the narrative difficult to follow. Claims like "Given these gaps in existing knowledge" (L131) are ineffective because the authors do not clearly signal knowledge gaps in preceding paragraphs.

Our reply: Thank you for identifying these issues with our introduction. In the revised manuscript, we have streamlined the narrative and explicitly stated the knowledge gaps our study seeks to address.

Revised text from lines 47-56: While several empirical studies offer evidence that less insured and less creditworthy households exhibit higher levels of financial distress following disasters (Billings et al., 2022; Collier et al., 2024; You and Kousky, 2024), few studies have attempted to quantify the prevalence and underlying drivers of credit constraints among flood-exposed property owners. Data limitations may be a contributing factor to these knowledge gaps: understanding whether a property owner has sufficient borrowing capacity to fund their recovery requires granular data on their income, debt obligations, property value, and level of uninsured damage exposure—information which is rarely captured by a single comprehensive dataset. In this context, simulation-based modeling approaches can help to address data scarcity issues by integrating data from multiple heterogeneous sources and explicitly representing the processes that give rise to post-flood credit constraints, allowing researchers to estimate the financial impacts of flood events in settings where direct observations are unavailable.

Some of the introduction texts reads like a literature review, which disrupts the flow. It might be helpful to break out a succinct introduction that introduces the paper's focus and contributions, and a separate literature review section. Given that the abstract's focus on default, the paragraph starting on L81 is the most relevant, but we only hear about default after three long preceding paragraphs.

Our reply: Thank you for this helpful suggestion. In the revised manuscript, we have included a succinct "introduction" section that describes the focus of our study and a separate "background" section that reviews the relevant literature.

• The text on flood insurance and limited coverage are important, but I wonder if they would be more effective after going straight into the main text about financial instability. There is a lot of literature about natural disasters and financial instability – why don't the authors just start on their focus? The issues with the NFIP and flood-risk information are

important contributors to these issues since a lack of insurance could be a major driver of bad financial outcomes for households, such as default.

Our reply: In the revised manuscript, we have moved the text describing the drivers of the flood insurance gap to the background section that now comes after the introduction.

• I don't know that the references in L77 are appropriate. It seems like the point here is that damaged properties are worth less on the market and home equity loan principal will be lower. I'm not sure that the cited studies about post-flood property prices are relevant here because these studies do not control for flood damages. Thus, it is unclear if the change in property prices after floods are due to a damage effect (or similar), and thus concentrated within segments of the housing market (which would be a form of double counting relative to the rest of this paragraph – if a property is structurally comprised, it is worth less and you can't take out a home improvement loan of the same value as preflood conditions). There are some studies that control for flood damage and find changes in post-flood prices unrelated to damage. The authors could cite Atreya and Ferreira (2015) - https://doi.org/10.1111/risa.12307 - but this is a small case study and there is conflicting evidence. See Davlasheridze and Fan (2019) https://doi.org/10.1007/s41885-019-00045-z- or Pollack and Kaufmann (2022) https://doi.org/10.1016/J.ECOLECON.2022.107350 for evidence on how and why property price changes may not be market-wide after a flood event and specifically indicate double counting in this context.

Our reply: We agree that failing to control for flood damages could lead to biased estimates of the "information effect" of recent flooding on property prices in several of the cited studies. In the revised manuscript, we have limited our citations to the following studies where we believe the potential for bias is likely to be low:

Ortega, F. and Taṣpınar, S.: Rising sea levels and sinking property values: Hurricane Sandy and New York's housing market, Journal of Urban Economics, 106, 81–100, https://doi.org/10.1016/j.jue.2018.06.005, 2018.

Fang, L., Li, L., and Yavas, A.: The Impact of Distant Hurricane on Local Housing Markets, J Real Estate Finan Econ, 66, 327–372, https://doi.org/10.1007/s11146-021-09843-3, 2023.

Bin, O. and Landry, C. E.: Changes in implicit flood risk premiums: Empirical evidence from the housing market, Journal of Environmental Economics and Management, 65, 361–376, https://doi.org/10.1016/j.jeem.2012.12.002, 2013.

Ortega and Taṣpınar (2018) analyzed the effects of Hurricane Sandy on property prices in New York City using a parcel-level dataset that includes information on the severity of flooding at each property location. They found that damaged properties experienced a large and immediate decrease in value (17-22%) following Sandy, while non-damaged properties located inside SFHAs experienced a more modest (8%) but still significant decrease in value that persisted for many years following the event.

Fang et al. (2023) analyzed the effects of a large-scale but distant hurricane on property prices in Miami-Dade County, Florida. They found that prior to Hurricane Sandy, properties located in SFHAs demanded a price premium of approximately 3.5%, which they attribute to water-related amenities. This price premium declined significantly following Sandy, despite the hurricane causing no direct damage within the study area; this effect was temporary and lasted for less than a year. The authors attribute these trends to increased awareness of flood risk among homebuyers due to media coverage of Hurricane Sandy.

Bin and Landry (2013) use property sales data before and after Hurricanes Fran and Floyd to measure the effect of these events on property values within flood zones in Pitt County, North Carolina. They observe that significant price discounts emerge after a major flood event, equivalent to a 5.7% decrease after Hurricane Fran and an 8.8% decrease after Hurricane Floyd. Although this study did not control for property-level flood damage, structures that suffered considerable damage during these events accounted for a small percentage of all flood zone properties in Pitt County, suggesting that the amount of bias this introduces into the results should be low.

We have also referenced Atreya and Ferreira (2015) to highlight that disentangling the effects of direct damage from changes in market perceptions of risk remains an ongoing challenge in many studies.

Revised text from lines 143-145: Prior studies suggest that flood events can depress property values in affected areas (Bin and Landry, 2013; Fang et al., 2023; Ortega and Taṣpınar, 2018), though disentangling the direct effects of flood damage from changes in market perceptions of risk can often be a challenge (Atreya and Ferreira, 2015).

• There is something difficult about the logic of the key outcome of focus, default. The authors talk about how most of the damage from several of the largest storms in US history was uninsured, and they also talk about a lack of insurance as a major indicator of mortgage default. So, where is the evidence from these storms that many uninsured households defaulted? The references to literature on this central framing point, starting on L81, would benefit substantially from reporting statistics from the papers to help the

reader understand the connection between uninsured damage and default. The most specific statistic in this paragraph is the "50 times higher" one from Kousky, but this is on 90-day delinquency. While an important negative financial outcome, the current study primarily frames a focus on default (e.g., L16-17 of the abstract: "Our framework estimates key financial variables to identify borrowers exhibiting financial conditions of default"). The evidence on that from Calabrese et al., (2024) should state what the quantitative evidence is so readers understand how large the effect is.

o I know the Kousky paper better than the Calabrese one, so I will focus on evidence in there that I believe the authors should pay more attention to in their framing, since they focus on mortgage default as the key outcome. Table 7 of that study seems to suggest that moderate to severe damage is the main predictor of 180 or more days delinquent or default. While the result for moderate to severe damage X in SFHA is not significant, that may be due to the very small sample size of treatment observations. The authors should highlight that damage amount is an important factor and can point out that the study doesn't control for whether a property is insured (which supports the authors' claims either way). However, the study combines 180 or more days delinquency and default into one outcome because the number of defaulted loans is not large enough for identifying a statistical effect. Out of 27,000 loans, there were only 24 defaults as of August 2019 (2 years after the storm). Overall, there are a rather small proportion of observations in the combined outcome. The authors should be transparent that there is not strong empirical evidence on the causal mechanisms their claims rely on.

Our reply: We have reframed the analysis around flood-related credit constraints and reduced the emphasis placed on mortgage default as the main outcome of interest. Nevertheless, we still believe it is valuable to highlight the potential linkages between uninsured damage, credit constraints, and negative financial outcomes such as personal bankruptcy and mortgage delinquency. In the revised manuscript, we reference Kousky et al. (2020) as evidence of the relationship between uninsured flood damage and mortgage delinquency, and reference Billings et al. (2022) as evidence that these effects are heterogenous for households with differing access to affordable credit. We have revised the text to make it clear that the outcome examined in the Kousky et al. (2020) is the event of a borrower becoming 180 or more days delinquent (as opposed to the much rarer outcome of default).

Revised text from lines 178-190: Numerous studies have linked floods and hurricanes to higher rates of mortgage delinquency (Calabrese et al., 2024; Du and Zhao, 2020; Kousky et al., 2020; Mota and Palim, 2024; Rossi, 2021) and personal bankruptcy (Billings et al., 2022; Collier et al., 2024), with effects varying based on households' access to insurance and affordable credit. After Hurricane Harvey, Kousky et al. (2020)

found that mortgaged properties with moderate to severe flood damage had over double the odds of becoming 180 or more days delinquent than undamaged properties—a relationship significant only outside the SFHA, where insurance uptake is low. Similarly, Billings et al. (2022) documented higher rates of bankruptcy and credit delinquency among Harvey-affected households outside the SFHA. This study found that post-Harvey increases in bankruptcy were largely concentrated in a specific segment of the population: mortgage borrowers located outside the SFHA with below-median incomes and credit scores. Property owners in this group faced high levels of uninsured damage but had limited ability to finance repairs through additional borrowing. Collectively, these studies suggest that uninsured property owners experience lasting financial consequences from flooding, particularly when income or collateral constraints prevent them from accessing low-cost forms of debt financing.

- I like the Thomson paper a lot, but that study is fully model-based and assumes the causal mechanism of a house being underwater leading to default. As this is the main paragraph on empirical evidence relating uninsured damage and financial preconditions to the main outcome of interest, default, the authors should also review the following related empirical literature (mostly flood, but also related to wildfire risk). These studies generally support the work cited in this study, but some offer surprising insights into household financial resilience in the wake of large natural disasters that this study should reconcile in its framing:
  - Biswas, S., Hossain, M., & Zink, D. (2023). California Wildfires, Property Damage, and Mortgage Repayment. Federal Reserve Board of Philadelphia Working Paper, 23-5.
  - Mota, N., & Palim, M. (2024). Mortgage Performance and Home Sales for Damaged Homes Following Hurricane Harvey. Fannie Mae Working Paper Series.
  - Del Valle, A., Scharlemann, T., & Shore, S. (2024). Household financial decisionmaking after natural disasters: Evidence from Hurricane Harvey. Journal of Financial and Quantitative Analysis, 1-27.
  - Hopkins, C., Marr, A., & Wilson, N. (2024). How Does Mortgage Performance Vary Across Borrower Demographics Following a Hurricane? Federal Housing Finance Agency Working Paper Series.
  - Issler, P., Stanton, R., Vergara-Alert, C., & Wallace, N. (2020). Mortgage markets with climate-change risk: Evidence from wildfires in california. Available at SSRN 3511843.
  - o Rossi, C. V. (2021). Assessing the impact of hurricane frequency and intensity on mortgage delinquency. Journal of Risk Management in Financial Institutions, 14(4), 426-442.

- Gallagher, Justin, and Daniel Hartley. 2017. "Household Finance after a Natural Disaster: The Case of Hurricane Katrina." American Economic Journal: Economic Policy 9 (3): 199–228.
- Deryugina, Tatyana, Laura Kawano, and Steven Levitt. 2018. "The Economic Impact of Hurricane Katrina on Its Victims: Evidence from Individual Tax Returns." American Economic Journal: Applied Economics 10 (2): 202–33.
- Deryugina, Tatyana. 2017. "The Fiscal Cost of Hurricanes: Disaster Aid versus Social Insurance." American Economic Journal: Economic Policy 9 (3): 168–98.

Our reply: We thank the reviewer for sharing these helpful studies. In the revised manuscript, we have included references to the papers by Mota and Palim (2024), Del Valle et al. (2022), Rossi (2021), Gallagher and Hartley (2017), and Deryugina (2017; 2018).

In particular, Deryugina (2017) and Deryugina et al., (2018) discuss drivers of financial outcomes after natural disasters that are largely missing from the present study's framing. The first study, particularly important for the authors to engage with, investigates the role of non-disaster-based social insurance that can actually improve some households' wellbeing after disasters. The second examines the role of employment and income, highlighting the role of savings in supplementing households in the aftermath of Katrina (which seems relevant especially here because households outside the SFHA may have lower probability of flooding, so may have higher savings if they don't pay into the insurance program over time; this may not be the case with NFIP because of its riskrating procedure before Risk Rating 2.0, but seems plausible and is worth mentioning).

Our reply: Thank you for sharing these two studies. We have incorporated both into the revised manuscript.

Revised text from lines 168-170: Households might also use retirement accounts to fund repairs when other savings prove insufficient: Deryugina et al. (2018) observed a large increase in withdrawals from retirement accounts among New Orleans residents following Hurricane Katrina.

Revised text from lines 175-177: While property owners may be able to supplement their savings with financial support from family and friends (You and Kousky, 2024) and social safety net transfers (Deryugina, 2017), it is unclear whether these sources provide sufficient funds to meet the recovery needs of those with severe damage to their residence.

• The paragraph on L119 seems too repetitive with previous paragraphs. Can the authors consolidate the presentation of each topic?

Our reply: In the revised manuscript, we have streamlined the introduction and background sections to avoid unnecessary repetition.

The "gaps in existing knowledge" (L131) are unclear, and it's not clear the authors address them. It seems like the gap (as far as I can tell, the authors state only one knowledge gap) is "Although prior studies such as those by Kousky et al. (2020) and Calabrese et al. (2024) have examined the association between insurance uptake, flood exposure, and mortgage credit risk, there exists a need for additional research into how the pre-flood financial conditions of a borrower (i.e., equity and liquidity) affect the relationship between uninsured damage exposure and the post-flood risk of mortgage default" (L127-130). However, the objectives of the study do not appear to reconcile the gap. While the authors describe a very impressive analytical workflow for linking flood damage to preconditions, the link between the preconditions and outcome of interest, default, is not addressed in this study. But the authors explicitly claim that there is additional research into how pre-flood financial conditions affect post-flood risk of mortgage default. The last paragraph reads as if the authors address something more like post-flood exposure to mortgage default – they link flood damage and pre-flood financial conditions to a post-flood financial state, but do not appear to build evidence on the link between those financial states and mortgage outcomes.

Our reply: Thank you for identifying this issue with our original framing of the analysis. Given that our study does not directly test the associations between model projections of post-flood financial states (e.g., negative equity) and observed mortgage outcomes, we have reframed the analysis around flood-related credit constraints and reduced the emphasis placed on mortgage default as the main outcome of interest. Because the capacity of uninsured property owners to take on additional debt is a direct function of their post-flood financial conditions, we believe that this more modest framing aligns our study objectives with our methodological approach.

• The Introduction needs to streamline the narrative around the research gaps and what the study focuses on. The current structure does not flow well because it is unfocused. It reads as if the authors developed the analytical workflow and then backed out the research gap the workflow could address, but did not identify a clear science question.

Our reply: In the revised manuscript, we have streamlined the introduction section and explicitly state the knowledge gaps and scientific questions our study aims to address.

Revised text from lines 47-56: While several empirical studies offer evidence that less insured and less creditworthy households exhibit higher levels of financial distress following disasters (Billings et al., 2022; Collier et al., 2024; You and Kousky, 2024), few studies have attempted to quantify the prevalence and underlying drivers of credit constraints among flood-exposed property owners. Data limitations may be a contributing factor to these knowledge gaps: understanding whether a property owner has sufficient borrowing capacity to fund their recovery requires granular data on their income, debt obligations, property value, and level of uninsured damage exposure—information which is rarely captured by a single comprehensive dataset. In this context, simulation-based modeling approaches can help to address data scarcity issues by integrating data from multiple heterogeneous sources and explicitly representing the processes that give rise to post-flood credit constraints, allowing researchers to estimate the financial impacts of flood events in settings where direct observations are unavailable.

Revised text from lines 57-61: In this study, we use an integrated modeling approach to simulate the impact of flood-related property damage on residential mortgage borrowers' financial conditions over a series of floods in North Carolina from 1996-2019 while examining the following research questions: (1) How much of the damage from these events was uninsured? (2) What share of flood-exposed borrowers faced credit constraints that were likely to impair their ability to access home repair loans? (3) Were these credit constraints driven by insufficient income, insufficient collateral, or both in combination?

• I think the Introduction needs to soften its claims about what the framework can achieve. The authors need to be more explicit and transparent about what the framework does (from what I have read so far, it is an advanced framework to estimate exposure to bad mortgage outcomes but does not estimate the risk of those outcomes).

Our reply: Thank you for bringing this issue to our attention—we agree that our original introduction was unclear about what our model framework can accomplish. We have made substantial revisions to the introduction to address this problem and take care to explicitly state that our study uses a simulation-based approach to estimate post-flood credit constraints.

Revised text from lines 14-18: In this study, we use a simulation-based approach to estimate the impact of uninsured damage on residential mortgage borrowers' financial conditions over a series of floods in North Carolina from 1996-2019. Our framework estimates key variables (e.g., damage cost, property value, mortgage balance) to project the number of flood-exposed borrowers experiencing credit constraints due to negative equity, liquidity issues, or both in combination.

It is now clear that the study is fully model based, not even using data on financial outcomes to calibrate the projections of post-flood financial conditions or vet its performance. This is concerning given the current framing of the study. The main framing of the Intro is "there exists a need for additional research into how the pre-flood financial conditions of a borrower (i.e., equity and liquidity) affect the relationship between uninsured damage exposure and the post-flood risk of mortgage default" (L129-130)." How does this study address this need if there is no data on mortgage default outcomes? At this point, I see a few options to reconcile this. First, the authors might instead frame their study as estimating post-flood exposure to mortgage default. This seems defensible because in the typical risk framing, risk is the potential for adverse consequences, driven by interactions of hazards, exposure, and vulnerability. The workflow the authors describe does not appear to fit this definition. Second, the authors could use the three introduced theories on default causal pathways as their representations of vulnerability. This would enable them to take their exposure estimates and translate them into default outcomes, conditioned on the assumption that one of the theories represents a valid causal pathway. The authors raise nice evidence that these theories have weaknesses, so they would have to be careful in their framing if they take this approach. Third, and complementary to the second, it seems like the authors could take a more exploratory approach by embracing the uncertainty in the system and investigating how different assumptions (e.g., about causal pathways of default) and model uncertainties (e.g., both well-characterized uncertainty around damage projections and deeper structural uncertainties about the integrated modeling chain) propagate into projections of default risk. I have a preference for the third option because it seems the most appropriate for the question stated on L129-130. Given the model-based approach of the study, with no observations on the outcome of interest, an exploratory modeling approach that identifies drivers of uncertainty in the outcome of interest appears the best way to investigate the relationship between uninsured damage exposure and the post-flood risk of mortgage default. The authors could also clarify their current framing in a revision and explain where I misunderstand a gap between their science question and methods.

Our reply: Thank you for providing these helpful suggestions for how to address the issues with the original framing of our study. We have chosen to pursue the first option by reframing our analysis around estimating post-flood credit constraints and reducing the emphasis placed on the endpoint of mortgage default. As the reviewer noted, our model framework estimates the degree to which uninsured flood damage increased the potential for financial states (e.g., negative equity and cashflow problems) that can impair a mortgage borrower's ability to fund repairs to their property through additional borrowing; however, given the lack of data on mortgage performance, the degree to which these states are predictive of default risk remains unclear. We believe that this

more modest framing aligns our study objectives and research questions more closely with our methodological approach. For more information on how this reframing was implemented in the revised manuscript, we refer the reviewer to the author's remarks on the first page of this document.

**Methods**

• The first sentence of this section seems like a more modest and appropriate framing than what the Introduction suggests. It also seems to support the need for an exploratory modeling and sensitivity analysis approach. The authors should consider streamlining the Introduction around this.

Our reply: In the revised manuscript, we have streamlined the introduction and have added a variance-based sensitivity analysis that utilizes the method of Sobol' to evaluate how uncertainty in key model parameters contribute to variance in borrower-level financial outcomes. For details regarding this sensitivity analysis, we refer the reviewer to our response to their "general comment" #4.

• Why only use loan-level data for initial financial conditions? If there are new originated loans over the full time period, isn't it possible to identify changes in financial conditions as well? I'm not familiar with the HMDA data. Using only data for initialization leads to very strong assumptions about income and loans over a 23 year period that again seems to support a framing around exploratory modeling and sensitivity analysis.

Our reply: To clarify, loan-level data is used to parameterize both the origination characteristics and repayment profiles of mortgage borrowers; however, certain variables (e.g., income, property value) are only observed in this data at the time of origination. As such, the amount of uncertainty in estimates of these quantities is likely to grow over time, though it is worth noting that most loans during the study period are repaid within 10 years (often as a result of a homeowner selling or refinancing their property). In the base case, we conservatively assume that borrower income grows steadily over time according to the annual growth in per-capita income of their county of residence. In the revised manuscript, we have added a sensitivity analysis in which borrower income evolves stochastically over time according to geometric Brownian motion (GBM). The results of this sensitivity analysis suggest that uncertainty in borrower income growth has a smaller impact on our model projections than other sources of uncertainty such as estimates of property value and flood damage costs. For details regarding this sensitivity analysis, we refer the reviewer to our response to their "general comment" #4.

I'm confused by the claim that strategic, cashflow, and double-trigger mechanisms are types of defaults (L146-149). The Introduction text specifically frames these as theories of default and talks about limitations in the first two theories. Why are all three theories modeled then? I think it would be helpful to reframe the paper to accommodate the methods. It seems like a worthwhile exercise to map the exposure to three types of default mechanisms, as long as the authors provide more context in the Introduction for why. One reason could be that although there is very insightful empirical research on drivers of default in the context of natural disasters, we don't have a complete understanding of the causal mechanisms (data limitations, a limited number of events, etc.,). There are competing theories about these causal mechanisms, which we can represent with bottom-up models (if taking this approach, please provide evidence that these theories are prominent and inform decisions – which is necessary to support some claims from the Introduction). I think it would be easy to frame the contribution in this way, and helps the authors explain that as more research comes out on the causal mechanisms, their framework could adapt to those pathways and better model mortgage default risk.

Our reply: We believe that the reframing of the analysis around flood-related credit constraints (as opposed to mortgage default) addresses this comment. For further details on how this reframing was implemented in the revised manuscript, we refer the reviewer to the author's remarks on page one of this document.

• I don't understand why the authors simulate financial conditions at a monthly time step over the 1996-2019 period but only focus on the 7 largest (when they point out that the state faced 14 major disaster declarations over the period). I greatly appreciate the transparency on this point. Can the authors please justify this modeling choice and explain its potential implications on their results? Why can't the authors simulate just at the storm time steps? What are the consequences of overlooking other major storms (in addition to other flooding events and possibly more important events that affect the outcome of interest such as the great financial recession of '08)?

Our reply: We focused on the largest seven events (in terms of associated NFIP claims) during the study period to ensure that the training data for our machine-learning-based flood damage model (model I) includes a sufficient number of inundated properties as training examples. As discussed in the paper by Garcia et al. (2025), which utilizes the same input data as this study, our training data will be imbalanced for smaller events, with many more non-inundated than inundated training examples. High levels of class imbalance can compromise the performance of standard machine learning algorithms (Haixiang et al., 2017); as such, we decided in the early stages of the project to limit our focus to the seven largest events, which accounted for a majority (53%) of all NFIP

claims filed in North Carolina during the study period. The paper by Garcia et al. (2025) examines additional floods not included in our study and does a good job of evaluating how the performance of machine-learning-based flood damage models varies by event size; however, given our focus on borrower financial conditions, we were mainly interested in capturing the larger events.

The main reason why we simulated borrower finances at a monthly timestep was because we were interested in capturing the cumulative effects of multiple flood exposures on borrower financial conditions. Updating the financial variables of borrowers each month over the life of their mortgage provided a natural way in which to model how the financial impacts (e.g., repair loan repayment) from one event extend through the occurrence of the next. Although it may be possible to simulate only at storm timesteps, we chose a monthly structure because it is more intuitive and facilitates the reuse and expansion of model components in future analyses.

• The stochastic sampling for certain variables (mortgage loan characteristics – what else?) again suggests the value of an exploratory modeling approach and sensitivity analysis. Is it just one draw from the tract distribution for each household? One draw could lead to spurious projections given the distribution of other model inputs that are correlated with the financial variables (but not sufficiently sampled with one draw).

Our reply: We have added the following text to the manuscript describing the number of samples drawn for stochastically generated variables. In essence, our simulation approach generates ten independent realizations of borrower outcomes. While we agree that this number of replicates could lead to spurious projections for small geographic units (e.g., census tracts) we found this number to be sufficient for generating stable estimates of the number of borrowers facing flood-related credit constraints across the study area. This can largely be attributed to the large number of mortgages (4.7 million) that are simulated in each model run. For clarity, we have added the following text to the revised manuscript:

Revised text from lines 580-586: For each simulation run, we simulate the financial conditions of 4.7 million borrowers with single-family mortgages originated during the 1992-2019 period at a monthly timestep over the life of their loan. Because certain variables describing the initial financial conditions and repayment profiles (model III) of mortgage borrowers are stochastically generated, model projections of flood-related credit constraints were averaged over ten simulation runs conducted with different random seeds. This number of replicates was found to be sufficient for achieving stable estimates of the number of borrowers facing flood-related credit constraints across the

study area; however, generating stable estimates for smaller geographic units (e.g., specific census tracts) would likely require additional simulation runs.

In general, the methods and text on cross validation are great and the authors are exceptional related to previous literature. Great job! However, since the point of this study appears to be about the integrated modeling approach to modeling financial stability, it's a first-order concern to investigate how sensitive the modeling framework is to uncertainties in the modeling steps and inputs. The validation does suggest sizable uncertainty in both interpolation & extrapolation, which begs the question: how much does this matter for projections of mortgage default? While I recognize this paper builds on methods under review elsewhere, it would be helpful to contextualize how the sensitivities in the underlying methods are particularly relevant with respect to this study's prediction goals. I think that given the interest in the connection between uninsured losses and mortgage defaults, the most relevant sensitivity is the degree to which the model may overestimate exposure and damage outside of the training data. The authors do a good job of talking about these uncertainties and how well their model does. But the authors probably recognize that there is something different about homes that don't have flood insurance than the homes that do (given the authors' framing around affordability and willingness to pay for insurance and their reference to Bradt et al., 2021, I think they recognize that there are different factors between these populations, including that houses facing higher hazard are more likely to purchase insurance even if they are outside the SFHA). There is also a selection of properties into the claims data, based on factors such as income, deductible, and loss size. I reviewed the Garcia et al., (2025) preprint and sections 4.2 and 4.3 in detail and it's hard to see how their sensitivity checks get at these concerns of selection. I think it's important for the authors, given their current framing and seeming data limitations on "validation" data for default, to incorporate uncertainty in the modeling steps and evaluate how those propagate into sensitivity in the mortgage default projections. I want to emphasize that the cross-validation approach is rigorous, and the authors did a great job writing about it, but in the context of this highly complex modeling workflow, it seems very important to evaluate how the uncertainties in each model component propagate. Again, this especially seems the case because the integrated modeling approach seems to be the study's main contribution (if this is not the case, the study needs to substantially reframe its title and introduction).

Our reply: We recognize that using a machine learning model trained on insurance data to estimate flood damage exposure creates the potential for selection bias due to differences between insured and uninsured households. However, we believe our model projections of flood damage are a conservative estimate of the true level of flood exposure for the following reasons. First, our model exhibited consistently high specificity for insured households both inside and outside the SFHA (Fig. S2). While

insured households outside the SFHA are an imperfect proxy for uninsured households (for whom we lack data), the fact that we do not observe a drop off in specificity when moving from higher-risk to lower-risk flood zones suggests that our model correctly classifies most non-flooded properties even in areas where the density of training data is low. Second, false negatives accounted for a much greater share of cross-validation errors made by our model than false positives (Table S4), suggesting that our model tends to underpredict flood damage exposure; this finding was consistent both inside and outside the SFHA. Finally, our model estimates that 66% of flood-related losses from the seven evaluated events were uninsured (Fig. 3), which is roughly in line with the uninsured share (70%) of expected annual U.S. flood losses estimated by Amornsiripanitch et al. (2025) and with the ratio of insured to uninsured damages estimated by catastrophe modeling firms for major past flood events such as Hurricanes Helene, Florence, Irma, and Harvey (CoreLogic, 2024; Reuters, 2017a, b; RMS, 2018).

We have added the following text to the manuscript to highlight potential sources of bias in our estimates of flood damage exposure. In addition, we have also added a variance-based sensitivity analysis examining the influence of uncertain model parameters (including damage costs) on key outcomes of interest. For details regarding this sensitivity analysis, please see our response to "general comment" #4.

Revised text from lines 877-887: The results of this analysis should be interpreted in the context of several limitations. First, we used a machine learning model trained on insurance policies and claims data to estimate flood damage exposure within the study area, which creates the potential for selection bias due to differences between insured and uninsured households. For example, properties in high-risk flood zones are overrepresented in our training data due to regulations requiring property owners with federally-backed mortgages to purchase flood insurance if their property is located inside the SFHA (GAO, 2021). In addition, higher-income households may also be overrepresented in our training data due to the positive association between wealth and flood insurance uptake (Atreya et al., 2015; Kousky, 2011). Although it is difficult to predict how these biases may influence our projections of flood damage, cross-validation results suggest that model performance was similar for insured properties inside and outside the SFHA (Fig. S2). While insured properties located outside the SFHA are an imperfect proxy for uninsured households (for whom we lack data), this group provides insight into how our model is likely to perform in areas that were underrepresented in the training data.

Revised text from lines 324-333: When identifying damaged properties, the model exhibited high accuracy ( $\geq$ 92%) and specificity ( $\geq$ 98%) but low sensitivity, with true positive rates of between 12% and 42% across events. This behavior is characteristic of

machine learning classifiers trained on class imbalanced data where the positive class (e.g., presence of flood damage) is rare compared to the negative class (Haixiang et al., 2017; He and Cheng, 2021). Among properties that were misclassified by our model in cross-validation, false positive and false negative predictions respectively accounted for 12% and 88% of model errors across the seven evaluated events (Table S4). Collectively, these results suggest that our model often fails to detect properties that were damaged, which is likely to lead to a systematic underestimation of the true level of flood exposure within the study area. As such, our projections of flood damage exposure (and, by extension, flood-related credit constraints) should be interpreted as a conservative bound as opposed to a central estimate.

• I don't see how the neighborhood-level cross validation results for the damage model are relevant if the outcome of interest gets simulated at the property level. The relevant validation metrics are at the property level, where errors can interact with other uncertain inputs and could propagate into errors in the default projections. The low R2 is not surprising given previous research but is concerning and it would be very insightful to see how uncertainty in damage at the property-level propagates (especially as it interacts with other uncertain inputs). The later sensitivity analysis that appears to uniformly lower and increase property-level damages by 20% does not justify if 20% is enough. The Wing et al., (2020) study that the authors cite suggest that for most inundation depth, damage at the property-level can vary from 0% to 100% of a structure's value, and that the damage is heteroskedastic with inundation. This suggests that a uniform 20% adjustment is not sufficient for sampling uncertainty and seeing how it propagates.

Our reply: We have added a variance-based sensitivity analysis examining the influence of uncertain model parameters (including damage costs) on key outcomes of interest. For details regarding this sensitivity analysis, please see our response to "general comment" #4.

• I would like to know more about the hedonic model. In particular, can the authors provide more information about the ATTOM dataset? Are the coordinates tax parcels or building footprints? Are the records complete across the state in all observable characteristics? For more context on why it's important to provide more details about this dataset, please see Nolte, Christoph, et al. "Data practices for studying the impacts of environmental amenities and hazards with nationwide property data." Land Economics 100.1 (2024): 200-221.

Our reply: We have included the following text in the revised manuscript to provide additional details regarding ATTOM dataset, the process used to geolocate property sale transactions, and methods for dealing with missing property attributes.

Revised text from lines 358-366: The time-varying market value of each property included in the analysis is estimated across the study period on a quarterly basis using a dataset of residential real estate sales acquired from ATTOM Data Solutions (ATTOM, 2021). This dataset includes 2.3 million property transactions from North Carolina during the 1990-2019 period, and contains information on the property location, sale price, and date on which the transaction occurred. Property transactions were geolocated to building footprints via a two-step process: (1) transactions were first spatially joined to parcels based on the reported latitude and longitude in the ATTOM dataset, and (2) each transaction's location was then refined to correspond to the largest building footprint within the associated parcel. The parcel and building datasets used in this process were the same as those described in Section 3.3. After discarding transactions that were not from single-family detached homes or which had missing data, the final dataset consisted of 1.8 million geolocated property sales.

Revised text from lines 263-270: The location and structural characteristics (e.g., foundation type, first floor elevation) of each individual property are specified using a statewide building inventory complied by NCEM's GIS team (NCEM, 2022) that represents an approximate snapshot of the building stock during the middle of the study period. This database includes information on occupancy classifications that allow for a distinction between various types of residential and commercial structures. This database is spatially joined to a statewide parcels dataset that delineates the boundaries of individual properties (NC OneMap, 2022). For properties with multiple structures (e.g., a main building and an outbuilding), property characteristics are evaluated based on the structure with the largest aerial footprint. For properties missing data on key attributes (e.g., year built) missing values were spatially imputed using nearest-neighbor interpolation.

How does the property value model account for the existence of flood exposure, events, and damage over the study period? Is there any heterogeneity in property valuation based on exposure, structural defense, or damage? On that note, does the damage model have input features related to structural defense? In addition to the overall sensitivity of the model, it is important to consider sensitivity of the default projections based on heterogeneity in inputs. The results aggregate on geographic and economic factors that may be relevant to flood resilience policy – what about accounting for those factors in modeling the outcome?

Our reply: We did not include predictors related to structural defense or past flood damage exposure in the property valuation model (model II). The damage model (model I) contains certain predictors that may serve as proxies for structural defense, including:

first floor elevation, foundation type, year built, and SFHA status (Table S1). We have added the following text to the manuscript to highlight how future work could potentially improve upon our methods by incorporating these factors into the property valuation model.

Revised text from lines 411-414: Future work could potentially enhance the performance of the property valuation model by introducing filters to identify armslength sales and by adding predictors that capture property-specific attributes related to structural defense and prior flood exposure (Nolte et al., 2024; Pollack and Kaufmann, 2022).

How many samples for each property from the HMDA data? How do the authors account
for correlation in income and factors such as property value when sampling from the
HMDA data? Accounting for the correlation structure in the HMDA data is great, but it's
crucial how those draws are assigned to properties with different hazard, exposure, and
vulnerability characteristics.

Our reply: Aggregate projections of flood-related credit constraints were averaged across ten simulation runs using different random seeds, meaning that each loan in the HMDA dataset is effectively sampled ten times. For each loan, the distribution of potential property values—and, by extension, the initial LTV—is simulated conditional on the following mortgage origination variables: borrower income, loan amount, DTI ratio, and interest rate. The conditional distribution of property values is then used to estimate the probability that a loan is assigned to a particular property within its reported census tract, which should indirectly account for the correlation between property values and borrower income at the time of origination.

Revised text from lines 580-586: For each simulation run, we simulate the financial conditions of 4.7 million borrowers with single-family mortgages originated during the 1992-2019 period at a monthly timestep over the life of their loan. Because certain variables describing the initial financial conditions and repayment profiles (model III) of mortgage borrowers are stochastically generated, model projections of flood-related credit constraints were averaged over ten simulation runs conducted with different random seeds. This number of replicates was found to be sufficient for achieving stable estimates of the number of borrowers facing flood-related credit constraints across the study area; however, generating stable estimates for smaller geographic units (e.g., specific census tracts) would likely require additional simulation runs.

Revised text from lines 443-446: Because HMDA mortgage origination data is anonymized to the census tract level, each mortgage loan is randomly assigned to a

specific property within the listed census tract at origination. The likelihood of a given property being matched to a loan is determined based on its estimated value at the time of origination (model II, Sect. 3.4) and the probability density function (PDF) of potential property values implied by the mortgage loan amount and LTV ratio distribution.

• Is there anything like Fig. S9 for defaults?

Our reply: We do not present a comparable figure for defaults because our mortgage repayment model (model III) only simulates loan terminations resulting from voluntary payoffs (i.e., prepayments and maturity payments) and does not track terminations from defaults or foreclosures. We have added the following text to the manuscript for clarity.

Revised text from lines 472-478: Our model only simulates loan terminations resulting from voluntary payoffs (i.e., prepayments and maturity payments) and does not track terminations from defaults or foreclosures. The omission of default-related terminations is unlikely to materially affect the loan age distribution, as the "background" rate of default was low relative to the rate of voluntary payoffs. Among GSE-backed single-family mortgages in North Carolina that were active at any point from 2000 to 2019, only 3.3% of loans were ever more than 120 days delinquent (a prerequisite for initiating foreclosure proceedings) and over 97% of loan terminations during this period resulted from voluntary payoffs (Fannie Mae, 2023; Freddie Mac, 2023).

I don't think Model IV has enough information. How does it account for things like heterogeneous savings for households that are insured or not? With the same income, a household without insurance would accrue more savings by not paying insurance costs. At the property level, there is also a difference in the degree to which a household saves money on purchasing their home based on salient price signals for risk (particularly, in North Carolina Pope (2008) showed how seller disclosures improved price signals to buyers looking to live in the SFHA: https://doi.org/10.3368/le.84.4.551). Seems like the model assumes everyone's income goes up? There is no job loss? What do recovery trajectories look like after flood events? Is it only possible through loans for uninsured properties? Does income grow for all homeowner types based on county-level annual trends? Do damaged properties recover value over some time period? Does price appreciation/depreciation occur differently for insured and uninsured properties? The authors mentioned some factors like employment and income in some of their mechanisms for default, but I don't see how it's incorporated in the modeling framework in a realistic way. Can the authors talk more about what their assumptions are about exogenous factors over the sample? It would help to have a conceptual model of household finances and indicate what this study includes/excludes and why.

Our reply: We thank the reviewer for these thoughtful comments. We fully agree that the factors mentioned—such as heterogeneity in household savings behavior by insurance status, risk-based price signals in property markets, and exogenous shocks to employment and income—can all play a role in shaping household financial outcomes following flood events. Our modeling framework was designed to focus on a narrower question: how does flood-related property damage affect the ability of uninsured borrowers to access low-cost forms of debt financing through its influence on combined loan-to-value (CLTV) and debt-to-income (DTI) ratios? To maintain tractability and isolate this mechanism, the model does not explicitly simulate savings behavior, labor market shocks, or differential property price trajectories across insured and uninsured households. Where possible, we conducted sensitivity analysis to evaluate the impact of these modeling choices on our findings; these include a sensitivity analysis on the amount of grant aid to uninsured households (which can also be thought of as a proxy for emergency savings) as well as a variance-based sensitivity analysis that examines the relative influence of uncertainty in borrower income, property values, and damage costs. We have clarified these simplifying assumptions in the revised manuscript and added a conceptual model (Table S6) outlining the financial variables and processes that are included versus excluded in our simulation framework.

Revised text in manuscript lines 587-595: Our approach to modeling household financial conditions focuses on how uninsured property damage affects the borrowing capacity of flood-exposed property owners through its influence on CLTV and DTI ratios. It does not, however, capture the full range of factors and processes that may play a role in shaping household financial outcomes following flood events. These include household saving behaviors, which may be heterogenous by wealth and insurance status; the timing of insurance claim payouts (which are assumed to immediately offset the cost of flood damage for insured borrowers); exogenous shocks to income arising from changes in employment status and negative life events; and the ability of households to supplement or replace home equity-based borrowing with other sources of funding for recovery, as described in Section 2. A conceptual overview of common household budget components that were included and excluded from our model is provided in Table S6.

Revised text in manuscript lines 898-904: Third, when modeling the financial conditions of residential mortgage borrowers, household income was assumed to grow over time at a rate equal to the change in average personal income for a given county and year. Data from longitudinal studies of income dynamics suggest that in reality, the rate of income growth varies depending on a household's initial wealth, and that year-to-year changes in income can be highly volatile even within a given income stratum (Fisher et al., 2016). In addition, our modeling approach does not consider exogenous income shocks arising from events such as job loss, illness, or divorce. Including these sources of

variability in household income would likely increase the number of mortgage borrowers projected to experience income-related credit constraints following exposure to flooding.

**Table S6 in the Supplementary Information:**

Table S6. Conceptual model of mortgage borrower finances.

| Line item a                        | Corresponding variable in model IV | Units         |
|-----------------------------------------------|------------------------------------|---------------|
| Household balance sheet                       |                                    |               |
| Assets                                        |                                    |               |
| Primary residence                             | $P_t$                              | USD (nominal) |
| Secondary and rental properties               | Not modeled                        |               |
| Liquid savings                                | Not modeled                        |               |
| Retirement and investment accounts            | Not modeled                        |               |
| Vehicles and other personal property          | Not modeled                        |               |
| Liabilities                                   |                                    |               |
| Primary mortgage                              | $B_{M,t}$                          | USD (nominal) |
| Home repair loans b                | $B_{R,i,t}$                        | USD (nominal) |
| Mortgages on other properties                 | Not modeled                        |               |
| Auto loans                                    | Not modeled                        |               |
| Student loans                                 | Not modeled                        |               |
| Credit cards                                  | Not modeled                        |               |
| Unpaid bills and other debt                   | Not modeled                        |               |
| Household cashflows                           |                                    |               |
| Cash inflows                                  |                                    |               |
| Stable and predictable income c    | $I_t$                              | USD per month |
| Fluctuating and variable income c  | Not modeled                        |               |
| Post-disaster aid                             | Not modeled                        |               |
| Cash outflows                                 |                                    |               |
| Primary mortgage payment                      | $c_M$                              | USD per month |
| Repair loan payments b             | $c_{R,i}$                          | USD per month |
| Other recurring debt obligations d | $c_{NM}$                           | USD per month |
| Taxes and insurance c              | Not modeled                        |               |

USD: United States dollars.

<sup>aThe entries listed within this table represent a non-exhaustive list of common household budget items.

<sup>bUninsured borrowers are assumed to finance flood-related repairs through home equity-based borrowing.

<sup>cBorrower income is initialized at origination and assumed to evolve deterministically over time according to county-level trends in personal income growth. We did not model exogenous shocks to household income or changes in employment status.

<sup>dIncludes payments on sources of debt which were not explicitly modeled (e.g., auto loans, credit cards) but which nevertheless affect a borrower's DTI ratio. These obligations are assumed to remain constant over time.

eWe did not model housing expenses associated with property taxes, homeowners' insurance, or flood insurance.

• I recognize that the methods state the study does not look at aid, but why not? How do the results of Deryugina (2017) on social insurance relate to recovery trajectories? In the US, IHP actually requires households to purchase insurance as a condition of the aid — why exclude this type of mechanism from the integrated workflow? It is crucial that the authors justify modeling choices based on their study framing, not based on simplicity. Modeling choices based on simplicity may not always be appropriate for satisfying a study's goals. I'm not sure that the current choices around simplicity are justifiable given the current framing. The study uses real historical flood events and projects defaults based on different theories of how financial conditions produce defaults, but ignores other real-world characteristics that would mediate these outcomes. The "real-world" framing requires very careful justification for departures from reality.

Our reply: Although we do not explicitly model the allocation of post-disaster aid, our analysis includes a sensitivity analysis that examines the potential impact of cash grants on household financial conditions (Fig. 9). Given the revised framing of our analysis around post-flood credit constraints (as opposed to loan defaults) we believe the exclusion of post-disaster aid and social safety net programs is appropriate for the scope of this analysis. We fully acknowledge that these programs are likely to shape the long-term recovery trajectories of uninsured households; however, our objective is to isolate how property damage affects access to credit through its influence on CLTV and DTI ratios. Modeling the broader recovery outcomes of credit constrained households would require simulating additional mechanisms—such as the timing, distribution, and behavioral responses to aid—that are beyond the scope of our present framework. We have clarified this in the manuscript while noting that future work could explicitly incorporate post-disaster aid to capture these downstream recovery dynamics.

Revised text in manuscript lines 915-927: Finally, we did not explicitly model the various sources of funding for post-disaster recovery that might be available to uninsured mortgage borrowers who lack sufficient equity or liquidity to obtain private home repair loans. These include: federal sources of post-disaster aid such as SBA loans and FEMA IHP grants; alternative finance sources such as payday lenders, auto title loans, and pawnbrokers; and liquid assets such as personal savings and retirement accounts. To examine how the availability of low-interest SBA loans and FEMA IHP grants may impact our findings, we conducted scenario analyses on the interest rate at which borrowers can finance home repairs as well as the amount of home repair assistance available to those without insurance. The number of credit constrained mortgage borrowers was sensitive to the amount of grant aid available but relatively insensitive to the interest rate on home repair loans. During Hurricanes Matthew and Florence, less than a third of property owners who applied for IHP aid were approved, and the average grant awarded was under \$5,000 (GAO, 2020b); thus, it appears unlikely that the

inclusion of IHP aid would substantially alter estimates of the number of mortgage borrowers facing flood-related credit constraints. Future research could examine how these programs are likely to shape the long-term recovery outcomes of credit constrained mortgage borrowers by explicitly incorporating the timing and distribution of post-disaster aid into the integrated modeling framework.

**Figure 9 in the Manuscript:**

Figure 9. Scenario analysis on the amount of home repair grant assistance available to mortgage borrowers should they experience uninsured flood damage. The amount of available assistance is varied between zero and \$42,500—the maximum award that households can receive from FEMA's Individuals and Households Program (IHP) as of 2024 (U.S. GPO, 2023).

• How does this framework account for the '08 financial recession, which occurs right before Irene? Does the property model adequately project the drop in value during this period? Do the financial models adequately account for the rise in poor financial conditions (and I assume defaults) at the household level? It also begs the question of contextualizing projected defaults from flood stress relative to defaults from the '08 recession. It seems to me that neglecting important exogenous drivers of poor financial conditions could lead to both under estimation in this modeling workflow. The decrease in property values and employment from the great recession, plus Irene in 2011, might mean the model underestimates a large default event based on the double-trigger mechanism?

Our reply: The property value model projects a drop in home prices during the 2008 global financial crisis (GFC) that aligns well with trends seen in observed sale prices (Fig. S7). As such, our modeling framework should capture the effects of the GFC on property values and (by extension) homeowner equity. However, our framework does not account for how elevated rates of unemployment and tightening mortgage lending standards during the GFC may have reduced the availability of credit to flood-affected homeowners. We have highlighted how these factors are likely to influence our findings in the revised manuscript.

Revised text from lines 905-914: Fourth, our model framework does not account for how factors related to the 2008 global financial crisis (GFC) may have impacted the financial health and credit access of mortgage borrowers during the study period. These include elevated rates of unemployment that persisted for several years following the GFC and a tightening of mortgage lending standards that reduced the availability of credit to property owners. Mortgage lending standards in the U.S. underwent a gradual loosening during the early 2000s leading up to the crisis, followed by a sharp tightening during the 2007-2009 period that led to increases in loan denial rates and more stringent LTV and DTI requirements (Vojtech et al., 2020). Of the seven flood events evaluated in this study, the effects of the GFC would be most relevant for Hurricane Irene, which occurred in 2011 when the economy was still recovering from the crisis. If the elevated rate of unemployment and reduced credit supply during this period were incorporated into our model, projections of credit constraints among borrowers exposed to flooding from Hurricane Irene would likely be higher.

**Figure S7 in the Supplementary Information:**

Figure S7. Property value model error by period.

The distribution of absolute error associated with cross-validation predictions for sales occurring in a given year are depicted by the black box-and-whisker plots. Whisker boundaries correspond to the 10th and 90th percentiles of absolute error. For comparison purposes, the median observed sale price of properties included in our sample in each year is depicted by the blue line, while the median predicted sale price is depicted by the red dashed line.

**Results**

Notes: In light of the reviewer's general feedback, we have reframed the analysis around flood-related credit constraints and reduced the emphasis placed on mortgage default as the main outcome of interest. For more information on how this reframing was implemented in the revised manuscript, please see the author's remarks on the first page of this document. As part of these changes, we have modified our terminology as follows:

| Old terminology                     | New terminology                             |  |
|-------------------------------------|---------------------------------------------|--|
| "at risk of default"                | "credit constrained"                        |  |
| "at risk of strategic default"      | "collateral constrained"                    |  |
| "at risk of cashflow default"       | "income constrained"                        |  |
| "at risk of double-trigger default" | "constrained by both income and collateral" |  |

Please note that our responses to the reviewer's comments on our results section utilize the new terminology.

• What is "flood damage exposure" in Fig 3 and 4a? Is it damage to structures?

Our reply: Figures 3 and 4a depict the dollar amount of flood damage to structures within the study area. For clarity, we have changed the axis labels of these figures from "flood damage exposure" to "flood damage cost".

• I'm surprised by the amount of attention in the results to describing the damage estimates both overall and stratified across a few groups. I think the authors should restructure their results to focus on their main goal, which is the degree to which default occurs under different assumptions of causal pathways of damage, financial preconditions, insurance, and mortgage debt.

Our reply: We appreciate the reviewer's comment and agree that the results should clearly align with the study's main objectives. We have revised the introduction to make it clear that one of the goals of our study is to quantify past exposure to uninsured flood damage. Our discussions with policymakers in North Carolina (including the funders of this study) suggest that this information is highly valued by state decision makers. For this reason, we believe it is important to retain and adequately highlight these results in our manuscript.

• L576-577, what is the proportion of both located outside SFHA and lacking flood exposure?

Our reply: Based on context of the sentence the reviewer is referring to in this comment, we think they most likely meant to inquire about the proportion of flood-exposed borrowers located outside the SFHA and lacking flood *insurance* (as opposed to lacking flood *exposure*). If we are mistaken about this, please let us know. We have included the following text in the revised manuscript based on our interpretation of their comment.

Revised text from lines 684-686: Among borrowers exposed to flood damage, 11,100 (50%) were located outside of the SFHA and 11,500 (52%) lacked flood insurance at the time of their exposure, with non-SFHA borrowers accounting for 73% of those exposed to uninsured damage.

• The results on default would be more interpretable if the authors showed how many defaults there would be under the three different causal pathways of default. Most readers will not be able to interpret the 6 metrics of Figure 6 and what it means for default projections. If not this, it could really help to reintroduce the acronyms in the results section and to not use acronyms in the figure caption. But I highly encourage taking the conditions required for certain default mechanisms (e.g., ACLTV > 100% and ADTI > 45% for double-trigger) and showing the proportion of households pre & post flood that met those conditions. I encourage the authors to really focus on their key outcome of interest and focus on interpretability since their topic is extremely important and many readers of this journal come from different disciplines and will need help understanding new concepts.

Our reply: We agree with the reviewer that improving the interpretability of the default results is important, particularly for readers from different disciplines. To address this, we have included a comparison of the number of borrowers projected to face post-flood credit constraints from different causes (insufficient income, insufficient collateral, or both in combination) in Figures 7 and 8. We have also revised the caption of Figure 6 to make financial metrics such as DTI and CLTV more accessible to a broader audience.

Revised caption to Figure 6: Damage-adjusted debt-to-income (DTI) and combined loan-to-value (CLTV) ratios among mortgage borrowers exposed to uninsured flood damage. The DTI ratio measures the share of a borrower's monthly income consumed by recurring debt obligations, while the CLTV ratio measures home equity as the total balance of all loans secured by a property divided by its market value. The post-flood adjusted DTI (ADTI) and adjusted CLTV (ACLTV) ratios capture the projected effects of financing flood-related repairs through home equity-based borrowing on borrowers'

cashflow and equity positions. Dashed lines indicate the ADTI and ACLTV thresholds used to classify borrowers as credit constrained following exposure to uninsured damage.

• I think it could be helpful to compare the default results for both uninsured and insured properties, perhaps to emphasize the additional risk of being uninsured. However, to do this comparison, one would have to account for the additional savings uninsured households might accrue by not paying for flood insurance. While the sample period predates Risk Rating 2.0, that is an especially important consideration as the price of insurance goes up. Also important is that many households, even if insured, need to rely on savings for some time after a large event due to slow timing in damage assessments and payouts. This ties into my questioning about whether the "real-world" framing is appropriate.

Our reply: We thank the reviewer for this thoughtful comment. We agree that comparing insured and uninsured properties while accounting for differences in savings and insurance premiums would be an interesting extension, particularly in light of rising costs under Risk Rating 2.0. However, this question lies outside the scope of our current analysis, which focuses on post-flood credit constraints rather than the broader financial tradeoffs of insurance participation. In our modeling framework, insurance payouts are assumed to immediately offset flood-related repair costs for insured borrowers, such that these households do not experience flood-induced credit constraints. We acknowledge that, in reality, delays in insurance claim payouts may create short-term financial distress for insured borrowers, which may require them to rely on savings, credit cards, or other coping strategies. We have clarified in the manuscript that our model does not capture saving behaviors nor the timing of insurance payouts.

Revised text from lines 587-595: Our approach to modeling household financial conditions focuses on how uninsured property damage affects the borrowing capacity of flood-exposed property owners through its influence on CLTV and DTI ratios. It does not, however, capture the full range of factors and processes that may play a role in shaping household financial outcomes following flood events. These include household saving behaviors, which may be heterogenous by wealth and insurance status; the timing of insurance claim payouts (which are assumed to immediately offset the cost of flood damage for insured borrowers); exogenous shocks to income arising from changes in employment status and negative life events; and the ability of households to supplement or replace home equity-based borrowing with other sources of funding for recovery, as described in Section 2. A conceptual overview of common household budget components that were included and excluded from our model is provided in Table S6.

• I also think it would be helpful to compare the default projections under the floods versus alternative "normal" rates of defaults (some households go under hard times and may have to default) and "financial crisis" rates of defaults (like '08) to help contextualize the projections.

Our reply: We thank the reviewer for this helpful suggestion. To better contextualize our results, we have compared our projections of negative equity among borrowers exposed to flooding (who are considered to be "collateral constrained" but not necessarily in default) to rates of negative equity observed during the peak of the global financial crisis.

Revised text from lines 695-697: Among borrowers exposed to flooding, 28% were projected to experience negative equity (ACLTV > 100%) after accounting for flood-related property damage; for comparison, 23% of U.S. mortgage borrowers had negative equity during the peak of the global financial crisis (James, 2009).

• The modeling assumptions seem to force the modeling results, particularly that uninsured households have few ways to financially recover between events and have to wait for their income to bounce back. What happens if the main income bringer lose their jobs (insured or uninsured households)? What happens if uninsured households end up with more aid? I'm not suggesting the study has to address all of these questions in its model, but I do think it needs to acknowledge how certain modeling assumptions will lead to certain outcomes when making comparisons across groups. What is left out of the model that might overstate the differences in default outcomes across groups of comparison? This is one reason a conceptual model could help.

Our reply: We have added the following text to the manuscript to highlight how certain modeling assumptions may influence comparisons across groups.

Revised text from lines 759-771: Our projections of flood-related credit constraints by income, property value, and loan age (Fig. 8) should be interpreted in light of several modeling assumptions that may influence comparisons across groups. First, we did not account for the positive correlation between income and flood insurance take-up when assigning loans to specific properties within a census tract (Section 3.5). Incorporating this source of heterogeneity in flood insurance adoption would likely increase projected credit constraints for low-income borrowers and reduce them for high-income borrowers, particularly in areas outside the SFHA where insurance purchase is voluntary. Second, we assumed that borrower income evolves according to county-level trends in per-capita income growth (Section 3.6) and did not model changes in employment status. This assumption may overstate the protective effect of loan age, particularly for incomerelated credit constraints. Finally, our framework focuses on the ability of uninsured

borrowers to finance repairs through home equity-based borrowing and does not capture how the ability to draw upon other sources of funding for recovery (such as savings and investments) may differ by income and property value. Because wealthier households tend to hold a greater share of their net worth in non-physical assets such as stocks (Jones and Neelakantan, 2023), the reliance on home equity-based borrowing for recovery is likely less pronounced among higher-income and higher-property-value households.

• I think comparing the number of at-risk default by causal pathways is great, which is what Figures 7 & 8 do, but I'm confused about the result. Because these are not mutually exclusively, these are likely not the correct visualization types for this result. Separately, it would help to contextualize these results in terms of the overall residential stock to help readers understand how large this problem is. Further, I think the current framing of the study leads the reader to expect much more results focused on these causal pathways and the default outcome. As I mentioned in my comments on the methods, I think it would be very effective to focus on how uncertainty in inputs & models propagates through the integrated modeling workflow and leads to different projections of default outcomes, conditioned on the different causal pathways one believes.

Our reply: We thank the reviewer for highlighting this potential point of confusion in Figures 7 and 8. In these figures, we use the labels "collateral constrained only" and "income constrained only" to distinguish those who faced collateral or income constraints but not both simultaneously. This ensures that the groups used within the stacked bar charts are mutually exclusive. We have clarified the definitions of these groups in the manuscript text and figure captions. Regarding the reviewer's points regarding contextualization and uncertainty analysis, please see our responses to their subsequent comments.

Text added to captions of Figures 7 and 8: Borrowers constrained by collateral only  $(ACLTV > 100\%, ADTI \le 45\%)$  are shown in red; those constrained by income only  $(ACLTV \le 100\%, ADTI > 45\%)$  in blue; and those constrained by both (ACLTV > 100%, ADTI > 45%) in purple.

• For Figure 8, proportions could be helpful to contextualize the results (like the % for the other bar charts). 22,100 damaged homes out of 4.7M is important context when thinking about multisector impacts. Are we talking about financial risks that affect homeowners exclusively, or are there cascading impacts across sectors? It would be helpful to see more about whether 11,000 is a large number or not in this setting.

Our reply: We thank the reviewer for this helpful suggestion. In response, we have incorporated proportions into the results text describing the share of mortgage borrowers

projected to face flood-related credit constraints at any point during the study period (i.e., the period prevalence). However, because the groups shown in Figure 8 are defined by time-varying characteristics (e.g., loan age), it is not straightforward to define an appropriate denominator for calculating the period prevalence within each group, as borrowers can belong to multiple groups over the life of their loan. An alternative approach would be to calculate the incidence rate of credit constraints (e.g., events per loan-year) while accounting for dynamic changes in group membership. However, we don't believe adding this to Figure 8 would substantially improve interpretability for readers.

**Revised text from lines 682-684:** Among 4.7 million single-family mortgages originated in the study area from 1992 to 2019, approximately 22,100 (0.47%) are estimated to have experienced flood damage at least once over the life of the loan from one or more of the seven evaluated events.

Revised text from lines 691-693: Over the study period, 7,180 mortgage borrowers were projected to face flood-related credit constraints as indicated by ACLTV > 100% or ADTI > 45%. This number represents a small share (0.15%) of all mortgages originated during the study period but a substantial fraction (32%) of those exposed to flooding.

• The text about liquidity (L594-L604) raises questions about the HMDA sampling. Is it possible that the sampling assigns lower incomes and higher loan amounts to certain high damage households? Looking at results in terms of the sensitivity analysis I suggested above could help readers understand what drives different default outcomes in terms of modeling assumptions. As it currently is, it seems like the authors may be overinterpreting results strongly reliant on plausible but highly uncertain and insufficiently sampled modeling implementation.

Our reply: Because the sampling procedure used to assign loans to specific properties within a census tract is agnostic to the flood damage exposure of each property, we have no reason to believe that our approach disproportionately assigns lower-income and higher-LTV borrowers to damaged properties. It is worth noting that properties towards the extreme ends (i.e., top and bottom 20%) of the property value distribution exhibited the highest levels of flood damage exposure over the study period (Fig. S17), which may help to explain why so many lower-income borrowers experienced flood-related credit constraints. In addition, the amount of damage required to trigger an income-related credit constraint will tend to be lower for borrowers with less disposable income.

• I think it is important not to overinterpret results, which I am worried about for the results shown in Figures 7 and 8. The authors could draw stronger conclusions about liquidity,

income, property values, and defaults if they took more of a sensitivity analysis approach and implemented a method like scenario discovery.

Our reply: We have added a paragraph to the results section to underscore that our results should be interpreted in the context of several modeling assumptions that may influence comparisons across groups. For more information, please see our response to a related comment on page 46 of this document. We have also added a variance-based sensitivity analysis that examines the relative influence of uncertainty in borrower income, property values, and damage costs on key outcomes of interest. For more information regarding this sensitivity analysis, please see our response to "general comment" #4.

• It's great to see the sensitivity analysis about the home repair grant program. It reinforces for me that the strongest insights in this paper would come from a more comprehensive approach to characterizing the drivers of sensitivity in default projections. I think it is the best way for the authors to make an important and reusable contribution in this area. In particular, the current default model results occur under the assumption that there is no grant for home repair, but there are major monetary transfers after large disasters (specifically presidentially declared disasters such as the focus of this study) so it is invalid to ignore this in the "baseline" case. The IHP program specifically requires uninsured households to purchase NFIP insurance, so that requirement seems like "good" policy in terms of reducing default (compliance is a different story, not modeled here but part of the data generating process). This again reinforces why a sensitivity analysis approach would be constructive and insightful. It would be unfortunate to overestimate default risk relative to the current policy environment (though with recent changes to FEMA the authors could frame some of their study around how important these programs are to avoid default risk!).

Our reply: We thank the reviewer for these thoughtful comments and are pleased that they found the sensitivity analysis on home repair grants useful. We agree that examining the sensitivity of households' long-term recovery outcomes to policies around post-disaster aid would be a valuable extension of our work; however, this is outside the scope of the present study, whose objective is to isolate how property damage affects access to credit through its influence on CLTV and DTI ratios. As discussed in our responses to earlier comments, the sensitivity analysis around home repair grants is intended to illustrate the potential influence of such omitted processes and to gauge the robustness of our findings to their exclusion. We see this as a useful foundation for future work that could explicitly model post-disaster aid programs and their effects on household financial trajectories.

• It would have been nice to signal earlier to readers that the authors do a sensitivity analysis on damage costs and property values. This sensitivity analysis approach needs a better description of the methods and a justification for whether it's valid to do a uniform application of the uncertainty instead of randomly sampling from an uncertain distribution for each unit. I'm not sure that it is. For example, the methods show that for the property valuation model, only 54% of observations fall within +/-20%. Given the large uncertainty in the projected outcome based on this under-representation of uncertainty, it seems very important to better contextualize sensitivity of the projected outcome to the actual uncertainty in the property valuation model. In addition, as mentioned earlier, 20% seems like an inadequate representation of uncertainty for the damage model, especially given previous findings on heteroskedasticity with inundation depth.

Our reply: We thank the reviewer for this suggestion and agree that a uniform adjustment of  $\pm 20\%$  is likely to be an overly simplistic representation of the uncertainty associated with estimates of property values and damage costs. With this in mind, we have added a variance-based sensitivity analysis in which these inputs are modeled as a random variable whose mean and variance are defined at the level of individual borrowers. For more information regarding this sensitivity analysis, please see our response to "general comment" #4.

• It would be better to sample from uncertain factors using an appropriate experimental design at the spatial resolution of the model inputs (i.e., property-level). It would be very insightful for the authors to evaluate at least first-order sensitivities of the default projections to uncertainty in inputs and the authors could interpret results using relatively fast methods such as Method of Morris and scenario discovery. Since there are only ~22k houses under study for the default analysis, it seems like this is feasible. Several of the co-authors are more expert in these methods than I am, so I am interested to hear more about why they did not approach this complex new modeling workflow in a sensitivity analysis approach (e.g., among many works by Saltelli, please see https://www.jstor.org/stable/2676831 given the predominant framing of the integrated modeling workflow)

Our reply: We thank the reviewer for prompting us to include a more comprehensive analysis of parameter sensitivity in our study. We have added a variance-based sensitivity analysis that utilizes the method of Sobol' to evaluate how uncertainty in estimates of flood damage (model I), property value (model II), and borrower income (model IV) contribute to variance in borrower-level financial outcomes. For more information, please

see our response to "general comment" #4.

**Discussion**

• Nice, tight framing in the first paragraph. I encourage the authors to reflect on the differences in this framing to that of the Introduction, and to better synchronize the two sections to have a consistent story throughout the paper. In addition, the Results should focus on the main story.

Our reply: Thank you for bringing this issue to our attention. As described in the author's remarks, we have made substantial revisions to the introduction section to better align our framing of the analysis with the capabilities of our modeling framework.

• One issue with the first paragraph is the claim on L683-684: "Our results underscore the status of pre-flood home equity and debt-to-income ratio as important determinants of post-flood financial resilience." Is that true, or is that an artifact of implementing theories on causal pathway of default that treat these as determinants of default?

Our reply: Given the lack of data on long-term recovery outcomes, we agree that this statement is not directly tested by our analysis. We have removed this sentence from the discussion and instead focus on how pre-flood financial conditions influence the types of funding sources for recovery that are available to flood-exposed mortgage borrowers (without making claims about long-term outcomes).

• The discussion is well-written. It also introduces claims that point to some unfocused results on the distribution of damages, as opposed to the most interesting points about the default projections and the uncertain factors surrounding those. For instance, the paragraph on L741 is very interesting. Why didn't the authors model this insurance policy with a deductible equal to 50% of a borrower's equity? Seems like to really illustrate the value of this integrated modeling approach, showing that an end-user can use the approach to stress-test candidate policies against a range of uncertain factors seems appropriate and very useful to the research and policy community. Just speculating about this when the authors have the tool at their disposal to test their hypotheses is underwhelming.

Our reply: We agree that there is an urgent need to compare and evaluate the impact of different policy interventions on household financial conditions; however, this question is beyond the scope of our present study, which introduces the integrated modeling framework, describes its development, and demonstrates how it can be used to generate projections of flood-related credit constraints for historical flood events. In future work, we plan to use the model components developed in this study to perform a detailed policy analysis examining different interventions for improving household financial resilience to floods such as novel insurance products and expansions to post-disaster aid programs. For more information, please see our response to "general comment" #5.

• I'm confused by the 67,000 exposure estimates on L758. Why make a comparison to this number if the main number of interest is the ~ 22,000 damaged properties with mortgages?

Our reply: The sentence referenced by this comment compares our estimates of the number of structures with past flood exposure to similar estimates produced by Garcia et al. (2025) for a larger set of 78 flood events. Because Garcia et al. (2025) measures exposure in terms of the number of flooded buildings, it is more appropriate to compare their numbers against our estimates of damage to all properties (as opposed to just single-family detached homes with mortgages).

• The discussion does a nice job of mentioning limitations, but generally doesn't justify why the study did not account for some uncertain factors. The idea that the Matthew and Florence grants were small and wouldn't affect the number of mortgages at risk of default is something the authors can test with their framework. Why just speculate?

Our reply: Although we did not explicitly model the allocation of post-disaster aid, the sensitivity analyses we conducted on the amount of grant assistance available for home repairs and on the interest rate at which borrowers can finance repairs should respectively capture the potential impacts of FEMA IHP grants and SBA loans on borrower financial conditions. For more information, please see our response to "general comment" #6.

• What's missing from the discussion is more acknowledgement about the deep uncertainty about whether the outcomes the authors measure actually are strong predictors of defaults. As discussed in the comments on the Introduction, the authors do not present that evidence. I think mortgage default exposure framing might be more appropriate than mortgage default risk, unless the authors clarify that the default projections are conditioned on the belief one causal mechanism holds. This calls for more of a deep uncertainty framing to the study, which would work well with the sensitivity analysis approach that I think this complex, prediction-based modeling integration study calls for.

Our reply: We thank the reviewer for bringing this issue to our attention. In light of the reviewer's feedback, we have reframed the analysis around flood-related credit constraints and reduced the emphasis placed on mortgage default as the main outcome of interest. We believe this framing is similar to the "mortgage default exposure framing" proposed by the reviewer in this comment, to the extent that credit constraints are likely to be a necessary (but not sufficient) condition for mortgage default and other negative financial outcomes. For more information on how this reframing was implemented in the manuscript, please see the author's remarks on the first page of this document as well as our responses to the reviewer's comments on the introduction section of our paper.

**Technical corrections**

• L26: "losses from flooding are expected to surpass..." -> not sure this reference helps or is accurate. For one, it doesn't seem like any of the framing points rely on claims about

changes in flood risk over time. Second, this is just one study and the reference reads more into the study than the study's findings support. For example, using the passive voice here for "are expected" makes this seem like the estimate is a confident one. It would be more appropriate to say that one study estimates annual losses from flooding may surpass \$40B... I also think the claim that losses will surpass \$40B as a result of increases in extreme precipitation under climate change is not supported by this reference. The study does not highlight the role of changes in extreme precipitation in changing risk estimates.

Our reply: We have removed this reference from the introduction.

• L38 – the text about household ability and willingness to pay and demand could benefit from more specificity. What do the authors mean by "further reduces demand" on L39? Depending on whether they mean the demand curve or quantity demanded, it may be more accurate to say "which reflect low demand."

Our reply: We have revised the sentence referenced by this comment to clarify that we are referring to the impact of willingness-to-pay on rates of flood insurance uptake.

Revised text from lines 88-90: Many households have limited ability or willingness to pay for flood insurance (Atreya et al., 2015; Kousky, 2011; Netusil et al., 2021), which poses a major barrier to increasing uptake.

• The Gourevitch et al., (2023) reference on L125 does not seem to apply here. Where is the event-based specification of Gourevitch? They estimate "overvaluation" based on the degree to which the market capitalized information about properties mapped into the SFHA between sales. The studies that the authors cited on L77 would be more appropriate here, but note the references I mentioned in my comment that suggest the evidence on post-flood price adjustments is mixed (with more detailed specifications on drivers of risk and household characteristics showing a more heterogeneous market adjustment).

Our reply: The paragraph in which this reference appeared was removed during our revisions of the introduction section.

[revised manuscript text omitted]

---

## Author Comment (AC2)

**Response to Reviewer #2**

**Author's remarks**

We thank the reviewer for their valuable comments and appreciate the time and effort they put into reviewing our manuscript.

In response to feedback received from Reviewer #1, we have reframed the analysis around flood-related credit constraints and reduced the emphasis placed on mortgage default as the main outcome of interest. Our model framework estimates the degree to which uninsured flood damage increased the potential for financial states (e.g., negative equity and cashflow problems) that can impair a mortgage borrower's ability to fund repairs to their property through additional borrowing; however, given the lack of data on mortgage performance, the degree to which these states are predictive of default remains unclear. Whether negative equity or cashflow problems trigger a borrower to default is also likely to depend on the availability of alternative sources of funding for home repairs that were not explicitly modeled, including disaster assistance grants, personal savings, and informal transfers from family and friends. As such, we have moved away from describing these borrowers as "at risk of default" and instead describe them as "credit constrained," reflecting their diminished capacity to fund repairs by taking on debt while acknowledging that their long-term recovery outcomes are uncertain and depend on several factors not captured by our analysis. We believe that this more modest framing aligns our study objectives and research questions more closely with our methodological approach.

As part of this reframing, we have made substantial changes to the introduction and background sections of the manuscript, and have modified our terminology as follows:

| Original terminology                | New terminology                             |
|-------------------------------------|---------------------------------------------|
| "at risk of default"                | "credit constrained"                        |
| "at risk of strategic default"      | "collateral constrained"                    |
| "at risk of cashflow default"       | "income constrained"                        |
| "at risk of double-trigger default" | "constrained by both income and collateral" |

Please be aware that our responses to Reviewer #2's comments utilize the new terminology, even when their comments use the original terminology. In addition, line numbers referenced in our response correspond to the revised manuscript and may differ from those referenced in the reviewer's comments. A point-by-point response to their review is included below, with reviewer comments shown in **black** and our replies in **blue**.

**Review of "Flood risks to the financial stability of residential mortgage borrowers: An integrated modeling approach"**

**General Comment Statement**

The paper presents a significant and well documented contribution to the field of climate financial risk. The integrated, "bottom-up" modelling framework, which links property-level flood damage to household financial distress, is a commendable and ambitious effort to advance the understanding of this critical issue.

Our reply: We thank the reviewer for their thoughtful evaluation of our analysis.

As I am not an expert in US insurance market, I focused the review mainly on the modelling part of the work. While the framework is conceptually sound, the analysis concludes that its current implementation contains a cascade of methodological limitations that is likely going to lead to a systematic underestimation of the true financial risk.

Our reply: We thank the reviewer for highlighting this important concern. We agree that our estimates of flood damage exposure and (by extension) post-flood credit constraints are likely to be conservative due to the low sensitivity of our flood damage model, which fails to detect many properties that were damaged. We have revised the manuscript in several places to clarify for readers that our findings should be interpreted in light of this limitation (see responses below).

**Line 18:** The finding that the evaluated floods "generated \$4.0 billion in property damage" is a key quantitative output. However, this figure should be interpreted as a conservative bound. As mentioned below (see comments on Line 271), the damage detection model fails to identify a majority of properties that actually sustained damage, meaning the true total damage might be higher.

**Our reply:** We have revised this sentence in the abstract to clarify that \$4.0 billion is a conservative bound.

Revised text from lines 18-19: Conservative projections suggest that the floods evaluated generated \$4.0 billion in property damage across the study area, of which 66% was uninsured.

**Line 20:** The statement that 32% of affected borrowers lacked sufficient income or collateral, "placing them at an elevated risk of default," is based on the underwriting criteria used. Is there any risk that the criteria used could lead to an underestimation of the number of borrowers who would be denied credit and thus be at risk?

Our reply: The underwriting criteria used to classify borrowers as income constrained or collateral constrained reflect the maximum allowable DTI and CLTV ratios under a government program that insures mortgages made by lenders to disaster-affected property owners; however, borrowers below these limits can still be denied a loan based on their credit history. While we believe that existing mortgage borrowers are likely to have high credit scores relative to the

general population, those with a history of missed payments may face additional challenges in accessing repair loans that are not captured by our modeling framework. We have included the following text in the manuscript to highlight how the omission of factors related to credit history are likely to influence our findings.

Revised text from lines 547-551: It should be noted that borrowers meeting these ratio-based criteria can still be denied a loan due to unsatisfactory credit history—a process that is not represented in our modeling framework. While existing mortgage borrowers have (by definition) previously met lending standards and likely possess higher credit scores than the general population, the omission of factors related to credit history may cause us to underestimate the share of flood-exposed borrowers who would be denied a loan.

**Lines 234-240:** The generation of "pseudo-absence" points is a pragmatic solution to incomplete data but introduces noise. The authors' own validation (Line 276) shows that model precision increases significantly when these points are excluded, suggesting that a number of these randomly generated "undamaged" points likely distorted the model's training (actually damaged?).

Our reply: We appreciate the reviewer's point and agree that the use of pseudo-absences introduces some label noise into the training data. However, this step was necessary to correct for the selection bias inherent in the address-level insurance data, which disproportionately captured damaged (claim) locations. Without pseudo-absences, flood presence locations are overrepresented in the training data, leading to systematic overprediction of flood damage across the study area, particularly for pre-2009 events where the number of missing address-level policies was high. While it is true that precision improves when pseudo-absences are excluded from the validation data, this result mainly reflects a change in how the model is being evaluated rather than a true increase in predictive performance. Since our goal is to generalize predictions to the broader set of properties (including those without insurance), including pseudo-absences in the training data provides a more representative sample, even if it introduces some label noise.

Revised text from lines 292-294: While the inclusion of pseudo-absences likely introduces some label noise into the training data, this step was necessary to correct for the bias inherent in the address-level insurance data, which disproportionately captured damaged (claim) locations.

Line 271: A very low sensitivity of just 12% to 42% means the model fails to identify between 58% and 88% of properties that were actually damaged. This is a foundational error that guarantees a systematic underestimation of the total number of impacted households and the total damage costs. All subsequent risk estimates are therefore performed on a small fraction of the true at-risk population.

Our reply: We thank the reviewer for prompting us to address this issue more explicitly in the manuscript. We agree that the model's low sensitivity indicates that many damaged properties were not detected, leading to an underestimation of true flood exposure. We have revised the

manuscript to more clearly acknowledge this limitation and to explain its cause and implications. Specifically, we now note that low sensitivity is characteristic of models trained on class-imbalanced data, and we clarify that our results should be interpreted as a conservative lower bound on total flood exposure rather than a central estimate.

Revised text from lines 324-333: When identifying damaged properties, the model exhibited high accuracy (≥92%) and specificity (≥98%) but low sensitivity, with true positive rates of between 12% and 42% across events. This behavior is characteristic of machine learning classifiers trained on class imbalanced data where the positive class (e.g., presence of flood damage) is rare compared to the negative class (Haixiang et al., 2017; He and Cheng, 2021). Among properties that were misclassified by our model in cross-validation, false positive and false negative predictions respectively accounted for 12% and 88% of model errors across the seven evaluated events (Table S4). Collectively, these results suggest that our model often fails to detect properties that were damaged, which is likely to lead to a systematic underestimation of the true level of flood exposure within the study area. As such, our projections of flood damage exposure (and, by extension, flood-related credit constraints) should be interpreted as a conservative bound as opposed to a central estimate.

**Lines 272-276:** The authors' framing of this result is misleading. The model's high precision is emphasized while downplaying the severe consequence of the high false-negative rate. In risk assessment, particularly for disaster aid, the cost of a false negative (failing to identify a household in need) is high.

Our reply: We have removed the text related to the model's high precision from the paragraph describing our cross-validation results, which now places greater emphasis on the consequences of our model's high false-negative rate. The revised paragraph is shown in our response to the reviewer's previous comment.

**Lines 343-345:** The reported accuracy is a major concern. Only 54% of the model's value predictions fall within  $\pm 20\%$  of the actual sale price. The authors later note this is the largest source of uncertainty in their final results (Lines 674).

Our reply: We acknowledge the substantial uncertainty in our property value estimates. To address this, we have added text in the revised manuscript highlighting potential sources of error in our property valuation model. In response to comments from Reviewer #1, we have also included a variance-based sensitivity analysis that quantifies the contribution of uncertain model inputs (including property values) to variation in our projections of post-flood credit constraints. This analysis confirms that property values are the largest source of uncertainty in our results.

Revised text from lines 406-414: The substantial uncertainty in our property value estimates likely arises from a combination of factors, including: (1) the limited number of property-specific details in NCEM's statewide building inventory, which describes basic structural attributes but lacks information on other price-relevant characteristics such as recent improvements or deferred

maintenance; (2) the presence of sales that do not reflect fair market values (e.g., intrafamily transfers) in the training and validation data, which can bias model predictions; and (3) geolocation errors that may result in mismatches between recorded sales and parcel geometries. Future work could potentially enhance the performance of the property valuation model by introducing filters to identify arms-length sales and by adding predictors that capture property-specific attributes related to structural defense and prior flood exposure (Nolte et al., 2024; Pollack and Kaufmann, 2022).

**Lines 357-360:** Is the use of GSE data to model the entire market, creating a bias of the "typical" borrowing population? If yes, it should be stipulated to keep the modelling results in perspective.

Our reply: We thank the reviewer for highlighting this important modeling assumption. We have added the following text to the manuscript to clarify the degree to which the GSE loans are representative of the broader U.S. mortgage market.

Revised text from lines 480-490: It is important to note that mortgages acquired by the GSEs—which account for approximately half of all U.S. mortgage originations (GAO, 2019)—consist of "conforming" loans that meet standardized requirements related to loan size, borrower credit quality, and documentation. Mortgages that are not represented in the GSE data include "jumbo" loans whose amounts exceed the conforming loan limit, which are typically associated with very expensive properties; "subprime" loans made to borrowers with questionable credit history or unverifiable income, which peaked at 15% of the U.S. mortgage market in the years leading up to the 2007 subprime mortgage crisis (Agarwal and Ho, 2007); and loans insured by government programs targeting specific groups such as first-time homebuyers, veterans, and active-duty military personnel (Jones, 2022; Perl, 2018). As such, borrower attributes that were simulated based on GSE data primarily reflect the characteristics of middle-income, creditworthy borrowers, and may underrepresent the characteristics of households at both the upper and lower ends of the wealth distribution and of communities in North Carolina with a large military presence such as Cumberland, Onslow, and Craven counties (N.C. Department of Military and Veterans Affairs, 2025).

**Line 583:** The conclusion must be interpreted as a conservative floor, not a central estimate, due to the cascading methodological issues outlined above and should be mentioned as such.

Our reply: We have added the following text to the manuscript to underscore that our projections of flood-related credit constraints are likely to be conservative.

Revised text from lines 691-695: Over the study period, 7,180 mortgage borrowers were projected to face flood-related credit constraints as indicated by ACLTV > 100% or ADTI > 45%. This number represents a small share (0.15%) of all mortgages originated during the study period but a substantial fraction (32%) of those exposed to flooding. Given the relatively low sensitivity of our flood damage model observed in cross-validation (Section 3.3), our projections

may underestimate the true number of borrowers exposed to flooding over the study period and prevalence of flood-related credit constraints.

Revised text from lines 894-897: In addition, cross-validation results suggest that our machine learning-based approach often failed to detect properties that were damaged, which is likely to contribute to a systematic underestimation of the true level of flood exposure within the study area. For these reasons, our projections of flood damage exposure and flood-related credit constraints should be interpreted as conservative bounds as opposed to central estimates.

**Line 494-675:** While the paper's focus is on flood risk, its analysis spans a period in which the U.S. housing market underwent its most significant shock in generations. From 2008, the model might overlook a critical variable that shaped housing values, credit availability, and the underlying financial health of borrowers.

Our reply: We thank the reviewer for encouraging us to consider how the 2008 global financial crisis (GFC) may have impacted the underlying financial health and credit access of mortgage borrowers during the study period. We have included the following text in the revised manuscript to highlight how these factors (which were not explicitly modeled) are likely to influence our findings. It is worth noting that our property value model, which uses observed property sales as an input, reflects the effects of the GFC on home prices and (by extension) borrower equity.

Revised text from lines 905-914: Fourth, our model framework does not account for how factors related to the 2008 global financial crisis (GFC) may have impacted the financial health and credit access of mortgage borrowers during the study period. These include elevated rates of unemployment that persisted for several years following the GFC and a tightening of mortgage lending standards that reduced the availability of credit to property owners. Mortgage lending standards in the U.S. underwent a gradual loosening during the early 2000s leading up to the crisis, followed by a sharp tightening during the 2007-2009 period that led to increases in loan denial rates and more stringent LTV and DTI requirements (Vojtech et al., 2020). Of the seven flood events evaluated in this study, the effects of the GFC would be most relevant for Hurricane Irene, which occurred in 2011 when the economy was still recovering from the crisis. If the elevated rate of unemployment and reduced credit supply during this period were incorporated into our model, projections of credit constraints among borrowers exposed to flooding from Hurricane Irene would likely be higher.

**References**

Agarwal, S. and Ho, C. T.: Comparing the Prime and Subprime Mortgage Markets, Federal Reserve Bank of Chicago, 2007.

GAO: Prolonged Conservatorships of Fannie Mae and Freddie Mac Prompt Need for Reform, Government Accountability Office, Washington, D.C., 2019.

Haixiang, G., Yijing, L., Shang, J., Mingyun, G., Yuanyue, H., and Bing, G.: Learning from class-imbalanced data: Review of methods and applications, Expert Systems with Applications, 73, 220–239, https://doi.org/10.1016/j.eswa.2016.12.035, 2017.

He, J. and Cheng, M. X.: Weighting Methods for Rare Event Identification From Imbalanced Datasets, Front Big Data, 4, 715320, https://doi.org/10.3389/fdata.2021.715320, 2021.

Jones, K.: FHA-Insured Home Loans: An Overview, Congressional Research Service, 2022.

N.C. Department of Military and Veterans Affairs: Military Bases in North Carolina, https://www.milvets.nc.gov/benefits-services/military-bases-north-carolina#Fort Bragg, 2025.

Nolte, C., Boyle, K. J., Chaudhry, A. M., Clapp, C., Guignet, D., Hennighausen, H., Kushner, I., Liao, Y., Mamun, S., Pollack, A., Richardson, J., Sundquist, S., Swedberg, K., and Uhl, J. H.: Data Practices for Studying the Impacts of Environmental Amenities and Hazards with Nationwide Property Data, Land Economics, 100, 200–221, https://doi.org/10.3368/le.100.1.102122-0090R, 2024.

Perl, L.: VA Housing: Guaranteed Loans, Direct Loans, and Specially Adapted Housing Grants, Congressional Research Service, 2018.

Pollack, A. B. and Kaufmann, R. K.: Increasing storm risk, structural defense, and house prices in the Florida Keys, Ecological Economics, 194, 107350, https://doi.org/10.1016/j.ecolecon.2022.107350, 2022.

Vojtech, C. M., Kay, B. S., and Driscoll, J. C.: The real consequences of bank mortgage lending standards, Journal of Financial Intermediation, 44, 100846, https://doi.org/10.1016/j.jfi.2019.100846, 2020.